# GALAX: Graph-Augmented Language Model for Explainable Reinforcement-Guided Subgraph Reasoning in Precision Medicine

**Heming Zhang**[1], **Di Huang**[2], **Wenyu Li**[2], **Michael Province**[3]
**Yixin Chen**[2], **Philip Payne**[1], **Fuhai Li**[1,2*]
[1]I2DB, [2]Dept. of CSE, [3]Dept. of Genetics, Washington University in St. Louis
*Correspondence: `fuhai.li@wustl.edu`

## Abstract

In precision medicine, quantitative multi-omic features, topological context, and textual biological knowledge play vital roles in identifying disease-critical signaling pathways and targets, guiding the discovery of novel therapeutics and effective treatment strategies. Existing pipelines capture only one or two of these, thereby limiting mechanistic interpretability. Although Process Reward Models (PRMs) aim to guide reasoning in LLMs, they remain limited by coarse step definitions, unreliable intermediate evaluation, and vulnerability to reward hacking with added computational cost. These gaps motivate jointly integrating quantitative multi-omic signals, topological structure with node annotations, and literature-scale text via LLMs, using subgraph reasoning as the principle bridge linking numeric evidence, topological knowledge and language context. To resolve this challenge, we propose **GALAX** (**G**raph **A**ugmented **LA**nguage model with e**X**plainability), an innovative framework that integrates pretrained Graph Neural Networks (GNNs) into Large Language Models (LLMs) via reinforcement learning guided by a Graph Process Reward Model (GPRM), which generates disease-relevant subgraphs in a step-wise manner initiated by an LLM and iteratively evaluated by a pretrained GNN and schema-based rule check, enabling process-level supervision without explicit labels. As an application, we also introduced **Target-QA**, a benchmark combining CRISPR-identified targets, multi-omic profiles, and biomedical graph knowledge across diverse cancer cell lines, which enables GNN pretraining for supervising step-wise graph construction and supports long-context reasoning over text-numeric graphs (TNGs), providing a scalable and biologically grounded framework for explainable, reinforcement-guided subgraph reasoning toward reliable and interpretable target and pathway discovery in precision medicine.

## 1 Introduction

Identifying therapeutic targets and elucidating disease mechanisms are main challenges in precision medicine (Steyaert et al., 2023; Topol, 2019). CRISPR-based gene editing has revolutionized functional genomic by enabling high-throughput perturbation of gene function across diverse cellular contexts (Shalem et al., 2014; Li et al., 2023). In oncology, large-scale CRISPR screens in cancer cell lines and patient-derived models have revealed context-specific genetic vulnerabilities, providing a robust experimental foundation for biomarker and target discovery (Shi et al., 2015). Despite these advances, computationally predicting key targets from multi-omic profiles and interpreting their mechanistic role in disease progression remains difficult. In particular, bridging omic data with interpretable explanations of molecular mechanisms continues to be a critical unmet need (Zhang et al., 2024a). Traditional approaches, such as differential expression analysis or essentiality scoring, lack the capacity to model the hierarchical and cross-modal dependencies in molecular networks, often overlooking key regulatory redundancies and pathway-level dynamics. Recent graph-based models have shown promise in outcome prediction tasks (Ren et al., 2024), yet they typically lack the structured supervision necessary for accurate target prioritization and mechanism discovery and seldom jointly integrate quantitative multi-omic features, topological structure with node annotations, and literature-scale text—limiting mechanistic interpretability.

Meanwhile, Large Language Models (LLMs) have demonstrated strong capabilities in natural language understanding and reasoning, particularly through techniques like in-context learning (ICL) (Brown et al., 2020) and chain-of-thought (CoT) prompting (Wei et al., 2022), which enable multi-step reasoning. However, LLMs often suffer from hallucination and lack grounding in structured knowledge, especially in scientific domains. To mitigate these issues, Retrieval-Augmented Generation (RAG) (Lewis et al., 2020) and its graph-based variants, such as RoG (Luo et al., 2023), SubgraphRAG (Li et al., 2024), GNN-RAG (Mavromatis & Karypis, 2025) and G-Retriever (He et al., 2024), have been proposed to enhance LLM performance by incorporating external knowledge graph. Despite their utility, these approaches still focus on final answer accuracy and give little attention to the reliability of intermediate reasoning. The retrieved subgraphs are noisy, large, and lack ground-truth mechanistic structure, making supervised retrieval unstable. Most existing models also fail to integrate numerical omic signals, causing the loss of cell line–specific information needed for target discovery. On the other hand, the Process Reward Model (PRM) framework has been introduced to provide fine-grained supervision over intermediate steps in reasoning tasks (Luo et al., 2024; Lightman et al., 2023; Uesato et al., 2022; Wang et al., 2023). PRMs provide step-wise supervision by assigning intermediate rewards to reinforcement learning (RL) agents, forming the foundation for Large Reasoning Models (LRMs) trained with Reinforcement Learning with Human Feedback (RLHF) (Bai et al., 2022), Proximal Policy Optimization (PPO) (Schulman et al., 2017), and Group Relative Policy Optimization (GRPO) (Shao et al., 2024). For example, StepGRPO (Zhang et al., 2025b) extends GRPO by incorporating rule-based step-wise rewards to supervise each intermediate reasoning step, addressing the sparse reward problem and enhancing multi-step reasoning in multimodal language models. However, PRMs face key limitations in defining fine-grained reasoning steps, verifying intermediate correctness, and hacking for model-based rewards (Gao et al., 2023), further complicating training.

These challenges are amplified in biomedicine: reasoning over multi-omic, gene-regulatory text–numeric graphs (TNGs) lacks ground-truth stepwise annotations, making intermediate supervision infeasible, and the combinatorial explosion of biological paths renders exhaustive planning or retrieval impractical. We propose **GALAX** (**G**raph-**A**ugmented **LA**nguage model with e**X**plainability), which couples LLMs with a pretrained GNN under reinforcement learning guided by a **Graph Process Reward Model (GPRM)**. Instead of explicit labels, GALAX uses the GNN as a stepwise supervisor and schema-based rule term to check validity, scoring intermediate subgraphs (partial signaling cascades) for biological plausibility and cancer relevance to provide fine-grained, graph-based rewards. GALAX prompts an LLM to propose candidate targets from multi-omic profiles and partial knowledge graphs, then an RL graph generator assembles task-specific cancer subnetworks under GPRM scoring—translating language reasoning into interpretable graph construction and yielding mechanistically grounded, patient-specific subnetworks for target prioritization. To evaluate, we introduce **Target-QA**, a benchmark integrating multi-omic data, biomedical graph knowledge, and CRISPR screening outcomes across diverse cancer cell lines. Together, GALAX and Target-QA deliver a scalable, reinforcement-guided solution for interpretable, patient-specific target identification and disease-mechanism discovery.

## 2 RELATED WORK

**LLMs Augmented with Knowledge and Graph Structures** Prompt tuning has emerged as a lightweight and scalable method for adapting LLMs to downstream tasks without full finetuning (Korbak et al., 2023; Lester et al., 2021). While effective, it operates over flat text representations and struggles to incorporate structured domain knowledge or multi-modal signals. To address this, Retrieval-Augmented Generation (RAG) (Lewis et al., 2020) and graph-augmented approaches such as G-Retriever (He et al., 2024), RoG (Luo et al., 2023), SubgraphRAG (Li et al., 2024) and GNN-RAG (Mavromatis & Karypis, 2025) have been proposed. However, these methods depend on accurate subgraph retrieval and still lack support for reliable reasoning. Most existing models also fail to integrate numerical omic signals, causing the loss of cell line–specific information needed for target discovery and leaving them poorly suited for large, patient-specific text–numeric graphs such as multi-omic signaling networks.

**Reinforcement Learning for Step-wise Reasoning** Reinforcement learning has been instrumental in aligning LLM behavior through methods like RLHF (Bai et al., 2022), PPO (Schulman et al., 2017), and GRPO (Shao et al., 2024). The Process Reward Model (PRM) (Luo et al.,

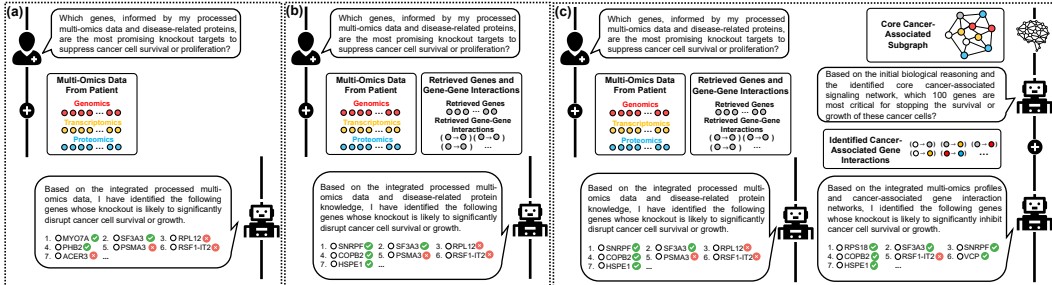

**Figure 1: Representative paradigms of patient-specific target prediction in-context LLM workflows and comparison with our approach. (a)** User-specific in-context prompts with basic query input. **(b)** Augmentation with retrieved knowledge relevant to the query. **(c)** GALAX framework: combining user-specific prompts, biomedical knowledge, and reinforcement-guided subgraph reasoning for explainable target prediction.

2024) enables step-wise supervision and has been adopted in Large Reasoning Models (LRMs). However, PRMs face key challenges: fine-grained step definitions are ambiguous, intermediate correctness is hard to validate, and model-based rewards can lead to reward hacking (Gao et al., 2023). These challenges are particularly acute in biomedical domains, where reasoning is inherently unstructured and lacks step-wise annotations, making conventional PRM pipelines impractical.

**Multi-omic Data Integration in Biomedical AI** From a biological standpoint, the integration of genomic, transcriptomic, and proteomic has been essential for understanding disease mechanisms and therapeutic vulnerabilities (Hasin et al., 2017; Kristensen et al., 2014). Traditional approaches rely on statistical fusion or dimensionality reduction (Meng et al., 2016; Shen et al., 2009; Rohart et al., 2017; Argelaguet et al., 2018; Nguyen & Wang, 2020), which overlook the hierarchical and interconnected nature of molecular data. More recently, GNN-based models like MOGONET (Wang et al., 2021) and MoGCN (Li et al., 2022) have demonstrated the value of structured graph reasoning for cancer subtype classification and biomarker identification. However, these models are primarily designed for outcome prediction and often fall short in identifying actionable biomarkers associated with specific disease mechanisms.

GALAX addresses the limitations of existing models by introducing a RL-guided framework that dynamically constructs biologically relevant subgraphs for each patient or cell line. This enables interpretable, context-sensitive target prioritization that adapts to both multi-omic features and disease-specific graph as the text-numeric format. To our knowledge, GALAX is the first to unify numerical multi-omic signals, literature-scale textual information, and biological topology under a reinforcement learning paradigm with biologically grounded supervision, which learns to reason through step-wise subgraph generation, guided by an authoritative biomedical GRPM.

## 3 PROBLEM FORMULATION

Identifying key targets and uncovering disease mechanisms remains a major challenge in precision medicine. To address this, we adopt the Text-Omic Signaling Graph (TOSG) (Zhang et al., 2025a), which integrates multi-omic features and biomedical context into a unified graph structure. Built upon TOSG, our model **GALAX** couples LLM-based hypothesis generation with reinforcement-guided subgraph reasoning to enable interpretable, patient-specific target predictions on the Target-QA benchmark.

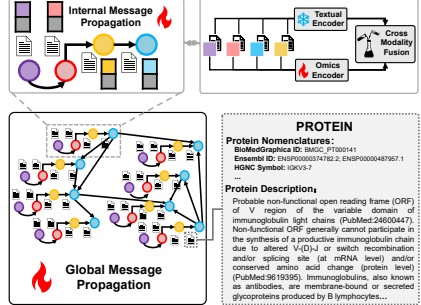

**Figure 2:** Demonstration of TOSG Construction and Graph Foundation Model

**TOSG Construction** We construct the TOSG by integrating three modalities: numerical omic evidence $\mathcal{X}^{(0)}$, textual entity descriptions $\mathcal{T}$, and topological information within a unified biomedical knowledge graph $\mathcal{G} = \{\mathcal{V}, \mathcal{E}\}$. Here, $\mathcal{V}$ represents the topological entities and $\mathcal{E}$ denotes the biological relations. The detailed formulation of each modality is presented as follows. **Feature Space:** Specifically, utilizing the DepMap as the source of numerical evidence (Dempster et al., 2019), we formulate this raw data as $\mathcal{X}^{(0)} = \{X_n^{(0)}\}_{n=1}^{N^{(0)}}$,

where $X_n^{(0)} \in \mathbb{R}^M$ comprises of $N^{(0)}$ samples with $M$ entities. These samples are paired with binary labels for cancerous or non-cancerous cell lines, denoted as $\mathcal{Y}^{(0)} \in \{0,1\}^{N^{(0)}}$. **Textual Space:** Each entity in the TOSG is associated with a name and description, represented as $\mathcal{T} = \{T_{\text{name}}, T_{\text{desc}}\}$, where $|T_{\text{name}}| = |T_{\text{desc}}| = M$. **Topological Space:** The $M$ entities in each sample are composed of multiple components derived from genomic, transcriptomic, and proteomic contexts. These multi-omic features are integrated into the TOSG using an existing integration tool, BioMed-Graphica (Zhang et al., 2024b), resulting in a text-attributed knowledge graph $\mathcal{G} = \{\mathcal{V}, \mathcal{E}\}$. The set of vertices is defined as $\mathcal{V} = \{\mathcal{V}^{(pm)}, \mathcal{V}^{(g)}, \mathcal{V}^{(t)}, \mathcal{V}^{(p)}\}$, representing promoter, gene, transcript, and protein entities, respectively. The size of each set is given by $|\mathcal{V}^{(pm)}| = m^{(pm)}$, $|\mathcal{V}^{(g)}| = m^{(g)}$, $|\mathcal{V}^{(t)}| = m^{(t)}$, and $|\mathcal{V}^{(p)}| = m^{(p)}$, such that $|\mathcal{V}| = m^{(pm)} + m^{(g)} + m^{(t)} + m^{(p)} = M$. Correspondingly, we map the cell-specific omic features for the $n$-th sample, denoted as $X_n^{(0)} \in \mathbb{R}^M$, using a compact vector concatenation of the four entity types: $X_n^{(0)} = \left[\mathbf{x}_n^{(pm)} \oplus \mathbf{x}_n^{(g)} \oplus \mathbf{x}_n^{(t)} \oplus \mathbf{x}_n^{(p)}\right]$, where $\mathbf{x}_n^{(\cdot)}$ represents the feature vector for the respective modality (e.g., $\mathbf{x}_n^{(p)} \in \mathbb{R}^{m^{(p)}}$ corresponds to protein levels). In detail, this graph can be decomposed into two subgraphs: $\mathcal{G}^{(\text{in})} = (\mathcal{V}^{(\text{in})}, \mathcal{E}^{(\text{in})})$ and $\mathcal{G}^{(\text{PPI})} = (\mathcal{V}^{(\text{PPI})}, \mathcal{E}^{(\text{PPI})})$. Here, $\mathcal{G}^{(\text{in})}$ captures the internal signaling processes for protein translation. As shown in Figure 2, internal propagation follows the central dogma (Crick, 1970): promoter (**purple**) $\rightarrow$ gene (**red**) $\rightarrow$ transcript (**yellow**) $\rightarrow$ protein (**blue**), with $\mathcal{V} = \mathcal{V}^{(\text{in})}$ and $|\mathcal{V}^{(\text{in})}| = M$. Meanwhile, $\mathcal{G}^{(\text{PPI})}$ represents the gene regulatory network structured around protein-protein interactions (PPI), where $\mathcal{V}^{(\text{PPI})} = \mathcal{V}^{(p)}$. In summary, the TOSG unifies numerical omic evidence, textual descriptions, and topological information into a Text-Numeric Graph, defined as $\mathcal{G} = \{\mathcal{X}^{(0)}, \mathcal{T}, \mathcal{V}, \mathcal{E}\}$.

**Target-QA Generation** We construct the feature set $\mathcal{X}$ by filtering for cancer cell lines that contain both comprehensive annotations and curated CRISPR-based target information, defining the collective multi-omic evidence as set $\mathcal{X} = \{X_n\}_{n=1}^N$, where $X_n \in \mathbb{R}^M$. Subsequently, to align this quantitative data with the reasoning capabilities of LLMs, we systematically structure the input context for each instance by integrating textual metadata, molecular profiles, and interaction graphs, detailed as follows. **Omics Information:** To augment context with patient-specific data, we incorporate the top-$K$ features for the LLM. Due to input token constraints of LLMs (Achiam et al., 2023) and the presence of CpG sites with methylation beta values saturated at 1, which dominate rankings and render top-$K$ selection uninformative, we derive a concise multi-omic representation $X_n^{(K)} = [\mathbf{g}_n^{(K)} \oplus \mathbf{t}_n^{(K)} \oplus \mathbf{p}_n^{(K)}]$ by extracting the top $K$ features from genomic, transcriptomic, and proteomic modalities. Here, $\mathbf{g}_n^{(K)}, \mathbf{t}_n^{(K)}$, and $\mathbf{p}_n^{(K)}$ denote the ordered lists of selected gene, transcript, and protein names, respectively, which correspond directly to the omic-related graph nodes $\mathcal{V}_n^{(\text{omic})}$. **Sample Information:** Disease entity has an associated name and textual description with $\mathcal{S} = \{S_{\text{name}}, S_{\text{desc}}\}$ where $|\mathcal{S}| = m^{(S)}$. Leveraging annotations from DepMap about cell lines, we mapped the cell line and disease names in samples to form the sets $\mathcal{C} = \{c_n\}_{n=1}^N$ and $\mathcal{S}' = \{s_n'\}_{n=1}^N$, respectively. **Disease-related Protein Subgraph:** For cell line, $c_n$, we use the cell line related disease entity $s_n'$ to retrieve from BioMedGraphica's disease-target interaction graph $\mathcal{G}^{(\text{DTI})} = \{\mathcal{V}^{(\text{DTI})}, \mathcal{E}^{(\text{DTI})}\}$, where $\mathcal{V}^{(\text{DTI})} = \{\mathcal{V}^{(S)}, \mathcal{V}^{(p)}\}$ includes disease and protein nodes. To provide structured graph context, we apply a subgraph retrieval strategy that extracts disease-relevant protein entities $\mathcal{V}_n^{(p)} \subset \mathcal{V}^{(p)}$ and their interactions $\mathcal{E}_n^{(\text{DTI})}$, along with $h$-hop protein neighbors $\mathcal{V}_n^{(h)}$ by extracting interactions from $\mathcal{E}^{(\text{PPI})}$, where $\mathcal{V}_n^{(h)} \subset \mathcal{V}^{(p)}$. This yields a sample-specific subgraph $\mathcal{G}_n^{(\text{sub})} = \{\mathcal{V}_n^{(\text{sub})}, \mathcal{E}_n^{(\text{sub})}\}$, where $\mathcal{V}_n^{(\text{sub})} = \mathcal{V}_n^{(p)} \cup \mathcal{V}_n^{(h)}$, with the full graph set $\mathcal{G}^{(\text{sub})} = \{\mathcal{G}_n^{(\text{sub})}\}_{n=1}^N$. Hence, each query $Q_n = \{c_n, s_n', X_n^{(K)}, \mathcal{G}_n^{(\text{sub})}\}$ is paired with answer $A_n$ describing top-$\gamma$ CRISPR targets $R_n = \{r_{n,1}, \ldots, r_{n,\gamma}\}$, yielding instance $D_n = (Q_n, A_n)$ in dataset $\mathcal{D} = \{D_n\}_{n=1}^N$, stratified by TCGA (Weinstein et al., 2013) types (e.g., $\mathcal{D}_{\text{LUAD}}, \mathcal{D}_{\text{BRCA}}$).

**Patient-Specific Target Prediction with Explainability** GALAX is designed to generate not only accurate but also interpretable predictions by explicitly modeling the reasoning process through subgraph construction (see Figure 3). Given a query $Q_n$, the model produces both a prioritized target list $\hat{A}_n$ and an explanatory subgraph $\mathcal{G}_n^\dagger$:

$$\hat{A}_n, \mathcal{G}_n^\dagger = f(Q_n, X_n, \mathcal{T}, \mathcal{E}; \theta_{\text{G}}, \theta_{\text{L}}) \tag{1}$$

The function $f$ is composed of three modules (see Section 4.2): (1) an initial language model $f_{\text{init}}$ that performs coarse reasoning and extracts candidate entities via initial answering; (2) a reinforcement-

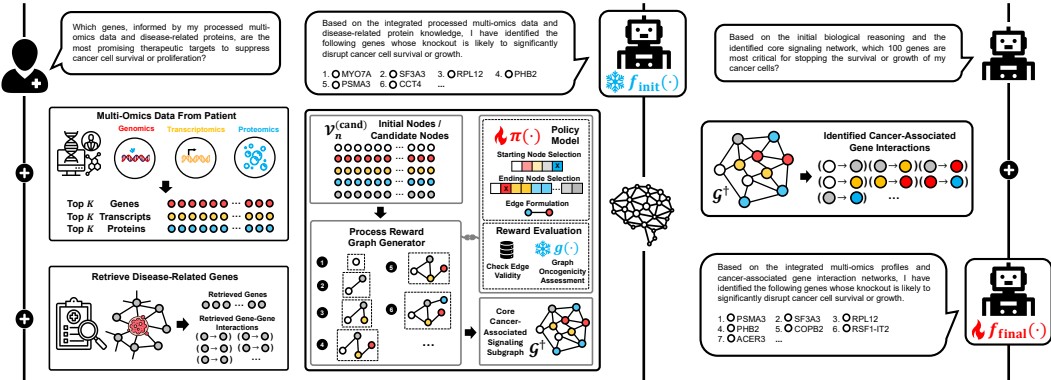

**Figure 3: An overview of the GALAX workflow.** Given processed multi-omic profiles (genomic, transcriptomic, proteomic), a subgraph is retrieved by identifying disease-associated proteins and their $h$-hop neighbors. Then $f_{\text{init}}(\cdot)$ proposes initial targets, refined by a reinforcement-guided graph generator $\pi(\cdot)$ supervised by a pretrained graph foundation model $g(\cdot)$. The final subgraph, $\mathcal{G}^\dagger$, combined with the query, is passed to a second-stage LLM $f_{\text{final}}(\cdot)$ for target prediction. The full pipeline enables explainable, patient-specific reasoning grounded in molecular biology and CRISPR evidence.

based graph generator $\pi(\cdot)$ that incrementally constructs the explainable subgraph $\mathcal{G}_n^\dagger$ under the guidance of a pretrained graph classifier $g(\cdot)$ parameterized by $\theta_{\text{G}}$, using step-wise biological plausibility rewards; and (3) a final language model $f_{\text{final}}$ that refines the prediction by reasoning over both the initial output and the generated subgraph context. The subgraph $\mathcal{G}_n^\dagger$ serves as a transparent rationale, offering deeper insights into the disease mechanism.

# 4 GALAX

## 4.1 FOUNDATION MODELS PRETRAINING

**LLM Pretraining** We pretrain a large language model, denoted as $f_{\text{L}}^{\text{pre}}$ with parameters $\theta_{\text{L}}^{\text{pre}}$, using curated text corpora. The input data comprises omic entity descriptions $\mathcal{T}$, disease annotations $\mathcal{S}$, protein–protein interactions $\mathcal{E}^{(\text{PPI})}$, and disease–target relationships $\mathcal{E}^{(\text{DTI})}$. As illustrated in Figure 5, this pretraining phase equips the model with foundational knowledge of biomedical terminology and relational structure (see Appendix C.1), thereby enhancing its capacity for downstream reasoning in biomedical tasks. Then we will continue pretraining the language model $f_{\text{init}}$ parameterized by $\theta_{\text{init}}$, which are detailed in Appendix C.2.

**Graph Foundation Model Pretraining** We pretrain the graph encoder $\theta_{\text{G}}^{\text{pre}}$ via a two-stage pipeline (see Figure 2). In stage one, a unified graph–language model $f_{\text{G}}^{\text{pre}}$ is trained over node attributes $\mathcal{X}^{(0)}$, textual features $\mathcal{T}$, and edge set $\mathcal{E}$, with protein–protein interaction edges stochastically masked as $\mathcal{E}_{\text{mask}} \sim \text{Bernoulli}(p)$, $p < 1$. The resulting representation is

$$\mathcal{H}^{\text{pre}} = f_{\text{G}}^{\text{pre}}(\mathcal{X}^{(0)}, \mathcal{T}, \mathcal{E}, \mathcal{E}_{\text{mask}}) \tag{2}$$

, where $\mathcal{H}^{\text{pre}} \in \mathbb{R}^{N^{(0)} \times M \times d^{(\text{pre})}}$ encodes contextualized entity states. In details, we generate edge mask $\mathcal{E}_{\text{mask}} \sim \text{Bernoulli}(p)$, where $p < 1$ is the ratio of the masked edges for $\mathcal{E}^{(\text{PPI})}$ to mask out the signaling flows in protein-protein interactions. Then, we apply internal message propagation with

$$\mathcal{H}_{\text{in}}^{\text{pre}} = \text{GNN}_{\text{in}}^{\text{pre}}(\text{ENC}_{\text{cross}}^{\text{pre}}(\mathcal{X}^{(0)}, \mathcal{T}), \mathcal{E}^{(\text{in})}) \tag{3}$$

, where $\text{ENC}_{\text{cross}}^{\text{pre}}$ is a cross-modal encoder to align textual and omic features and $\mathcal{H}_{\text{in}}^{\text{pre}} \in \mathbb{R}^{N^{(0)} \times M \times d^{(\text{in})}}$. The first-stage pretraining captures gene regulatory patterns by performing masked global message passing over $\mathcal{E}^{(\text{PPI})}$ by

$$\mathcal{H}^{\text{pre}} = \text{GNN}_{\text{PPI}}^{\text{pre}}(\mathcal{H}_{\text{in}}^{\text{pre}}, \mathcal{E}^{(\text{PPI})}, \mathcal{E}_{\text{mask}}) \tag{4}$$

In the second stage, downstream model $f_{\text{G}}$ is initialized with pretrained parameters $\theta_{\text{G}}^{\text{pre}}$ and is used to predict disease types from multi-omic inputs. The predicted class for each sample is given by:

$$\hat{\mathcal{Y}}^{(0)} = \arg\max_{o \in \mathcal{O}} \text{Softmax}\left[\text{MLP}_{\text{G}}\left(f_{\text{G}}(\mathcal{X}^{(0)}, \mathcal{T}, \mathcal{E}; \theta_{\text{G}}^{\text{pre}})\right)\right] \tag{5}$$

, where $\mathcal{O}$ denotes the set of disease types, and $f_{\mathrm{G}}$ consists of the same architecture as $\mathrm{ENC}_{\mathrm{cross}}$, $\mathrm{GNN}_{\mathrm{in}}$, and $\mathrm{GNN}_{\mathrm{PPI}}$. The model is trained to minimize the cross-entropy loss between the predicted probability distribution by $\mathrm{Softmax}[\mathrm{MLP}_{\mathrm{G}}(f_{\mathrm{G}}(\cdot))]$ and the ground-truth label. The pretrained graph foundation model $f_{\mathrm{G}}$ serves as a structural proxy to guide step-wise reasoning in downstream tasks.

## 4.2 Model Training

The pipeline of model couples initial answering, a reinforcement-guided subgraph generator, and final answering. Given the structured prompt $P_n^{(\mathrm{init})}$ from query $Q_n$, the pretrained model $f_{\mathrm{init}}$ outputs $A_n^{(\mathrm{init})}$ and a biomedical named entity recognition (NER) $\phi$ extracts entities $R_n^{(\mathrm{init})}$ and maps them to proteins $\mathcal{V}_n^{(\mathrm{init})}$, which will contribute to forming the start set $\mathcal{V}_n^{(\mathrm{start})}$ depending on situation. Node features are embedded by pretrained graph foundation model to obtain $\mathcal{H}_{\mathrm{in}}$ and formed $X_n^{(\mathrm{cand})}$. Subgraph construction is framed as reinforcement learning problem consisting of four elements: **state** $\mathcal{G}_n^{(i)} = (\mathcal{V}_n^{(i)}, \mathcal{E}_n^{(i)})$; **action** $\Delta_n^{(i)} = (v_{\mathrm{src}}^i, v_{\mathrm{tgt}}^i)$ adds a single edge under feasibility masks; **policy** uses a message propagation (MSG) module to produce $X_n^{(i)} = \pi_{\mathrm{MSG}}(\mathcal{G}_n^{(i)}, X_n^{(\mathrm{cand})})$ and two masked probability function $\pi_{\mathrm{SRC}}, \pi_{\mathrm{TGT}}$ to sample $v_{\mathrm{src}}^i$ and $v_{\mathrm{tgt}}^i$; **reward** combines feedback from a pretrained classifier $g(\cdot)$, a rollout averaging $L$ simulated continuations, and a rule-based term $\mathcal{R}_{\mathrm{rule}}$ that penalizes schema violations. We accept an action e only when $\mathcal{R}_{\mathrm{total}}^{(i)} > 0$ and update the generator with reward-weighted cross-entropy with early stopping; the best subgraph $\mathcal{G}_n^{\dagger}$ is retained in $\Omega$ stochastic run. For final answering, $\mathcal{G}_n^{\dagger}$ is verbalized in expert mode (Fatemi et al., 2023) and appended to $Q_n$ to form $P_n^{(\mathrm{final})}$; the model $f_{\mathrm{final}}$ is finetuned with token-level cross-entropy against $A_n$. What follows is a detailed exposition of the model design and training framework.

**Initial Answering** With the pretrained language model $f_{\mathrm{init}}$ based on $f_{\mathrm{L}}$, parameterized by $\theta_{\mathrm{init}}$, the input to the model is a structured prompt $P_n^{(\mathrm{init})}$ derived from the original query $Q_n$, specificially for turning $\mathcal{G}_n^{(\mathrm{sub})}$ into a graph expert format (see Appendix C.2 for details), and it will output $A_n^{(\mathrm{init})}$. Afterwards, an NER function, $\phi$, is applied to extract biomedical entities from $A_n^{(\mathrm{init})}$ by $R_n^{(\mathrm{init})} = \phi(A_n^{(\mathrm{init})})$, where $R_n^{(\mathrm{init})} = \{r_{n,1}^{(\mathrm{init})}, r_{n,2}^{(\mathrm{init})}, \ldots, r_{n,\alpha}^{(\mathrm{init})}\}$. And which are then mapped to corresponding protein nodes, $\mathcal{V}_n^{(\mathrm{init})} = \{v_{n,\mathrm{init},1}^{(p)}, v_{n,\mathrm{init},2}^{(p)}, \cdots, v_{n,\mathrm{init},\alpha'}^{(p)}\}$.

**Process Reward Graph Generator** The initial node set $\mathcal{V}_n^{(\mathrm{start})}$ is selected based on a predefined priority: if the disease-related protein set $\mathcal{V}_n^{(p)}$ is available, the top $\eta$ most relevant entities are used; if not, the top $\eta$ entities from the initialization set $\mathcal{V}_n^{(\mathrm{init})}$ are selected. If both are unavailable, $\eta$ nodes are randomly sampled from the omic-derived set $\mathcal{V}_n^{(\mathrm{omic})}$. Formally,

$$\mathcal{V}_n^{(\mathrm{start})} = \begin{cases} \mathrm{Top}\text{-}\eta(\mathcal{V}_n^{(p)}), & \text{if } \mathcal{V}_n^{(p)} \neq \emptyset \\ \mathrm{Top}\text{-}\eta(\mathcal{V}_n^{(\mathrm{init})}), & \text{else if } \mathcal{V}_n^{(\mathrm{init})} \neq \emptyset \\ \mathrm{Sample}\text{-}\eta(\mathcal{V}_n^{(\mathrm{omic})}), & \text{otherwise} \end{cases} \qquad (6)$$

The candidate set was generated based on $\mathcal{V}_n^{(\mathrm{cand})} = \mathcal{V}_n^{(\mathrm{init})} \cup \mathcal{V}_n^{(\mathrm{sub})} \cup \mathcal{V}_n^{(\mathrm{omic})}$ and $\mathcal{V}_n^{(\mathrm{start})} \subset \mathcal{V}_n^{(\mathrm{cand})}$. And the features of $\mathcal{X}$ are precomputed using a graph encoder $f_{\mathrm{G}}^{\mathrm{pre}}$ with parameters $\theta_{\mathrm{G}}^{\mathrm{pre}}$, followed by pretrained modules $\mathrm{ENC}_{\mathrm{cross}}$ and $\mathrm{GNN}_{\mathrm{in}}$ with parameters $\theta_{\mathrm{cross}}^{\mathrm{G}}$ and $\theta_{\mathrm{in}}^{\mathrm{G}}$, respectively:

$$\mathcal{H}_{\mathrm{in}} = \mathrm{GNN}_{\mathrm{in}} \left( \mathrm{ENC}_{\mathrm{cross}} \left( f_{\mathrm{G}}^{\mathrm{pre}}(\mathcal{X}, \mathcal{T}, \mathcal{E}; \theta_{\mathrm{G}}^{\mathrm{pre}}), \mathcal{T}; \theta_{\mathrm{cross}}^{\mathrm{G}} \right), \mathcal{E}^{(\mathrm{in})}; \theta_{\mathrm{in}}^{\mathrm{G}} \right) \in \mathbb{R}^{N \times M \times d^{(\mathrm{in})}} \qquad (7)$$

The candidate node features $X_n^{(\mathrm{cand})}$ are selected from the precomputed representation $\mathcal{H}_{\mathrm{in}}$. Then we introduce a reinforcement-guided graph generator, denoted as $\pi(\cdot)$, where the policy operates over sample-specific graph states and candidate sets, enabling personalized subgraph construction. At step $i$, the current graph state is defined as $\mathcal{G}_n^{(i)} = \{\mathcal{V}_n^{(i)}, \mathcal{E}_n^{(i)}\}$ after applying the action of constructing edge $(v_{\mathrm{src}}^i, v_{\mathrm{tgt}}^i)$. The next graph state $\mathcal{G}_n^{(i+1)}$ is formed by adding an edge between a sampled source node $v_{\mathrm{src}}^i$ and target node $v_{\mathrm{tgt}}^i$. To compute the probabilities of selecting these nodes, we embed the node features from $\mathcal{G}_n^{(i)}$ together with the candidate features $X_n^{(\mathrm{cand})}$ as:

$$X_n^{(i)} = \pi_{\mathrm{MSG}}(\mathcal{G}_n^{(i)}, X_n^{(\mathrm{cand})}) \qquad (8)$$

, where $\pi_{\text{MSG}}$ consists of a message propagation (MSG) module and a feature selection mechanism that extracts node embeddings corresponding to $\mathcal{V}_n^{(i)}$ from the propagated features. Based on embeddings, the source and target nodes are sampled according to the generated probabilities with

$$v_{\text{src}}^i \sim \pi_{\text{SRC}}(X_n^{(i)}, \mathcal{M}_{\text{SRC}}); \quad v_{\text{tgt}}^i \sim \pi_{\text{TGT}}(X_n^{(i)}, \mathcal{M}_{\text{TGT}}; v_{\text{src}}^i) \tag{9}$$

, where $\pi_{\text{SRC}}$ and $\pi_{\text{TGT}}$ are returned with probability implemented via MLPs and softmax by masking $\mathcal{M}_{\text{SRC}}$ and $\mathcal{M}_{\text{TGT}}$, which restrict source selection to nodes in $\mathcal{G}_n^{(i)}$ and exclude the source when selecting the target. Based on the probability, nodes $v_{\text{src}}^i$ and $v_{\text{tgt}}^i$ will be selected by the sampling function. The selected node pair then forms the updated graph state $\mathcal{G}_n^{(i+1)}$ for next step.

**Reinforcement-Guided Reward and Training** To guide the graph generation process, we define a rollout-based reward function that combines immediate classifier feedback with future trajectory simulation. At generation step $i$, the intermediate graph is represented as $\mathcal{G}_n^{(i+1)} = \{\mathcal{V}_n^{(i+1)}, \mathcal{E}_n^{(i+1)}\}$ after applying the action of constructing edge $(v_{\text{src}}^i, v_{\text{tgt}}^i)$. We define $g(\cdot)$ as a pretrained graph classifier that computes class probabilities by applying a GNN encoder $\text{GNN}_{\text{PPI}}^{\text{G}}$ followed by a projection head $\text{MLP}_{\text{G}}$, formally expressed as $g(\cdot) = \text{Softmax}\left[\text{MLP}_{\text{G}}\left(\text{GNN}_{\text{PPI}}^{\text{G}}(\cdot)\right)\right]$. The model is parameterized by pretrained weights $\theta_{\text{PPI}}^{\text{G}}$ and $\theta_{\text{MLP}}^{\text{G}}$, and outputs a probability distribution over classes in $\mathcal{O}$. Let $o^\star \in \mathcal{O}$ denote the target class. The reward $\mathcal{R}_n^{(i)}$ for step $i$ is defined as:

$$\mathcal{R}_n^{(i)} = g_{o^\star}(\mathcal{G}_n^{(i+1)}) - \frac{1}{|\mathcal{O}|} + \lambda \cdot \frac{1}{L} \sum_{\ell=1}^{L} \left[ g_{o^\star}(\text{Rollout}_\ell(\mathcal{G}_n^{(i+1)})) - \frac{1}{|\mathcal{O}|} \right] \tag{10}$$

Here, $g_{o^\star}(\mathcal{G})$ refers to the probability assigned to the target class $o^\star$, and $\text{Rollout}_\ell(\cdot)$ simulates the $\ell$-th full trajectory by continuing generation from the current partial graph using the current policy. The hyperparameter $\lambda$ balances intermediate and future rollout-based feedback. To ensure reasoning aligning with biological plausibility, we incorporate a rule-based reward term $\mathcal{R}_{\text{rule}}(\mathcal{G}_n^{(i+1)})$ that penalizes invalid edges according to relations from BioMedGraphica. The final reward is:

$$\mathcal{R}_{\text{total}}^{(i)} = \mathcal{R}_n^{(i)} + \lambda_{\text{rule}} \cdot \mathcal{R}_{\text{rule}}(\mathcal{G}_n^{(i+1)}) \tag{11}$$

This formulation guides the generation process toward subgraphs that are both predictive of the target class and consistent with domain-specific biological priors. And we used the greedy acceptance where if $\mathcal{R}_{\text{total}}^{(i)} > 0$, set $\mathcal{G}^{(i+1)}$ as current state; otherwise keep the previous state. Then in each step $i$, the model will be trained with loss function,

$$\mathcal{L}_{\text{step}} = -\mathcal{R}_{\text{total}}^{(i)}[\text{CE}(v_{\text{src}}^i, \pi_{\text{SRC}}(X_n^{(i)}, \mathcal{M}_{\text{SRC}})) + \text{CE}(v_{\text{tgt}}^i, \pi_{\text{TGT}}(X_n^{(i)}, \mathcal{M}_{\text{TGT}}; v_{\text{src}}^i))] \tag{12}$$

, where the generator is optimized with reward-weighted cross-entropy (CE) function. In practice, we sample multiple candidate subgraphs under the policy parameterized by $\theta_\pi$ across multiple runs $\Omega$, and select the optimal subgraph $\mathcal{G}_n^\dagger$.

**Final Answer Generation with Prompt Tuning** The optimal subgraph $\mathcal{G}_n^\dagger$ is converted into a structured textual description via expert mode (see details in Appendix D.1), which will be appended to original query $Q_n$ to form final graph-augmented prompt $P_n^{(\text{final})}$. With language model $f_{\text{final}}$ based on pretrained $f_{\text{init}}$, it generates the output sequence $\hat{A}_n = \{\hat{a}_{n,1}, \hat{a}_{n,2}, \ldots, \hat{a}_{n,J'}\}$ according to:

$$\xi_{\theta_{\text{final}}}(\hat{A}_n \mid Q_n, \mathcal{G}_n^\dagger) = \prod_{j=1}^{J'} \xi_{\theta_{\text{final}}}(\hat{a}_{n,j} \mid \hat{a}_{n,<j}, P_n^{(\text{final})}) \tag{13}$$

To align the model output with the refined ground-truth answer $A_n$, which contains the top $\gamma$ CRISPR-prioritized gene targets for sample $n$, we finetune the model by:

$$\mathcal{L}_{\text{final}} = -\sum_{n=1}^{N} \sum_{j=1}^{J'} \log \xi_{\theta_{\text{final}}}(a_{n,j} \mid a_{n,<j}, P_n^{(\text{final})}) \tag{14}$$

This objective encourages the model to internalize the structured reasoning encoded in $\mathcal{G}_n^\dagger$ to generate biologically grounded answers. After generation, an NER function $\phi$ will extract entities from the model's output $\hat{A}_n$ with $\hat{R}_n = \phi(\hat{A}_n) = \{\hat{r}_{n,1}, \hat{r}_{n,2}, \ldots, \hat{r}_{n,\beta}\}$. These predicted protein targets are used for evaluating biological relevance and overlap within reference targets in $A_n$.

## 5 EXPERIMENTS

**Datasets** We construct the Target-QA dataset over multiple TCGA cancer types using DepMap, integrating multi-omic features (epigenomic, genomic, transcriptomic, proteomic) and metadata from cancer cell lines. Each QA pair consists of an input query, including multi-omic and cell line information, and an output answer comprising the top-$\gamma$ ($\gamma = 100$) CRISPR-prioritized targets. The final dataset contains 363 QA pairs from cancerous cell lines after integration and preprocessing. We use an 80/20 train-test split and repeat experiments across four randomized seeds to ensure stability and generalization. For foundation model pretraining, we collect dataset of 336 samples with multi-omic features from unannotated samples, including both disease and control groups (297 cancerous, 39 non-cancerous), and apply stratified sampling to address class imbalance. Full data processing and cohort composition are described in Appendix B.

**Experimental Setup** We initialize the language model with LLaMA3-8B-Instruct (Grattafiori et al., 2024), pretrained on biomedical terminology and curated textual descriptions involving protein–protein and disease–protein relationships from BioMedGraphica to enhance domain-specific vocabulary and biological context understanding (see Figure 5). For graph encoding, we use BioBERT-v1.1 (Lee et al., 2020) for text embeddings and Graph Attention Networks (GAT) (Veličković et al., 2017) to learn topological features from protein interaction graphs, incorporating random edge masking to improve robustness (see Figure 2 global message propagation). The pretrained GNN achieves 64.4% AUC in edge prediction, and 99.46% / 96.15% accuracy on disease type classification (train/test). Named entities are extracted using GPT-4o-mini via ChatGPT API (Hurst et al., 2024).GALAX is trained using the Adam optimizer on two NVIDIA H100 GPUs (80GB). We set the number of top omic features per modality to $K=10$, the maximum subgraph rollout depth to $L=5$, and the number of candidate starting nodes $\eta=20$. The reward formulation includes both rollout- and rule-based components, each weighted equally with $\lambda=1$ and $\lambda_{\text{rule}}=1$. The reasoning task is formulated as a binary classification problem ($|\mathcal{O}|=2$), using a 1-hop ($h=1$) protein neighborhood from the disease-annotated subgraph. Model outputs are evaluated using precision, recall, F1-score, Jaccard similarity, Hit@5 and Hit@10, by comparing predicted target sets $\hat{R}_n$ against reference labels $R_n$. Additional details are provided in the Appendix D.2.

**Table 1:** Performance of models across datasets and metrics

| Model | Overall | | LUAD | | BRCA | |
|---|---|---|---|---|---|---|
| | Precision ↑ | Recall ↑ | Precision ↑ | Recall ↑ | Precision ↑ | Recall ↑ |
| M2T | 0.0016 | 0.0011 | 0.0020 | 0.0014 | 0.0000 | 0.0000 |
| GAT | 0.0006±0.0000 | 0.0006±0.0000 | 0.0000±0.0000 | 0.0000±0.0000 | 0.0033±0.0000 | 0.0033±0.0000 |
| L3 + Omics | 0.0071±0.0032 | 0.0013±0.0002 | 0.0079±0.0137 | 0.0005±0.0008 | 0.0020±0.0035 | 0.0017±0.0029 |
| L3 + Omics + KG | 0.0125±0.0032 | 0.0029±0.0003 | 0.0014±0.0025 | 0.0010±0.0016 | 0.0073±0.0068 | 0.0033±0.0029 |
| L3-FT(Med) + Omics | 0.0179±0.0045 | 0.0133±0.0064 | 0.0091±0.0018 | 0.0105±0.0044 | 0.0110±0.0106 | 0.0106±0.0106 |
| L3-FT(Med) + Omics + KG | 0.0158±0.0030 | 0.0058±0.0011 | 0.0081±0.0071 | 0.0024±0.0016 | 0.0149±0.0057 | 0.0050±0.0000 |
| L3-FT(QA) + Omics | 0.5250±0.0282 | 0.4959±0.0435 | 0.5201±0.0408 | 0.4905±0.0532 | 0.5074±0.0498 | 0.4856±0.0570 |
| L3-FT(QA) + Omics + KG | 0.5185±0.0240 | 0.4908±0.0402 | 0.5214±0.0242 | 0.4952±0.0432 | 0.4856±0.0395 | 0.4656±0.0436 |
| G-Retriever + pre-GAT | 0.4763±0.0004 | 0.3929±0.0063 | 0.4642±0.0181 | 0.3881±0.0264 | 0.4414±0.0099 | 0.3772±0.0010 |
| RoG | 0.5248±0.0134 | 0.4726±0.0445 | 0.5213±0.0227 | 0.4562±0.0848 | 0.4791±0.0575 | 0.4311±0.0721 |
| SubgraphRAG | 0.5280±0.0044 | 0.4617±0.0027 | 0.5123±0.0105 | 0.4448±0.0386 | 0.4708±0.0317 | 0.3917±0.0376 |
| GNN-RAG | 0.5258±0.0126 | 0.4735±0.0190 | 0.5334±0.0225 | 0.5052±0.0170 | 0.4787±0.0453 | 0.4389±0.0584 |
| **GALAX** | **0.5472±0.0053** | 0.5332±0.0031 | 0.5345±0.0185 | 0.5157±0.0043 | **0.5608±0.0031** | **0.5533±0.0033** |
| GALAX (Qwen2.5-7B) | 0.5445±0.0114 | **0.5405±0.0101** | **0.5475±0.0019** | **0.5462±0.0111** | 0.5171±0.0474 | 0.5206±0.0419 |

**Table 2:** Hit@10 and Hit@5 for models across datasets

| Model | Overall | | LUAD | | BRCA | |
|---|---|---|---|---|---|---|
| | Hit@10 ↑ | Hit@5 ↑ | Hit@10 ↑ | Hit@5 ↑ | Hit@10 ↑ | Hit@5 ↑ |
| M2T | 0.0029 | 0.0000 | 0.0000 | 0.0000 | 0.0000 | 0.0000 |
| GAT | 0.0000±0.0000 | 0.0000±0.0000 | 0.0000±0.0000 | 0.0000±0.0000 | 0.0000±0.0000 | 0.0000±0.0000 |
| L3 + Omics | 0.0021±0.0037 | 0.0032±0.0055 | 0.0048±0.0082 | 0.0095±0.0165 | 0.0000±0.0000 | 0.0000±0.0000 |
| L3 + Omics + KG | 0.0122±0.0033 | 0.0085±0.0037 | 0.0000±0.0000 | 0.0000±0.0000 | 0.0056±0.0096 | 0.0111±0.0192 |
| L3-FT(Med) + Omics | 0.0122±0.0072 | 0.0116±0.0097 | 0.0000±0.0000 | 0.0000±0.0000 | 0.0111±0.0192 | 0.0000±0.0000 |
| L3-FT(Med) + Omics + KG | 0.0132±0.0040 | 0.0106±0.0048 | 0.0048±0.0082 | 0.0095±0.0165 | 0.0111±0.0192 | 0.0000±0.0000 |
| L3-FT(QA) + Omics | 0.8693±0.0157 | 0.8889±0.0168 | 0.8667±0.0218 | 0.8476±0.0165 | 0.8389±0.0096 | **0.8889±0.0509** |
| L3-FT(QA) + Omics + KG | 0.8529±0.0153 | 0.8794±0.0114 | 0.8048±0.0541 | 0.7905±0.0436 | 0.8222±0.0347 | 0.8778±0.0192 |
| G-Retriever + pre-GAT | 0.8550±0.0046 | 0.8804±0.0037 | 0.8524±0.0165 | 0.8857±0.0000 | **0.8667±0.0000** | 0.8667±0.0000 |
| RoG | 0.8450±0.0350 | 0.8593±0.0318 | 0.8238±0.0218 | 0.8095±0.0436 | 0.7611±0.1110 | 0.7667±0.0577 |
| SubgraphRAG | 0.8476±0.0167 | 0.8624±0.0120 | 0.8238±0.0082 | 0.8190±0.0165 | 0.7333±0.1014 | 0.7556±0.0839 |
| GNN-RAG | 0.8323±0.0205 | 0.8656±0.0302 | 0.7571±0.0623 | 0.7905±0.0719 | 0.8222±0.0674 | 0.8444±0.0385 |
| **GALAX** | 0.8815±0.0033 | **0.9249±0.0048** | **0.8810±0.0082** | **0.9238±0.0436** | 0.8500±0.0441 | **0.8889±0.0839** |
| GALAX (Qwen2.5-7B) | **0.8841±0.0126** | 0.9079±0.0084 | 0.8667±0.0082 | 0.9048±0.0165 | 0.8000±0.0764 | 0.8556±0.0385 |

**Baseline Models** Traditional method, **M2T** (Multiomic2Target(Deng et al., 2024)), serves as a baseline that uses only multi-omic features without graph or language modeling and performs poorly across all metrics. And we perform ablation studies to isolate the contribution of core components of language and graph modules in GALAX. On the language axis, a non–task-tuned LLaMA3 (**L3+Omics**) is weak; domain-adaptive finetuning on biomedical text (**L3-FT(Med)+Omics**) yields modest gains; task-adaptive finetuning on Target-QA (**L3-FT(QA)+Omics**) produces the step-change. On the graph axis, **GAT** incorporates graph foundation model and trained to predict the

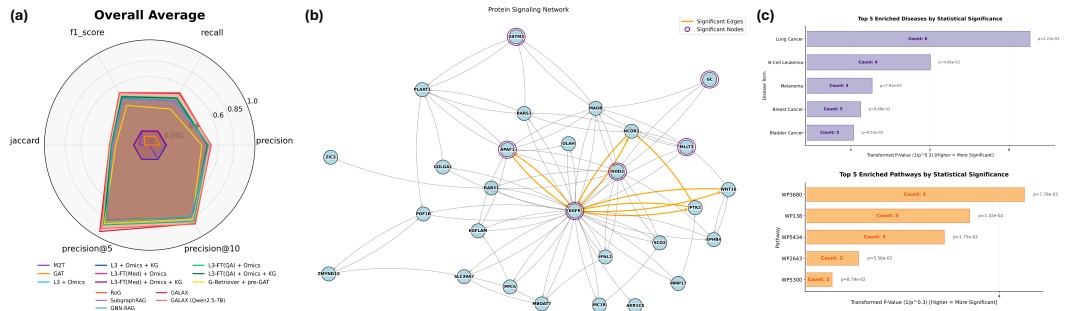

**Figure 4: Model performance and analysis.** **(a)** Overall performance across metrics. **(b)** LUAD (ACH-000860): cancer-relevant subgraph highlighting disease-associated nodes (**purple**) and enrichment-supported edges (**orange**). **(c)** Enrichment analysis: top pathway and disease terms with $p$-values and gene counts.

CRISPR knockout effects, shows limited improvements. Furthermore, integrating graph modules into language models comes with different outputs. Adding a static KG to each language foundation models (**[L3 / L3-FT(Med) / L3-FT(QA)]+Omics+KG**) barely improve or even decrease model performances, and graph retrieval with pretrained GAT (**G-Retriever+pre-GAT**) outperform some task-adaptive finetuned language models but not reliably due to the difficulty of extracting relevant subgraphs from millions of nodes/edges. **RoG**, **SubgraphRAG** (Li et al., 2024), and **GNN-RAG** (Mavromatis & Karypis, 2025) augment the language model by retrieving optimal paths, achieving moderate improvements over L3-FT(QA)+Omics+KG. Reinforcement-guided subgraph construction on top of QA-tuned language (**GALAX: L3-FT(QA)+Omics+KG+RL**) delivers consistent, cross-dataset gains of roughly 2%-5% on each metric, indicating that a reinforcement-guided subgraph generator outperforms other graph augmented models across all datasets by enabling process-level reasoning over biologically plausible subgraphs (shown in Tables 1–2). Details of baseline models are provided in Appendix D.3.

**Computational Complexity** We denote the language model complexity by $O(\kappa)$. The retrieved KG subgraph includes $M$ nodes, and $\varepsilon$ denotes the graph embedding cost. When augmented with KG retrieval, L3-FT(QA)+Omics+KG introduces an additional $O(M^2)$ per query, yielding $O(\kappa + M^2)$ for both training and inference. G-retriever+preGAT embeds all $M$ nodes and thus incurs $O(M\varepsilon + M^2\varepsilon)$. RoG and GNN-RAG follow the same cost at retrieval, since they embed entities and relations, requir-

| Model | Training & Inference |
|---|---|
| L3-FT(QA)+Omics | $O(\kappa)$ |
| L3-FT(QA)+Omics+KG | $O(\kappa + M^2)$ |
| SubgraphRAG | $O(\kappa + M^2\varepsilon)$ |
| G-retriever+preGAT | $O(\kappa + M\varepsilon + M^2\varepsilon)$ |
| RoG | $O(\kappa + M\varepsilon + M^2\varepsilon)$ |
| GNN-RAG | $O(\kappa + M\varepsilon + M^2\varepsilon)$ |
| GALAX | $O(\kappa + M\varepsilon + M^2\varepsilon)$ |

**Table 3: Complexity comparisons**

ing $O(M\varepsilon + M^2\varepsilon)$. SubgraphRAG reduces this by retrieving only relation triplets, which costs $O(M^2\varepsilon)$ at retrieval. And G-retriever. RoG, GNN-RAG and SubgraphRAG all requires an $O(\kappa)$ at both training and inference for language models. GALAX augments the language model with reinforcement-guided subgraph construction, with embedding cost $O(M\varepsilon + M^2\varepsilon)$. The model requires an $O(\kappa)$ forward pass to initialize the top $\eta$ candidates and another for final answer generation; since $\eta \ll M$, the graph-embedding term dominates the RL reward cost. Overall, the training and inference complexity of GALAX is therefore $O(\kappa + M\varepsilon + M^2\varepsilon)$ (see Table 3).

**Main Results** Tables 1–2 summarize the performance of GALAX and several competitive baselines on the full test dataset. GALAX outperforms all baselines on every metric, reaching an overall precision of **0.5472** and recall of **0.5332**. To further assess target prioritization quality, we report Hit@10 and Hit@5 in Table 2, where GALAX again achieves the highest accuracy with an overall Hit@10 of **0.8815** and Hit@5 of **0.9249**. We also replaced the backbone with Qwen2.5–7B–Instruct and observed similar performances on all metrics, indicating that GALAX maintains stable performance under backbone changes (See Appendix D.4 for full experiment results). We further tested generalization by forming three holdout sets in which the selected TCGA cancer types were excluded from training and treated as unseen during evaluation. Across all holdout sets, GALAX showed only modest performance declines, indicating that it preserves strong accuracy and generalizes reliably to previously unseen cancer types (Details are provided in Appendix D.5). Beyond Target-QA, we evaluate GALAX on the pediatric cancer dataset from PedDep (Dharia et al., 2021), which offers multi-omic profiles and CRISPR-based targets for 31 tumor cell lines, using zero-shot inference due to its small sample size. Even under this setting, GALAX surpasses all baselines on all metrics,

**Table 4:** Overall performances for ablation studies on omic inputs and KG structure

| Config | Setting | Recall ↑ | Precision ↑ | Hit@5 ↑ | Hit@10 ↑ |
|--------|---------|----------|-------------|---------|----------|
| **GALAX** | | | | | |
| E20 | Drop 20% edges | $0.5061_{\pm 0.0268}$ | $0.5362_{\pm 0.0059}$ | $0.9005_{\pm 0.0128}$ | $0.8741_{\pm 0.0060}$ |
| E40 | Drop 40% edges | $0.2871_{\pm 0.0060}$ | $0.5079_{\pm 0.0039}$ | $0.8762_{\pm 0.0138}$ | $0.8471_{\pm 0.0090}$ |
| E60 | Drop 60% edges | $0.2753_{\pm 0.0020}$ | $0.4961_{\pm 0.0059}$ | $0.8635_{\pm 0.0084}$ | $0.8307_{\pm 0.0033}$ |
| E80 | Drop 80% edges | $0.2775_{\pm 0.0134}$ | $0.4943_{\pm 0.0177}$ | $0.8434_{\pm 0.0524}$ | $0.8185_{\pm 0.0494}$ |
| N20 | Drop 20% nodes | $0.2697_{\pm 0.0013}$ | $0.5034_{\pm 0.0014}$ | $0.8786_{\pm 0.0070}$ | $0.8560_{\pm 0.0092}$ |
| N40 | Drop 40% nodes | $0.2675_{\pm 0.0033}$ | $0.4901_{\pm 0.0035}$ | $0.8878_{\pm 0.0066}$ | $0.8503_{\pm 0.0048}$ |
| N60 | Drop 60% nodes | $0.2617_{\pm 0.0056}$ | $0.4929_{\pm 0.0045}$ | $0.8698_{\pm 0.0290}$ | $0.8385_{\pm 0.0247}$ |
| N80 | Drop 80% nodes | $0.2653_{\pm 0.0032}$ | $0.4825_{\pm 0.0090}$ | $0.8341_{\pm 0.0105}$ | $0.8103_{\pm 0.0134}$ |
| Omic-M | Remove epigenomic data | $0.4810_{\pm 0.0137}$ | $0.5163_{\pm 0.0086}$ | $0.8857_{\pm 0.0145}$ | $0.8614_{\pm 0.0115}$ |
| Omic-G | Remove genomic data | $0.3121_{\pm 0.0052}$ | $0.4277_{\pm 0.0056}$ | $0.8550_{\pm 0.0037}$ | $0.8402_{\pm 0.0033}$ |
| Omic-T | Remove transcriptomic data | $0.3377_{\pm 0.0016}$ | $0.4065_{\pm 0.0042}$ | $0.8720_{\pm 0.0037}$ | $0.8672_{\pm 0.0037}$ |
| Omic-P | Remove proteomic data | $0.3347_{\pm 0.0013}$ | $0.3980_{\pm 0.0058}$ | $0.8540_{\pm 0.0138}$ | $0.8466_{\pm 0.0040}$ |
| Omic-All | Remove all omics | $0.3024_{\pm 0.0032}$ | $0.3793_{\pm 0.0019}$ | $0.8237_{\pm 0.0066}$ | $0.7967_{\pm 0.0040}$ |
| GALAX | Original | $0.5332_{\pm 0.0031}$ | $0.5472_{\pm 0.0053}$ | $0.9249_{\pm 0.0048}$ | $0.8815_{\pm 0.0033}$ |
| **L3-FT(QA) + Omic + KG** | | | | | |
| E20 | Drop 20% edges | $0.4599_{\pm 0.0820}$ | $0.5214_{\pm 0.0036}$ | $0.8587_{\pm 0.0157}$ | $0.8373_{\pm 0.0258}$ |
| E40 | Drop 40% edges | $0.2742_{\pm 0.0090}$ | $0.5064_{\pm 0.0099}$ | $0.8429_{\pm 0.0173}$ | $0.8226_{\pm 0.0169}$ |
| E60 | Drop 60% edges | $0.2676_{\pm 0.0062}$ | $0.4991_{\pm 0.0099}$ | $0.8296_{\pm 0.0305}$ | $0.8111_{\pm 0.0175}$ |
| E80 | Drop 80% edges | $0.2611_{\pm 0.0074}$ | $0.4880_{\pm 0.0086}$ | $0.8254_{\pm 0.0361}$ | $0.8063_{\pm 0.0370}$ |
| N20 | Drop 20% nodes | $0.2662_{\pm 0.0059}$ | $0.4916_{\pm 0.0024}$ | $0.8434_{\pm 0.0186}$ | $0.8222_{\pm 0.0193}$ |
| N40 | Drop 40% nodes | $0.2658_{\pm 0.0049}$ | $0.4838_{\pm 0.0169}$ | $0.8709_{\pm 0.0422}$ | $0.8339_{\pm 0.0335}$ |
| N60 | Drop 60% nodes | $0.2648_{\pm 0.0000}$ | $0.4742_{\pm 0.0150}$ | $0.7857_{\pm 0.0247}$ | $0.7690_{\pm 0.0303}$ |
| N80 | Drop 80% nodes | $0.2689_{\pm 0.0042}$ | $0.4722_{\pm 0.0063}$ | $0.7111_{\pm 0.0055}$ | $0.6974_{\pm 0.0111}$ |
| Omic-M | Remove epigenomic data | $0.4602_{\pm 0.0361}$ | $0.4878_{\pm 0.0256}$ | $0.8794_{\pm 0.0055}$ | $0.8466_{\pm 0.0142}$ |
| Omic-G | Remove genomic data | $0.3213_{\pm 0.0047}$ | $0.3962_{\pm 0.0098}$ | $0.8455_{\pm 0.0073}$ | $0.8349_{\pm 0.0136}$ |
| Omic-T | Remove transcriptomic data | $0.3244_{\pm 0.0034}$ | $0.3996_{\pm 0.0087}$ | $0.8550_{\pm 0.0073}$ | $0.8381_{\pm 0.0097}$ |
| Omic-P | Remove proteomic data | $0.3266_{\pm 0.0012}$ | $0.3872_{\pm 0.0004}$ | $0.8497_{\pm 0.0048}$ | $0.8265_{\pm 0.0056}$ |
| Omic-All | Remove all omics | $0.2669_{\pm 0.0075}$ | $0.3577_{\pm 0.0093}$ | $0.7830_{\pm 0.0182}$ | $0.7538_{\pm 0.0174}$ |
| L3-FT(QA)+Omic+KG | Original | $0.4908_{\pm 0.0402}$ | $0.5185_{\pm 0.0240}$ | $0.8794_{\pm 0.0114}$ | $0.8529_{\pm 0.0153}$ |

showing strong transfer to external dataset (Details are provided in Appendix D.6). Figure 4b-c illustrates the explainable subgraph generated for the lung cancer cell line ACH-000860. To further validate the biological relevance of the extracted subgraph, we performed functional enrichment analysis. The results reveal significant enrichment in cancer-associated signaling pathways, including the cancer pathway WP5434 and EGFR-related receptor signaling pathways such as WP138 and WP3680, as cataloged in WikiPathways (Agrawal et al., 2024). Notably, EGFR a well-established therapeutic target in NSCLC (Steuer & Ramalingam, 2015) appears in five enriched terms, together with PTK2 and WNT16 which are known to regulate invasion, epithelial mesenchymal transition, and therapeutic resistance (Tong et al., 2019; Sun et al., 2012). The observed pathway enrichment provides strong biological support for the relevance of the selected targets in lung cancer. Additional disease enrichment using the GAD DISEASE database further supports this conclusion (Sherman et al., 2022), with lung cancer identified as the top associated disease term with a p value of 0.0022, involving GSTM3, APAF1, NOD2, MLLT3, GC and EGFR. Full enrichment details across all cancer types are included in the Appendix E.1. In addition, human and LLM evaluation results in Appendix E.2 show that most generated subgraphs are biologically plausible.

**Ablation Studies**   As shown in Table 4, we evaluated GALAX under systematic perturbations to omic inputs and KG structure. The KG provides both the guidance for graph generation and the retrieved subgraphs that supply biological context to the language model, so removing portions of the KG naturally reduces overall performance. Under KG deletions, GALAX remained stronger than the baseline across all edge-removal levels because node attributes and pretrained GNN embeddings enabled the model to assemble meaningful subnetworks from sparse structure. Node deletion was more destructive since removing 20% of nodes removed about 35% of edges, but GALAX maintained more stable Hit@5 and Hit@10 scores than the baseline across all deletion levels. Removing epigenomic data produced the smallest decline due to methylation saturation, while removing genomic, transcriptomic, or proteomic signals caused larger drops, yet GALAX still outperformed the L3-FT(QA)+Omic+KG baseline even when all omics were removed. Overall, these findings show that GALAX generalizes well under shifts in omic distributions and reduced KG connectivity.

## 6   CONCLUSION

We present **GALAX**, a graph-augmented language model that unifies numerical multi-omic evidence, literature-scale textual information, and stepwise graph construction under reinforcement learning with biologically grounded supervision through a Graph Process Reward Model (GRPM), which scores intermediate subgraphs for biological plausibility and cancer relevance and guides the system to generate patient-specific mechanistic subgraph for target prioritization without explicit labels. To facilitate evaluation, we introduce **Target-QA**, a benchmark combining multi-omic data, CRISPR outcomes, and graph knowledge for target discovery. GALAX consistently outperforms baseline models on this dataset.

ETHICS STATEMENT

This dataset is derived from the Broad Institute's Cancer Dependency Map (DepMap[1]) and will be released strictly for non-commercial, internal research and academic use, consistent with DepMap's Terms of Use. We do not redistribute original DepMap files; instead, we provide derived, non-identifiable annotations and processing scripts/pointers so users can obtain the source data directly from DepMap after accepting its terms. The dataset is not intended for clinical applications and must not be used for any Commercial Use (e.g., direct sale, incorporation into a product, or training/developing/enhancing ML/AI models beyond internal academic research). Users agree to acknowledge DepMap and the Broad Institute using the acknowledgement wording specified by DepMap, and to respect any third-party rights that may attach to the underlying data. Users must preserve confidentiality and refrain from any re-identification attempts. This statement summarizes our compliance posture and does not constitute legal advice; users are responsible for ensuring their own compliance with DepMap's Terms and applicable policies.

REPRODUCIBILITY STATEMENT

We release the source code together with preprocessing pipelines, thereby enabling reproduction of Target-QA datasets and reported experiments. The **Target-QA**[2] and **GALAX**[3] are publicly available at Huggingface and GitHub.

ACKNOWLEDGMENTS

This research was partially supported by NLM 1R01LM013902-01A1.

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

## A  THE USE OF LARGE LANGUAGE MODELS

We used ChatGPT-5 as a writing assistant. All LLM-suggested text was reviewed, fact-checked, and edited by the authors, who take full responsibility for the final content. The LLM is not an author and is not eligible for authorship under the ICLR Code of Ethics.

## B  DATASET

### B.1  BIOMEDICAL TERMINOLGY CORPUS

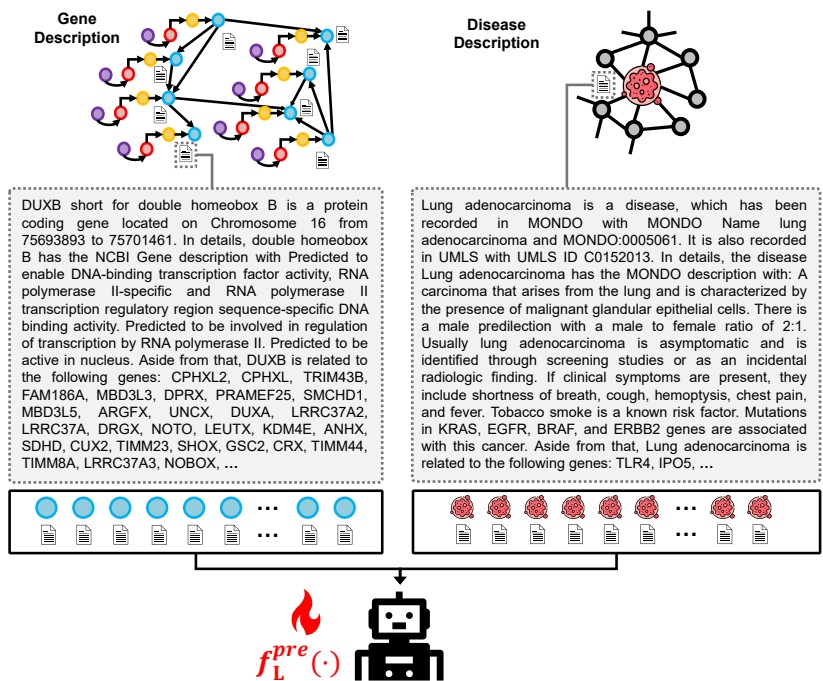

**Figure 5: Biomedical Corpus Composition.** Pretraining LLM $f_L^{\text{pre}}(\cdot)$ on biomedical terminology and structural patterns using protein-protein and disease-protein interactions.

Each entities or nodes in TOSG has the name and description about it with $\mathcal{T} = \{T_{\text{name}}, T_{\text{desc}}\}$, where $|T_{\text{name}}| = |T_{\text{desc}}| = M$. And disease entity has has an associated name and textual description with $\mathcal{S} = \{S_{\text{name}}, S_{\text{desc}}\}$ where $|\mathcal{S}| = M^{(S)}$. In the Figure 5, we pretrained $f_L^{\text{pre}}$ with curated text copora in BioMedGraphica[4], where they provided the data collection and integration source code. Following their processed descriptions, we incorporate $\mathcal{G}^{(\text{PPI})}$ and $\mathcal{G}^{(\text{DTI})}$ to enrich the corpus by appending protein-protein (PPI) and disease-protein (DTI) interaction information as textual descriptions after each protein and disease entity. For entities without known interactions, we assign empty strings during corpus construction, resulting in the intermediate representations $\mathcal{T}$ and $\mathcal{S}$. In practice, we exclude these empty entities to derive the final input sets $\mathcal{T}'$ and $\mathcal{S}'$, where $\mathcal{T}'$ includes 42,224 protein descriptions and $\mathcal{S}'$ includes 22,340 disease descriptions (i.e., $|\mathcal{T}'| = 42,224$ and $|\mathcal{S}'| = M'^{(S)} = 22,340$). This yields a combined corpus of 64,564 text samples for pretraining.

### B.2  DEPMAP DATA PREPROCESSING

As shown in Figure 6a, after multi-omics integration in BioMedGraphica of DepMap cohort (see Table 6) comprises $N^{(0)}$ ($N^{(0)} = 985$) cell-line samples, which are organized into three datasets: a pretraining set, Target-QA, and Drug-QA. Of these 985 samples, 336 samples lack disease/tcga-code annotations or belong to non-cancerous samples, while 649 samples cancerous. Within the annotated cancer set, 363 samples overlap with DepMap CRISPR multi-omic data. Since we would like to utilize as many as samples for training GALAX, we set $N = 363$ as Target-QA, $N^{(0)} = 336$

---

[4] https://huggingface.co/datasets/FuhaiLiAiLab/BioMedGraphica

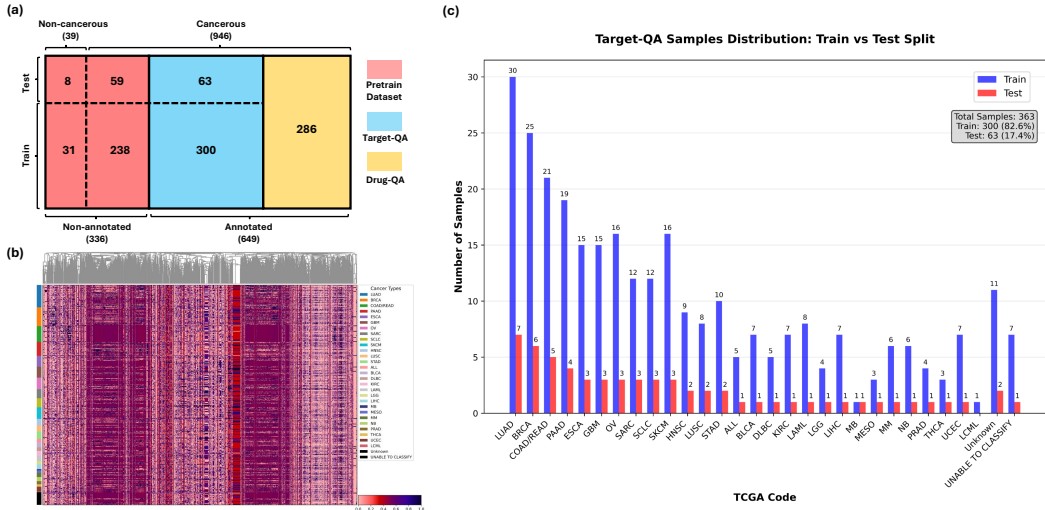

**Figure 6: Dataset composition, feature landscape, and split statistics.** **(a)** Block diagram of the corpus partitioned by phenotype (non-cancerous vs. cancerous) and by task: pretraining pool (pink), Target-QA (blue), and Drug-QA (yellow). Numbers within boxes denote sample counts; the dashed horizontal line marks the train/test division and the dashed vertical line marks the non-cancerous/cancerous and non-annotated/annotated boundaries. **(b)** Heatmap of standardized multi-omic features (top 5,000 most variance features) for Target-QA samples. Rows are samples (annotated by TCGA cancer type); columns are features with top variances. The colored sidebar encodes TCGA cancer types (legend at right). **(c)** Distribution of Target-QA samples by TCGA code for train (blue) and test (red); totals shown in the inset ($N = 363$; train $= 300$ (82.6%), test $= 63$ (17.4%)).

**Table 5:** TCGA cancer type codes and their full names.

| TCGA Code | Full Name | TCGA Code | Full Name |
|---|---|---|---|
| Overall Average | Overall Average | MB | Medulloblastoma |
| LUAD | Lung Adenocarcinoma | ALL | Acute Lymphoblastic Leukemia |
| BRCA | Breast Invasive Carcinoma | LGG | Brain Lower Grade Glioma |
| COAD/READ | Colon/Rectum Adenocarcinoma | NB | Neuroblastoma |
| PAAD | Pancreatic Adenocarcinoma | MESO | Mesothelioma |
| GBM | Glioblastoma Multiforme | LIHC | Liver Hepatocellular Carcinoma |
| SARC | Sarcoma | LAML | Acute Myeloid Leukemia |
| OV | Ovarian Serous Cystadenocarcinoma | DLBC | Lymphoid Neoplasm Diffuse Large B-cell Lymphoma |
| SKCM | Skin Cutaneous Melanoma | MM | Multiple Myeloma |
| ESCA | Esophageal Carcinoma | KIRC | Kidney Renal Clear Cell Carcinoma |
| SCLC | Small Cell Lung Cancer | THCA | Thyroid Carcinoma |
| HNSC | Head and Neck Squamous Cell Carcinoma | BLCA | Bladder Urothelial Carcinoma |
| LUSC | Lung Squamous Cell Carcinoma | UCEC | Uterine Corpus Endometrial Carcinoma |
| STAD | Stomach Adenocarcinoma | PRAD | Prostate Adenocarcinoma |

as pretraining set. To pretrain the graph foundation model, $f_{\mathrm{G}}^{\mathrm{pre}}$, $f_{\mathrm{G}}$, we consider $|\mathcal{O}| = 2$ classes—cancerous (297 samples) and non-cancerous (39 samples)—and perform an 80/20 random split, yielding 269 training samples (238 cancerous, 31 non-cancerous) and 67 test samples (59 cancerous, 8 non-cancerous). The raw omics feature matrices include promoter and gene modalities $m^{(pm)}$ and $m^{(g)}$ (each with 86,238 entities), a transcript modality $m^{(t)}$ (412,039 transcript-level entities), and a protein modality $m^{(p)}$ (121,419 protein-level entities). The promoter modality is represented as a virtual node type in the graph encoder—entity-wise duplicates of genes whose omic values are drawn from DepMap methylation. Together, these modalities yield $M = 834,809$ omics entities (see Table 7). The unified knowledge graph integrates protein–protein interactions ($|\mathcal{E}| = 27{,}087{,}971$) and disease–target associations ($|\mathcal{E}^{(\mathrm{PPI})}| = 17{,}151{,}453$). Every node in the Text-Omic Signaling Graph (TOSG) carries text attributes $\mathcal{T} = \{T_{\mathrm{name}}, T_{\mathrm{desc}}\}$ with $|T_{\mathrm{name}}| = |T_{\mathrm{desc}}| = M$; empty fields are set to the empty string to preserve schema alignment. Hence, we construct the TOSG by linking multi-omics features with biomedical relational knowledge using BioMedGraphica and pretrain the graph encoder $f_{\mathrm{G}}^{\mathrm{pre}}$, $f_{\mathrm{G}}$ (parameters $\theta_{\mathrm{G}}^{\mathrm{pre}}$, $\theta_{\mathrm{G}}$) with a masked-edge modeling objective that encourages recovery of held-out interactions from context, capturing implicit omic relationships and signaling dependencies.

**Table 6:** Descriptions and sources of raw data files used in this study

| File Type | File Name | Download Site |
|---|---|---|
| Promoter feature | CCLE_RRBS_TSS1kb_20181022.txt | `https://depmap.org/portal/data_page/?tab=allData` |
| Gene feature | OmicsCNGene.csv | `https://depmap.org/portal/data_page/?tab=allData` |
| Transcript feature | See footnote[†] | `https://depmap.org/portal/data_page/?tab=allData` |
| Protein feature | protein_quant_current_normalized.csv | `https://depmap.org/portal/data_page/?tab=allData` |
| CRISPR gene effect | CRISPRGeneEffect.csv | `https://depmap.org/portal/data_page/?tab=allData` |
| Cell line annotation | Table_S1_Sample_Information.xlsx | `https://depmap.org/portal/data_page/?tab=allData` |
| Cell line annotation | cellosaurus.obo | `https://ftp.expasy.org/databases/cellosaurus/cellosaurus.obo` |
| Cell line status | cell-lines-in-Non-Cancerous.csv | `https://depmap.org/portal/context/Non-Cancerous` |

[†] Transcript file name: `OmicsExpressionProteinCodingGenesTPMLogp1BatchCorrected.csv`

**Table 7:** Summary of feature dimensions across omics and samples

| Modality | Raw Matrix | Processed Matrix |
|---|---|---|
| Promoter | 21,337 rows, 846 samples | 86,238 entities, 985 samples |
| Gene | 38,590 rows, 1,928 samples | 86,238 entities, 985 samples |
| Transcript | 19,138 rows, 1,672 samples | 412,039 entities, 985 samples |
| Protein | 12,755 rows, 378 samples | 121,419 entities, 985 samples |
| Cell Line Annotation | 1,019 samples | 985 samples |
| Non-cancer Samples | 137 samples | 39 samples |

## B.3 TARGET-QA GENERATION

Based on the raw data provided from the DepMap CRISPR gene effect data with 1178 samples, we get the overlapped 363 ($N = 363$) samples with the pretraining samples. And we do 80/20 train/test split with 80/20 ratiofor 300 training samples and 63 test samples at random seeds. In total, we collected test samples from LUAD (7 samples), BRCA (6 samples), COAD/READ (5 samples), PAAD (4 samples), ESCA (3 samples), GBM (3 samples), OV (3 samples), SARC (3 samples), SCLC (3 samples), SKCM (3 samples), HNSC (2 samples), LUSC (2 samples), STAD (2 samples), etc. (see Figure 6C). Given that methylation values in DepMap have so many are 1 (full methylated over the promoter region around the trasnscription start site), which means that many values are ranked as top $K$ ($K$=10), so we just omit the methylation (epigeonomic) values by only providing geomic, transcriptomic and proteomic values. Afterwards, each QA sample is indexed by a unique key corresponding to the cancer cell line, such as:

- `ACH-000098`: The identifier for a glioblastoma cancer cell line.

The corresponding JSON object contains the following fields:

- **cell_line_name**: Name of the cancer cell line (e.g., `GAMG`).
- **sample_dti_index**: Index for omics numpy data to be fetched.

- **disease**: Name of the associated disease (e.g., `glioblastoma`).
- **disease_bmgc_id**: BioMedGraphica-Conn identifier of the disease (e.g., `BMGC_DS00965`).
- **input**:
  - **top_k_gene**, **top_k_transcript**, **top_k_protein**:
    * `hgnc_symbols`: List of gene/transcript/protein names.
    * `protein_bmgc_ids`: Corresponding BioMedGraphica-Conn identifiers.
    * `protein_llmname_ids`: Other synonymy names or IDs for corresponding genes/transcripts/proteins.
  - **knowledge_graph**:
    * `disease_protein`: Includes `bmgc_ids`, `hgnc_symbols`, and `indices` for disease-associated proteins.
    * `ppi_neighbors`: PPI-linked proteins with similar structure as above.
    * `protein_relationships`: Textual descriptions of biological interactions (e.g., `"BRCA1 → TP53"`).
- **ground_truth_answer**: Contains the validated target(s) used for evaluation:
  - `hgnc_symbols`: HGCN symbol names for CRISPR targets
  - `protein_bmgc_ids`: Correpsonded CRISPR targets names for BioMedGraphica-Conn names
  - `protein_llmname_ids`: Other synonymy names or IDs

This hierarchical structure supports multi-modal reasoning by organizing omic features, biomedical knowledge graphs, and ground-truth target labels, where the cell line name is denoted as $c_n$, the disease name as $s'_n$, the top-ranked genes, transcripts, and proteins as $X_n^{(K)}$, and the associated knowledge graph as $\mathcal{G}_n^{(\mathrm{sub})}$.

## C PRETRAINING OF FOUNDATION MODELS

### C.1 LLM PRETRAINING ON BIOMEDICAL CORPUS

We pretrain a large language model, denoted as $f_{\mathrm{L}}^{\mathrm{pre}}$ with parameters $\theta_{\mathrm{L}}^{\mathrm{pre}}$, using curated text corpora from final input sets $\mathcal{T}'$ and $\mathcal{S}'$. Hence, by pretraining the Llama3-8B-Instruct for 3 epochs using a per-device batch size of 16 and a gradient accumulation step of 8, resulting in an effective batch size of 128. The optimizer was AdamW with a learning rate of 1e-5, no weight decay, and a cosine learning rate schedule with a warm-up ratio of 10%. Gradient clipping was applied with a maximum gradient norm of 1.0. To improve memory efficiency during training, we enabled gradient checkpointing and utilized `bf16` precision while disabling `fp16`. Training proceeded for 336 steps, with the loss decreasing from approximately 2.2 at the start to around 0.7 by the end (see Figure 7).

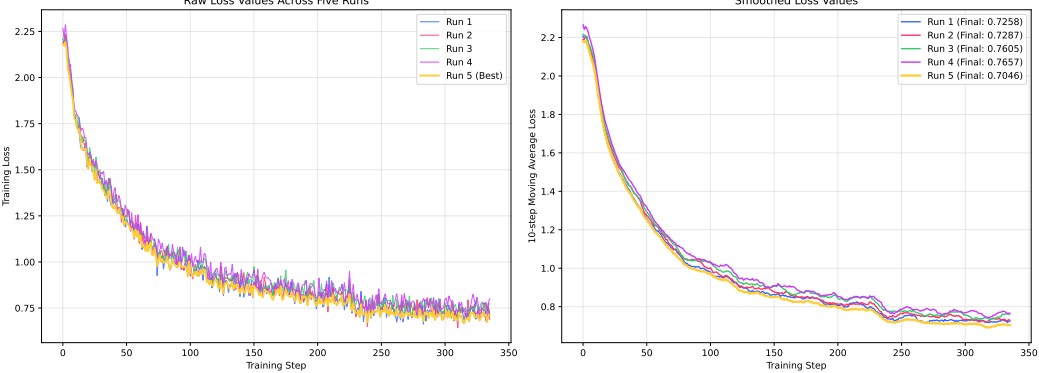

**Figure 7:** Pretraining language model loss on biomedical corpus

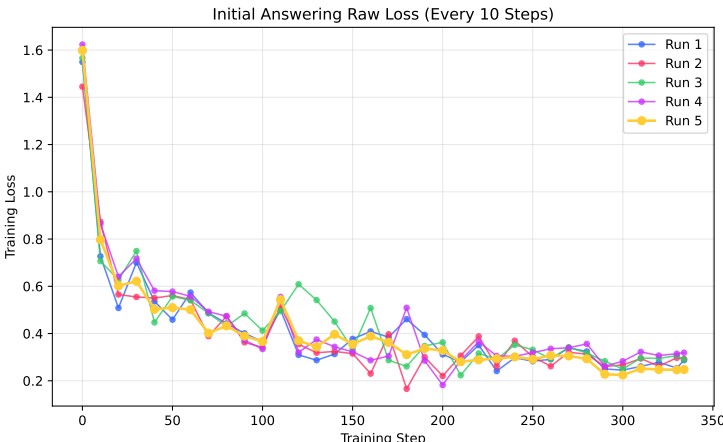

**Figure 8:** Initial langugage answering training loss

## C.2 LLM Pretraining on Target-QA

We will supervise the language model using the refined answer $A_n$ as the ground-truth label from Target-QA with:

$$\mathcal{L}_{\text{init}} = -\sum_{n=1}^{N} \sum_{j=1}^{J} \log \xi_{\theta_{\text{init}}, \theta_{\text{L}}}(a_{n,j} \mid a_{n,<j}, P_n^{(\text{init})}) \qquad (15)$$

, where $P_n^{(\text{init})}$ is the structured prompt as shown in Table 8. This objective enables the model to refine its initial answering based on user-specific and naive graph information. In details, for each sample $c_n$, a question is constructed as $Q_n = \{c_n, s_n', X_n^{(K)}, \mathcal{G}_n^{(\text{sub})}\}$, which integrates the cell line identifier, disease label, multi-omics features, and a knowledge subgraph specific to the sample context. The corresponding answer $A_n$ is a sentence that enumerates the top $\gamma$ CRISPR-prioritized gene targets for sample $n$, denoted as $R_n = \{r_{n,1}, r_{n,2}, \ldots, r_{n,\gamma}\}$. Each data instance is represented as a tuple $D_n = (Q_n, A_n)$, and the full dataset is given by $\mathcal{D} = \{D_1, D_2, \ldots, D_N\}$. The input to the model is a structured prompt $P_n^{(\text{init})}$ derived from $Q_n$, in which the knowledge subgraph $\mathcal{G}_n^{(\text{sub})}$ is translated into a natural language format designed for expert-level graph reasoning. For example, a subgraph $\mathcal{G}_n^{(\text{sub})}$ may contain nodes BRCA1, TP53, EGFR, MAPK1, AKT1, PIK3CA, MTOR, PTEN, and CDK2, with observed interactions: BRCA1 → TP53, TP53 → EGFR, EGFR → MAPK1, EGFR → AKT1, AKT1 → MTOR, PIK3CA → AKT1, PIK3CA → PTEN, PTEN → MTOR, and MAPK1 → CDK2. This subgraph defines the molecular context for reasoning about gene knockout effects in the cell line $c_n$ under disease condition $s_n'$. We conduct five independent finetuning trials using the constructed Target-QA dataset, each initialized with a different random seed. The resulting training loss trajectories are shown in Figure 8. Among these, the best-performing run (run 5) exhibits stable convergence, beginning with an initial loss of approximately 1.3 and reaching a final loss of around 0.3. We select this run as the final model checkpoint and designate it as our initial model $f_{\text{init}}$, which serves as the foundation for downstream reasoning and refined target prioritization.

For pretraining $f_{\text{init}}$ with parameterized $\theta_{\text{init}}$, we pretrain for 5 epochs with per-device batch size 1 and gradient accumulation 2 (effective global batch size = $2 \times$ GPUs), using AdamW (`adamw_torch`) with learning rate $1 \times 10^{-5}$, cosine schedule with $10\%$ warmup, and gradient clipping at $0.5$. We enable gradient checkpointing and `bf16` (with `fp16` disabled), run on $2\times$NVIDIA H100 80GB with DeepSpeed ZeRO-3. Checkpointing and evaluation occur every 67 steps with up to 5 checkpoints retained. Data loading uses 4 workers, pinned memory, no last-batch drop, and no length grouping; we set `seed`$= 42$.

## C.3 Graph Foundation Models

**Pretraining for Capturing the Edge Mechanism**   We pretrain a graph model, denoted as $f_G^{\text{pre}}$ with parameters $\theta_G^{\text{pre}}$, using pretraining samples from $\mathcal{X}^{(0)}$. To effectively model graph-structured biological relationships, we pretrain our model using a masked edge prediction objective com-

**Table 8:** Prompt design for $P_n^{(\text{init})}$ and expected output for initial protein target reasoning.

| Section | Content |
|---|---|
| **Instruction** | Identify the 100 priority genes whose knockout causes the strongest negative effect on the viability or proliferation of cell line $c_n$ in the context of disease $s'_n$, based on multi-omics and knowledge graph signals. |
| **Input** | - Top 10 ranked genes with amplification from copy number data: $g_1'^{(n)}, g_2'^{(n)}, \dots, g_K'^{(n)}$
 - Top 10 ranked transcripts with high expression: $t_1'^{(n)}, t_2'^{(n)}, \dots, t_K'^{(n)}$
 - Top 10 ranked proteins from RPPA: $p_1'^{(n)}, p_2'^{(n)}, \dots, p_K'^{(n)}$
 - Disease-associated proteins from the knowledge graph: $\mathcal{V}_n^{(\text{sub})}$
 - Known protein–protein/disease–protein relationships: $\mathcal{E}_n^{(\text{sub})}$ |
| **Output** | Based on the integrated multi-omics data and knowledge graph, I identified the 100 genes whose knockout is predicted to have the most severe negative impact on the viability or proliferation of the $c_n$ cell line in $s'_n$. The prioritized gene list is as follows:
 1. $r_{n,1}^{(\text{init})}$
 2. $r_{n,2}^{(\text{init})}$
 3. $r_{n,3}^{(\text{init})}$
 $\dots$

These genes represent critical vulnerabilities for the given cell line under the disease context. |

bined with random walk-based graph sampling. The process starts by masking a small fraction ($p = 0.0001$) of edges, allowing the model to infer missing interactions based on surrounding omics and text-derived features. We adopt a GAT-based encoder with two layers, each consisting of 8 hidden channels. Decoder layers use 4 channels. Both encoder and decoder modules apply dropout at a rate of 0.2. The model supports optional batch normalization and uses a `leaky_relu` activation function. An internal encoder stack of up to 4 layers enables deeper relational modeling. All architecture and training options are managed through a reproducible `argparse` interface. Multimodal inputs include one omic feature and a text embedding of dimension 1, initialized using BioBERT v1.1. The model optionally supports training of the text encoder via `train_text`. We use a pretraining batch size of 4 for omics data and 64 for text. The optimizer is AdamW with a learning rate of 0.001, weight decay of $5 \times 10^{-5}$, and gradient norm clipping at 1.0.

The pretrained model is evaluated using average loss, AUC, and average precision (AP) over validation batches. As shown in Figure 9, the model demonstrates progressive improvement across steps, reaching a minimum batch loss of 0.140, peak AUC of 0.644, and peak AP of 0.619. These results confirm that the model successfully learns meaningful edge semantics and multimodal associations during pretraining. All training is conducted on a single NVIDIA H100 GPU (80GB).

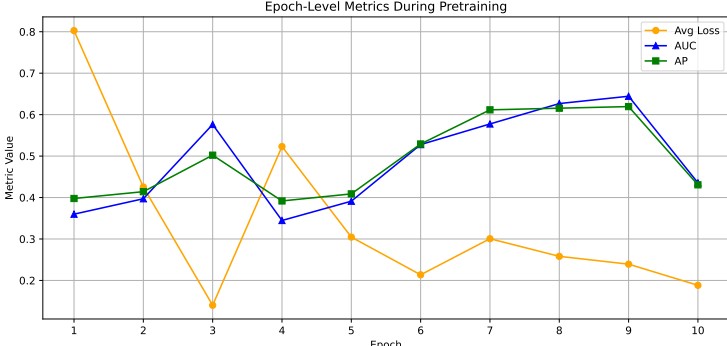

**Figure 9:** Pretraining loss to capture gene regulatory mechanism

**Pretraining for Capturing the Cancerous Status** Due to the severe class imbalance in the training set, we apply random oversampling to the minority class (non-cancerous) to balance the class

**Table 9:** Refined prompt design for $P_n^{(\text{final})}$ and output with graph context

| Section | Content |
| --- | --- |
| **Instruction** | Based on initial LLM reasoning and the subsignaling gene regulatory network identified by the subgraph generator, please identify the 100 priority genes whose knockout causes the strongest negative effect on the viability or proliferation of cell line $c_n$ in the context of disease $s_n'$. |
| **Input** | - Top 10 ranked genes with amplification from copy number data: $g_1'^{(n)}, \ldots, g_K'^{(n)}$
- Top 10 ranked transcripts with high expression: $t_1'^{(n)}, \ldots, t_K'^{(n)}$
- Top 10 ranked proteins from RPPA: $p_1'^{(n)}, \ldots, p_K'^{(n)}$
- Disease-associated proteins from the biomedical knowledge graph: $\mathcal{V}_n^{(\text{sub})}$
- Known protein–protein/disease–protein relationships: $\mathcal{E}_n^{(\text{sub})}$
- Identified Subsignaling Gene Regulatory Network from Graph Generator
    - Involved genes in best connected subgraph: $\mathcal{V}_n^{(\text{conn})}$
    - Inferred signaling cascade (edge text): $\mathcal{E}_n^{(\text{sub})}$ |
| **Refined Reasoning** | Based on the integrated multi-omics data and knowledge graph, I identified the 100 genes whose knockout is predicted to have the most severe negative impact on the viability or proliferation of the $c_n$ cell line in $s_n'$. The prioritized gene list is as follows:
    1. $\hat{r}_{n,1}$
    2. $\hat{r}_{n,2}$
    3. $\hat{r}_{n,3}$
    ...

These genes represent critical vulnerabilities for the given cell line under the disease context. |

distribution during the pretraining of the graph encoder $f_{\text{G}}$. We pretrained a Graph Attention Network (GAT) model $f_{\text{G}}^{\text{pre}}$ with parameters $\theta_{\text{G}}^{\text{pre}}$, and selected the best-performing checkpoint based on test accuracy. The selected model achieved a training loss of 0.036, training accuracy of 99.46%, and training F1 score of 0.996. On the test set, it obtained a loss of 0.370, accuracy of 96.15%, and F1 score of 0.973, demonstrating strong generalization to both classes.

## D  EXPERIMENT DETAILS

### D.1  PROMPT AND CONTEXT DESIGN FOR FINETUNING

Same as aforementioned prompt design shown in Table 8, we construct the final-stage prompt $P_n^{(\text{final})}$, which incorporates both the initial answering and the subgraph-based regulatory context, as detailed in Table 9. We perform five additional finetuning trials using this refined prompt format. The corresponding training are illustrated in Figure 10, where the best run (run 5) demonstrates smooth convergence and is selected as the final model checkpoint $f_{\text{final}}$ for generating the ultimate target predictions.

### D.2  HYPERPARAMETERS

For finetuning, the model is trained for 5 epochs with a per-device batch size of 1 and a gradient accumulation step of 2, resulting in an effective batch size of 2. We use the AdamW optimizer (`adamw_torch`) with a learning rate of $1 \times 10^{-5}$, no weight decay, and cosine learning rate scheduling with a 10% warm-up ratio. Gradient clipping is applied with a maximum gradient norm of 0.5. To support memory efficiency and large model training, we enable gradient checkpointing and use `bf16` precision (with `fp16` explicitly disabled). Training is accelerated using DeepSpeed Stage 3 parallelism, configured via an external JSON file. All experiments are conducted using 2 NVIDIA H100 GPUs, each with 80GB of memory. The DeepSpeed configuration enables ZeRO Stage 3 with both optimizer and parameter offloading to CPU, memory pinning, communication overlap, and gradient contiguity. Key parameters include automatic tuning of reduce bucket sizes and micro-batch size, sub-grouping disabled, and support for 16-bit weight gathering upon model saving. Gradient accumulation steps, clipping, learning rate, weight decay, and total training steps are all set to `"auto"` for adaptive scaling. Warmup is controlled via `WarmupDecayLR`, which

adjusts the schedule based on total steps. Model checkpoints and evaluation are performed every 67 steps, with up to 5 checkpoints retained. Logging is performed at every step. For data loading, we use 4 dataloader workers and retain all batches (i.e., `drop_last=False`). Both the initial answering finetuning and second stage finetuning for $f_{\text{final}}$.

To generate high-quality and biologically reasonable subgraph explanations, we implement a reinforcement-guided generator, denoted as $\pi(\cdot)$, which operates over sample-specific graph states and candidate sets. To enhance robustness under noisy or unstable generation dynamics, we introduce a retry-based mechanism that adaptively tunes key hyperparameters with multiple runs ($\Psi = 6$, and $\psi$ denotes the current number of runs), dynamically adjusting the configuration to encourage exploration and improve convergence. Specifically, the number of training epochs is reduced from 5 to a minimum of 2, promoting faster reinitialization. Simultaneously, the learning rate is increased linearly from an initial value of $1 \times 10^{-3}$ to $0.001 \cdot (1 + \psi)$ to escape local minima. The maximum number of nodes per graph is reduced from 200 to 100 in steps of 25. The maximum number of graph construction steps $i$ per rollout is similarly reduced from 50 to 20 in steps of 5. The retry mechanism enables efficient navigation of the search space while preserving biological plausibility through domain-specific priors embedded in the reward formulation. To further study the effect of sparse versus dense rewards, we vary the reward calculation frequency by introducing a step interval parameter $\mathbf{s}$. Specifically, when $\mathbf{s} = 2$, we compute the reward only every 2 steps during graph generation. Larger values of $\mathbf{s}$ produce sparser, more delayed reward signals, while smaller values (approaching $\mathbf{s} = 1$) provide denser, more immediate feedback. As shown in Table 10, GALAX demonstrates consistent stability across all reward calculation intervals. Notably, performance improves as rewards become denser (smaller $\mathbf{s}$), with optimal results achieved at $\mathbf{s} = 1$, the default GALAX configuration. This trend indicates that the graph generator benefits most from immediate biological feedback provided at each step by the graph process reward model. In contrast, less frequent reward calculations (larger $\mathbf{s}$) introduce noisier, longer-horizon estimates that dilute the signal quality. Importantly, even under highly sparse reward conditions (e.g., $\mathbf{s} = 10$), GALAX consistently outperforms the baseline L3-FT(QA) + Omics + KG, demonstrating robustness to reward sparsity.

**Table 10:** Ablation study on reward density controlled by step interval $\mathbf{s}$

| Reward Interval (s) | Precision ↑ | Recall ↑ | F1 Score ↑ | Jaccard ↑ | Hit@5 ↑ | Hit@10 ↑ |
|---|---|---|---|---|---|---|
| 10 | $0.5312 \pm 0.0044$ | $0.5163 \pm 0.0019$ | $0.5232 \pm 0.0031$ | $0.3582 \pm 0.0035$ | $0.8996 \pm 0.0050$ | $0.8753 \pm 0.0044$ |
| 8 | $0.5347 \pm 0.0062$ | $0.5224 \pm 0.0043$ | $0.5284 \pm 0.0054$ | $0.3619 \pm 0.0055$ | $0.9063 \pm 0.0079$ | $0.8771 \pm 0.0043$ |
| 6 | $0.5398 \pm 0.0051$ | $0.5218 \pm 0.0036$ | $0.5316 \pm 0.0044$ | $0.3657 \pm 0.0044$ | $0.9024 \pm 0.0079$ | $0.8794 \pm 0.0040$ |
| 4 | $0.5389 \pm 0.0054$ | $0.5278 \pm 0.0025$ | $0.5302 \pm 0.0041$ | $0.3673 \pm 0.0046$ | $0.9137 \pm 0.0056$ | $0.8789 \pm 0.0038$ |
| 2 | $0.5463 \pm 0.0057$ | $0.5296 \pm 0.0032$ | $0.5384 \pm 0.0040$ | $0.3718 \pm 0.0043$ | $0.9218 \pm 0.0048$ | $0.8806 \pm 0.0033$ |
| 1 (GALAX) | $\mathbf{0.5472 \pm 0.0053}$ | $\mathbf{0.5332 \pm 0.0031}$ | $\mathbf{0.5399 \pm 0.0041}$ | $\mathbf{0.3726 \pm 0.0037}$ | $\mathbf{0.9249 \pm 0.0048}$ | $\mathbf{0.8815 \pm 0.0033}$ |
| L3-FT(QA) + Omics + KG | $0.5185 \pm 0.0240$ | $0.4908 \pm 0.0402$ | $0.5038 \pm 0.0327$ | $0.3393 \pm 0.0298$ | $0.8794 \pm 0.0114$ | $0.8529 \pm 0.0153$ |

Furthermore, GALAX employs four RL components to collectively prevent reward hacking. First, the **frozen graph oncogenicity classifier (Graph)** derives reward signals from a pre-trained, frozen foundation model rather than a co-trained reward model, thereby eliminating co-evolution exploits. Second, the **schema-based rule term (Rules)** validates each action against biomedical knowledge graph ground truth to ensure that proposed edges conform to established biological constraints. Third, the **rollout-based future reward (Rollout)** evaluates each action based on its long-term consequences, preventing the generator from exploiting local reward irregularities or pursuing myopic gains. Fourth, **stepwise quality gating (Gating)** rejects actions yielding negative cumulative reward, ensuring that only biologically valid steps—those satisfying both schema rules and oncogenicity criteria—are retained. Together, these four components constitute a multi-faceted defense against reward hacking. To validate this design, we conduct an ablation study by systematically removing each component. As shown in Table 11, the robustness of GALAX does not stem from any single component but rather from the interplay of all four mechanisms. The rollout and gating modules help avoid short-sighted decisions and filter out invalid reasoning paths, while the graph supervisor and schema rules provide the underlying constraints that encourage the model to generate biologically plausible subgraphs. Notably, removing either the graph supervisor or the schema rules leads to consistent performance drops, suggesting that the frozen graph oncogenicity classifier offers reliable biological guidance, and that the schema constraints effectively block structurally incorrect actions. In sum, these mechanisms work collectively to push the generator toward producing biologically plausible subgraphs rather than outputs that score well but lack plausibility, making our RL considerably more robust than prior approaches.

**Table 11:** Ablation study of reward components

| Model Variant | Precision ↑ | Recall ↑ | F1 ↑ | Jaccard ↑ | Hit@5 ↑ | Hit@10 ↑ |
|---|---|---|---|---|---|---|
| L3-FT(QA) + Omics | $0.5250_{\pm 0.0282}$ | $0.4959_{\pm 0.0435}$ | $0.5094_{\pm 0.0365}$ | $0.3449_{\pm 0.0338}$ | $0.8889_{\pm 0.0168}$ | $0.8693_{\pm 0.0157}$ |
| L3-FT(QA) + Omics + KG | $0.5185_{\pm 0.0240}$ | $0.4908_{\pm 0.0402}$ | $0.5038_{\pm 0.0327}$ | $0.3393_{\pm 0.0298}$ | $0.8794_{\pm 0.0114}$ | $0.8529_{\pm 0.0153}$ |
| GALAX w/o Graph | $0.5314_{\pm 0.0045}$ | $0.5275_{\pm 0.0077}$ | $0.5336_{\pm 0.0053}$ | $0.3680_{\pm 0.0043}$ | $0.9048_{\pm 0.0084}$ | $0.8741_{\pm 0.0056}$ |
| GALAX w/o Rules | $0.5305_{\pm 0.0074}$ | $0.5154_{\pm 0.0115}$ | $0.5221_{\pm 0.0098}$ | $0.3570_{\pm 0.0078}$ | $0.8974_{\pm 0.0073}$ | $0.8746_{\pm 0.0042}$ |
| GALAX w/o Rollout | $0.5362_{\pm 0.0060}$ | $0.5196_{\pm 0.0052}$ | $0.5271_{\pm 0.0046}$ | $0.3628_{\pm 0.0043}$ | $0.9067_{\pm 0.0062}$ | $0.8769_{\pm 0.0055}$ |
| GALAX w/o Gating | $0.5387_{\pm 0.0050}$ | $0.5243_{\pm 0.0042}$ | $0.5308_{\pm 0.0024}$ | $0.3651_{\pm 0.0032}$ | $0.9118_{\pm 0.0056}$ | $0.8782_{\pm 0.0046}$ |
| **GALAX (Full Model)** | $\mathbf{0.5472}_{\pm \mathbf{0.0053}}$ | $\mathbf{0.5332}_{\pm \mathbf{0.0031}}$ | $\mathbf{0.5399}_{\pm \mathbf{0.0041}}$ | $\mathbf{0.3726}_{\pm \mathbf{0.0037}}$ | $\mathbf{0.9249}_{\pm \mathbf{0.0048}}$ | $\mathbf{0.8815}_{\pm \mathbf{0.0033}}$ |

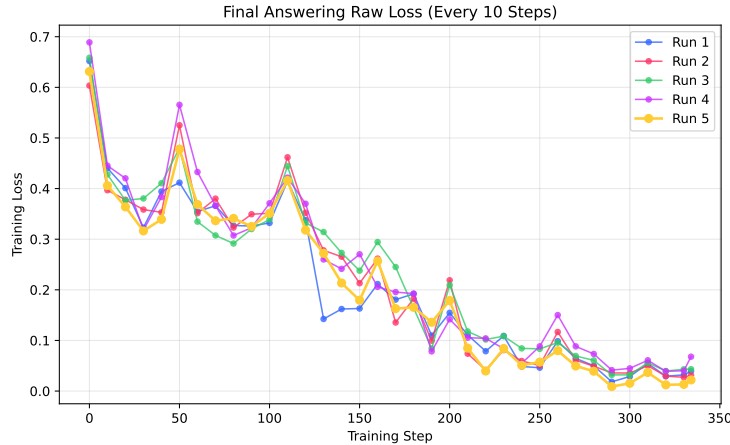

**Figure 10:** Final langugage answering training loss

### D.3 BASELINE MODELS

**Multiomic2Target**   Multiomics2Target[5] is a statistical and knowledge-based framework that integrates transcriptomics, proteomics, and phosphoproteomics to identify cancer-specific therapeutic targets. It leverages curated background knowledge databases and enrichment algorithms to compare tumor profiles against normal tissues, aiming to highlight targets uniquely activated in cancer. Specifically, the method filters out candidates that lack protein-level evidence or are not phosphorylated, and prioritizes those enriched in oncogenic pathways and subtype-specific signaling patterns. The workflow accounts for tumor heterogeneity by enabling both subtype-level and patient-specific analyses, thereby offering a safer and more context-aware target identification strategy. However, as a non-learning-based baseline, Multiomics2Target relies purely on statistical associations, which limits its precision in complex settings. As shown in the evaluation (see Figure 11), this method demonstrates poor performance compared to other approaches, failing to accurately prioritize the most effective targets. The model takes as input the cell line's multi-omics features and outputs a ranked report of candidate targets, but its statistical scoring alone is insufficient to capture deeper, context-specific biological relevance.

**GAT**   As a graph-based baseline, we adapt a Graph Attention Network (GAT) to assess the effect of gene knockouts using multi-omics data and CRISPR-derived gene effect scores. The input graph represents a biological hierarchy, where nodes correspond to genes, transcripts, and proteins, and edges capture central dogma relationships and known molecular interactions. For each gene of interest, we simulate the perturbation process by masking the node corresponding to the gene and recursively masking its downstream elements—namely its associated transcripts and translated proteins—mimicking the transcriptional and translational disruptions observed during a real gene knockout. This masked subgraph is then passed to the GAT model based on the pretrained model $f_G^{pre}$ (parameterized by $\theta_G^{pre}$), which aggregates information from the unmasked neighborhood to predict the impact of the knockout on cellular viability, matching the gene effect values from the CRISPR dataset. This design allows us to evaluate how well the local subgraph structure and omics context can recover the observed CRISPR perturbation effect. Although this method incorporates the biological topology and omics signals through attention-weighted message passing, it lacks explicit global reasoning for disease mechanism and has low performance on CRISPR target prediction.

---

[5]https://multiomics2targets.maayanlab.cloud/

**G-Retriever** G-Retriever is a graph language framework that enables conversational interaction with large, complex graphs enriched with textual node and edge attributes. It facilitates question answering by aligning user queries in natural language with the underlying graph structure, returning both textual responses and highlighted subgraph evidence. This setup is particularly useful for scenarios where graph elements (e.g., proteins, drugs, phenotypes) are associated with rich descriptions or biomedical annotations. In our implementation, we instantiate G-Retriever with two pretrained components: (1) a language model $f_{\text{init}}$ that has been fine-tuned on domain-specific question-answering tasks—specifically, CRISPR-target-related biomedical questions; (2) a graph encoder based on a pretrained Graph Attention Network (GAT), denoted as $f_{\text{G}}^{\text{pre}}$, parameterized by $\theta_{\text{G}}^{\text{pre}}$. The GAT is trained to encode signaling pathway graphs and capture topological and contextual dependencies between biological entities such as genes, proteins, and molecular interactions. This dual-modality initialization, textual reasoning via $f_{\text{init}}$ and structural embedding via $f_{\text{G}}^{\text{pre}}$, allows G-Retriever to support context-aware retrieval over large graphs that exceed the LLM's context window. Through this integration, the framework effectively aligns natural language queries with underlying graph structures, enabling biologically grounded reasoning over complex networks. While G-Retriever offers modest performance improvements over conventional GraphRAG baselines—particularly in terms of contextual relevance and retrieval accuracy—it remains limited in its ability to construct biologically interpretable subgraphs with strong explanatory power. Specifically, G-Retriever lacks a mechanism for enforcing structural coherence or pathway-level constraints, which are essential for generating subgraphs with high fidelity to known signaling or regulatory cascades. Consequently, its generated outputs often fall short of the domain-specific interpretability and performance achieved by our proposed method, which integrates graph-aware reward signals and multi-step reasoning for robust target identification.

**Other Models** RoG, SubgraphRAG, and GNN-RAG are evidence-subgraph-dependent methods that share a common assumption: a small, clean, and reliable subgraph exists from which a retriever module can be trained to provide reasoning paths to the LLM. However, this assumption breaks down in large-scale biomedical knowledge graphs, which are inherently noisier and lack validated reasoning paths. In the Target-QA setting, each question is paired with a one-hop disease–protein subgraph and a one-hop PPI neighborhood extracted from BioMedGraphica. These subgraphs are extremely noisy and large (containing thousands of nodes and edges), and lack ground-truth mechanistic subgraphs, making supervised retrieval difficult. Specifically, RoG, SubgraphRAG, and GNN-RAG require ground-truth reasoning paths for training. Since such annotations are unavailable in our setting, we employ ChatGPT 5.1 to generate candidate reasoning paths relevant to the annotated cell lines, incorporating disease context and multi-omic profiles. However, this GPT-derived supervision is inherently noisy and biologically incomplete, limiting retriever effectiveness. All baselines use the fine-tuned Llama3-8B-Instruct as the backbone LLM.

## D.4 RESULTS

We summarize detailed per-cohort results across various cancer types from TCGA in Figure 11 and Tables 16–43, demonstrating that GALAX achieves the highest performance across nearly all evaluated metrics and cancer cohorts. M2T, serving as a baseline relying solely on multi-omics data without any structured graph or language modeling, performs poorly, underscoring the necessity of incorporating richer contextual information. Adding graph structure via a pretrained Graph Attention Network (GAT) provides some improvements, albeit limited, indicating that static graph representations alone are insufficient for capturing the complex biological interactions inherent in the data. To further elucidate the value of incorporating language models, we evaluate multiple LLaMA3 (L3) variants. Models that lack Target-specific finetuning, such as L3 combined only with omics data (L3) or with additional knowledge graphs (L3 + KG), deliver suboptimal performance, even with external knowledge enrichment. Biomedical-domain-specific finetuning (L3-FT(Med)) significantly enhances performance in identifying relevant targets, and integrating knowledge graphs yields modest incremental improvements. Nonetheless, even the variant supervised by Target-QA (L3-FT(QA) + KG) demonstrates limited additional benefit from knowledge graph integration, primarily due to its inability to dynamically generate structured subgraphs tailored to the reasoning task. The G-Retriever model, augmenting a pretrained graph attention network with retrieval mechanisms, achieves stronger performance compared to earlier baselines but remains limited by its absence of step-wise supervision and interpretability in biological reasoning. In contrast, GALAX uniquely integrates a Target-QA finetuned large language model, a pretrained Graph Attention Network, and a reinforcement learning-driven subgraph generator guided by a graph process reward

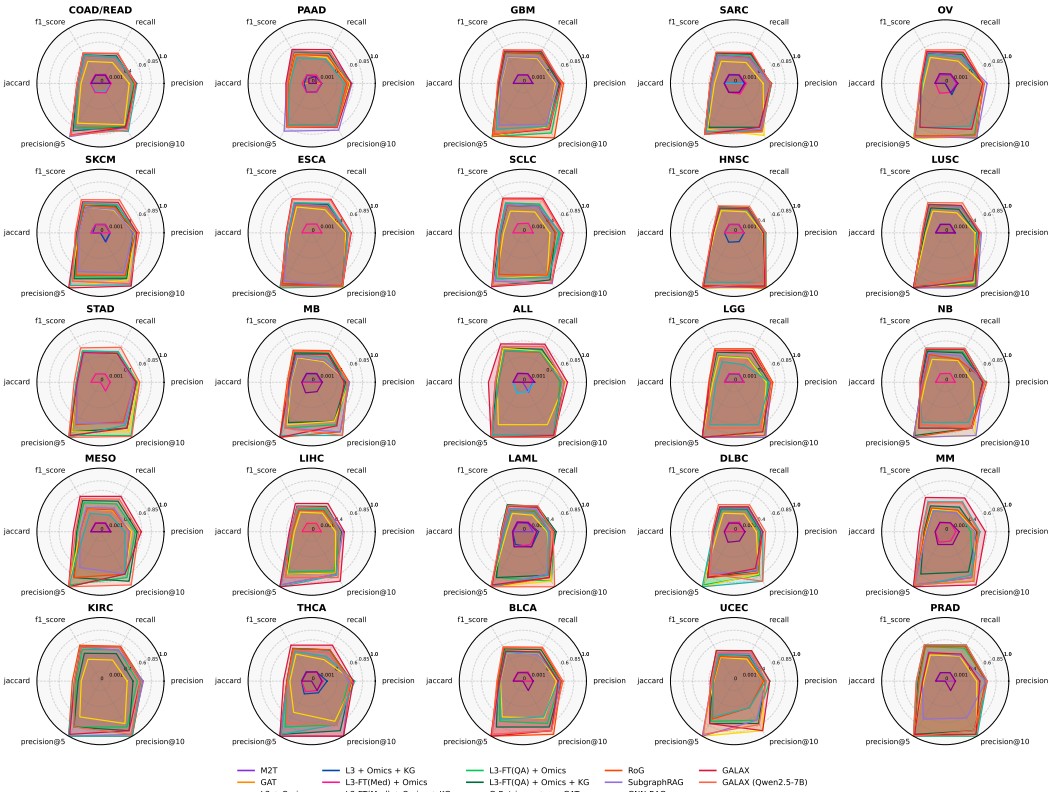

**Figure 11:** Model performance plots over main Target-QA datasets

model. This sophisticated architecture facilitates interpretable and biologically informed reasoning by dynamically generating structured subgraphs relevant to the biological context. Consequently, GALAX significantly surpasses all baseline models, achieving state-of-the-art performance across both overall average metrics and individual cancer cohort-specific evaluations.

## D.5 GENERALIZATION TO UNSEEN CANCER TYPES

To assess how well GALAX generalizes across cancer types, we constructed three holdout sets in which all cell lines under same TCGA codes were removed during training and used only for evaluation (see Table 12). This design forces the model to generate CRISPR target predictions for cancer types not observed during optimization, providing a direct measure of cross-cancer transfer and robustness to unseen molecular contexts.

As expected, performance decreases when entire cancer types are held out, since no fine-tuning data from those cancers is available. Nevertheless, the results in Table 13 show that GALAX maintains stable performance across the three held-out cancer groups. Precision, recall, F1, and Jaccard values remain consistent, with an average F1 of 0.4862 and Hit@10 above 0.85. While these scores are lower than the full-data GALAX model, the gap is moderate, indicating that the model retains the ability to infer relevant targets even when the TCGA code of the input cell line has never appeared during training, as long as it has been well pretrained with sufficient samples. This demonstrates that GALAX does not overfit to specific cancer labels and can transfer its reasoning across diverse cell-line backgrounds, addressing concerns regarding generalization to new cancer types.

**Table 12:** Held-out cancer type fold assignments

| Holdout Set | Train | Test | Train/Test TCGA Codes | Hold-out Cancer TCGA Codes |
|---|---|---|---|---|
| Holdout Set 1 | 238 (65.6%) | 125 (34.4%) | 20 / 10 | COAD/READ, DLBC, KIRC, LCML, LUSC PAAD, PRAD, SARC, SKCM, STAD |
| Holdout Set 2 | 289 (79.6%) | 74 (20.4%) | 21 / 9 | BLCA, GBM, LAML, LGG, LIHC MESO, MM, NB, UCEC |
| Holdout Set 3 | 220 (60.6%) | 143 (39.4%) | 21 / 9 | ALL, BRCA, ESCA, HNSC, LUAD MB, OV, SCLC, THCA |

**Table 13:** GALAX performance on unseen cancer types under held-out evaluation

| Holdout Set | Precision ↑ | Recall ↑ | F1 ↑ | Jaccard ↑ | Hit@5 ↑ | Hit@10 ↑ |
|---|---|---|---|---|---|---|
| Holdout Set 1 | 0.4931 | 0.4501 | 0.4699 | 0.3100 | 0.9024 | 0.8688 |
| Holdout Set 2 | 0.5124 | 0.5151 | 0.5123 | 0.3465 | 0.8703 | 0.8581 |
| Holdout Set 3 | 0.4832 | 0.4715 | 0.4765 | 0.3161 | 0.8993 | 0.8448 |
| **Average** | 0.4962 | 0.4789 | 0.4862 | 0.3242 | 0.8907 | 0.8572 |
| **GALAX** | $0.5472_{\pm 0.0053}$ | $0.5332_{\pm 0.0031}$ | $0.5399_{\pm 0.0041}$ | $0.3726_{\pm 0.0037}$ | $0.9249_{\pm 0.0048}$ | $0.8815_{\pm 0.0033}$ |

## D.6 EXTERNAL DATASETS

To examine generalization beyond Target-QA, we performed an external evaluation on the pediatric cancer dataset from PedDep[6], which contains CRISPR-based gene effect profiles and cell-line annotations for 31 pediatric cancer samples. The data were processed following the same steps as Target-QA. Due to the limited sample size, we did not train or finetune on this dataset and instead evaluated all models in a zero-shot setting. Given that all samples are cancerous, the M2T baseline cannot be applied. As shown in Table 14, baseline graph models and language-model variants still have near-zero performance. Graph-augmented methods such as RoG, SubgraphRAG, and GNN-RAG yield moderate gains, but GALAX attains the highest precision, recall, F1, and Jaccard scores, along with notable improvements in Hit@5 and Hit@10 finetuned on Target-QA with multi-omics and disease-related proteins information. These results demonstrate that GALAX can extend its target-prediction ability to external cancer datasets without retraining, providing strong evidence of its robustness and transferability beyond the Target-QA benchmark.

**Table 14:** Model performance on PedDep cancer dataset

| Model | Precision ↑ | Recall ↑ | F1 ↑ | Jaccard ↑ | Hit@5 ↑ | Hit@10 ↑ |
|---|---|---|---|---|---|---|
| GAT | $0.0005_{\pm 0.0008}$ | $0.0005_{\pm 0.0008}$ | $0.0005_{\pm 0.0008}$ | $0.0002_{\pm 0.0004}$ | $0.0000_{\pm 0.0000}$ | $0.0000_{\pm 0.0000}$ |
| L3-FT(Med) + Omics | $0.0144_{\pm 0.0081}$ | $0.0114_{\pm 0.0049}$ | $0.0099_{\pm 0.0028}$ | $0.0050_{\pm 0.0014}$ | $0.0210_{\pm 0.0421}$ | $0.0109_{\pm 0.0218}$ |
| L3-FT(Med) + Omics + KG | $0.0167_{\pm 0.0161}$ | $0.0060_{\pm 0.0047}$ | $0.0068_{\pm 0.0049}$ | $0.0035_{\pm 0.0025}$ | $0.0072_{\pm 0.0143}$ | $0.0073_{\pm 0.0145}$ |
| L3 + Omics | $0.0054_{\pm 0.0108}$ | $0.0011_{\pm 0.0021}$ | $0.0018_{\pm 0.0036}$ | $0.0009_{\pm 0.0018}$ | $0.0000_{\pm 0.0000}$ | $0.0000_{\pm 0.0000}$ |
| L3 + Omics + KG | $0.0192_{\pm 0.0188}$ | $0.0032_{\pm 0.0022}$ | $0.0050_{\pm 0.0035}$ | $0.0026_{\pm 0.0018}$ | $0.0142_{\pm 0.0165}$ | $0.0214_{\pm 0.0247}$ |
| L3-FT(QA) + Omics | $0.2608_{\pm 0.0268}$ | $0.2605_{\pm 0.0252}$ | $0.2606_{\pm 0.0260}$ | $0.1517_{\pm 0.0152}$ | $0.5810_{\pm 0.0330}$ | $0.4524_{\pm 0.0502}$ |
| L3-FT(QA) + Omics + KG | $0.2619_{\pm 0.0075}$ | $0.2552_{\pm 0.0095}$ | $0.2584_{\pm 0.0086}$ | $0.1489_{\pm 0.0055}$ | $0.5048_{\pm 0.0165}$ | $0.4952_{\pm 0.0218}$ |
| G-Retriever + pre-GAT | $0.2624_{\pm 0.0212}$ | $0.2610_{\pm 0.0206}$ | $0.2617_{\pm 0.0209}$ | $0.1522_{\pm 0.0121}$ | $0.5143_{\pm 0.0495}$ | $0.4524_{\pm 0.0825}$ |
| RoG | $0.2730_{\pm 0.0050}$ | $0.2667_{\pm 0.0092}$ | $0.2697_{\pm 0.0071}$ | $0.1566_{\pm 0.0047}$ | $0.5143_{\pm 0.0286}$ | $0.4714_{\pm 0.0623}$ |
| SubgraphRAG | $0.2736_{\pm 0.0091}$ | $0.2690_{\pm 0.0095}$ | $0.2712_{\pm 0.0092}$ | $0.1579_{\pm 0.0062}$ | $0.5619_{\pm 0.0165}$ | $0.5286_{\pm 0.0571}$ |
| GNN-RAG | $0.2760_{\pm 0.0092}$ | $0.2700_{\pm 0.0094}$ | $0.2728_{\pm 0.0093}$ | $0.1589_{\pm 0.0064}$ | $0.5714_{\pm 0.0495}$ | $0.5333_{\pm 0.0360}$ |
| **GALAX** | $0.2914_{\pm 0.0115}$ | $0.2889_{\pm 0.0131}$ | $0.2901_{\pm 0.0123}$ | $0.1703_{\pm 0.0086}$ | $0.6357_{\pm 0.0589}$ | $0.5179_{\pm 0.0513}$ |
| **GALAX (Qwen2.5-7B)** | $\mathbf{0.2921_{\pm 0.0055}}$ | $\mathbf{0.2895_{\pm 0.0050}}$ | $\mathbf{0.2908_{\pm 0.0053}}$ | $\mathbf{0.1708_{\pm 0.0038}}$ | $\mathbf{0.6667_{\pm 0.0595}}$ | $\mathbf{0.5381_{\pm 0.0705}}$ |

## E REINFORCEMENT LEARNING GENERATED SUBGRAPH DETAILS

Figure 12 provides a high-level overview of the explainable subgraphs of part of cell lines, highlighting key proteins and their predicted functional interactions that define the unique molecular signatures of each cell line. These interpretable network maps identify the most salient features prioritized by our model. To move beyond structural insights and assess their biological significance, we conduct enrichment analyses to evaluate whether the identified molecules are significantly over-represented in curated knowledge bases such as KEGG and WikiPathway. This analysis enables us to contextualize the salient features within relevant targets and signaling pathways. The resulting cell-line-specific interpretations bridge model-driven discovery with established biological knowledge and are detailed in the following sections.

### E.1 ENRICHMENT ANALYSIS

Enrichment analysis of **sample ACH-000054 (HT-1080)**, representing fibrosarcoma (metastasis; SARC), revealed a significant association with **Sarcoma** (P-value = $1.625 \times 10^{-4}$, Adj.P = $4.08 \times 10^{-2}$) driven by **COL1A1**, **WWTR1**, **APAF1**, **PLK1**, **EPHB4**, and **SH2B1**. Pathway-level signals underscored apoptotic and stress-response programs, with strong enrichment of **Apoptosis Modulation by HSP70** (WP384; P-value = $1.864 \times 10^{-6}$, Adj.P = $1.45 \times 10^{-4}$) and **Apoptosis** (WP254; P-value = $1.73 \times 10^{-4}$, Adj.P = $5.26 \times 10^{-3}$), each supported by

---

[6]https://peddep.org/

**Figure 12:** Part of explainable disease mechanism by core signaling subgraphs

**MAPK10**, **APAF1**, and **TNFRSF1A**. Additional inflammatory and mitogenic axes were indicated by **TNF alpha Signaling Pathway** (WP231; P-value = $2.27 \times 10^{-4}$, Adj.P = $5.26 \times 10^{-3}$) involving **APAF1**, **PLK1**, and **TNFRSF1A**, and by the **MAPK Signaling Pathway** (WP382; P-value = $2.70 \times 10^{-4}$, Adj.P = $5.26 \times 10^{-3}$) featuring **MAPK10**, **DUSP1**, **RASGRF2**, and **TNFRSF1A**. Finally, **Nanoparticle-mediated activation of receptor signaling** (WP2643; P-value = $6.02 \times 10^{-4}$, Adj.P = $8.99 \times 10^{-3}$) highlighted extracellular matrix and receptor-proximal cues via **COL1A1** and **MAPK10**. Collectively, recurrent involvement of apoptosis regulators (**APAF1**, **TNFRSF1A**), MAPK components (**MAPK10**, **DUSP1**, **RASGRF2**), and proliferative drivers (**PLK1**) points to coordinated apoptotic, inflammatory, and receptor–MAPK signaling programs characteristic of fibrosarcoma biology.

For **sample ACH-000001 (NIH:OVCAR-3)**, an ovarian adenocarcinoma line (metastasis; OV), analysis indicated only a weak disease-level association with **ovarian cancer** (P-value = $1.41 \times 10^{-1}$, Adj.P = $1.65 \times 10^{-1}$), primarily linked to **ERBB2**. While this signal was not significant after multiple-testing correction, pathway-level evaluation revealed several strongly dysregulated processes. The **Prolactin Signaling Pathway** (WP2037; P-value = $2.90 \times 10^{-4}$, Adj.P = $1.49 \times 10^{-2}$) was prominently enriched through **RPS6KB1**, **NOS2**, and **ERBB2**, pointing to growth factor–driven oncogenic signaling. Similarly, enrichment of the **Leptin Signaling Pathway** (WP2034; P-value = $2.90 \times 10^{-4}$, Adj.P = $1.49 \times 10^{-2}$) and the **ErbB Signaling Pathway** (WP673; P-value = $4.92 \times 10^{-4}$, Adj.P = $1.69 \times 10^{-2}$) implicated **RPS6KB1**, **ERBB2**, and **PLCG2**, reflecting interconnected receptor tyrosine kinase and metabolic networks. Additional pathways included **IGF1–Akt signaling** (WP3850; P-value = $1.26 \times 10^{-3}$, Adj.P = $3.09 \times 10^{-2}$) driven by **RPS6KB1** and **TNFSF9**, and **BDNF–TrkB Signaling** (WP3676; P-value = $1.52 \times 10^{-3}$, Adj.P = $3.09 \times 10^{-2}$) via **ARC** and **RPS6KB1**. These results underscore the central role of **ERBB2**-mediated receptor activity and downstream **RPS6KB1**-linked pathways, consistent with known mechanisms of ovarian carcinoma progression.

In **sample ACH-000219 (A-375)**, representing amelanotic melanoma (primary; SKCM), enrichment testing identified a highly significant signal for **melanoma** (P-value = $1.08 \times 10^{-6}$, Adj.P = $1.59 \times 10^{-4}$) supported by a broad gene panel including **CCL25**, **STAT5B**, **HSP90AA1**, **SERPINB1**, **VWF**, **TYR**, **THBS1**, **RELA**, **TJP1**, **GHR**, **CDH5**, **IRF1**, **TNFSF10**, **TRIM24**, **NCAM1**, **ATG7**, **TNFRSF4**, and **IL9R**. Immune and inflammatory cascades dominated the pathway enrichments, with the **NOD Pathway** (WP1433; P-value = $1.77 \times 10^{-6}$, Adj.P = $2.28 \times 10^{-4}$) enriched via **HSP90AA1**, **NLRP12**, **MEFV**, and **RELA**, pointing to innate immune activation. Cytokine-related processes were also evident, including the **IL-4 Signaling Pathway** (WP395; P-value = $5.42 \times 10^{-6}$, Adj.P = $2.65 \times 10^{-4}$) and the **IL-9 Signaling Pathway** (WP22; P-value = $6.16 \times 10^{-6}$, Adj.P = $2.65 \times 10^{-4}$), with contributions from **STAT5B**, **PIK3R1**, and **IL9R**, consistent with Th2/Th9 regulation. The **Oncostatin M Signaling Pathway** (WP2374; P-value = $1.14 \times 10^{-5}$, Adj.P = $3.60 \times 10^{-4}$) further implicated **STAT5B**, **TYK2**, **PIK3R1**, and **RELA**, linking cytokine–STAT and NF-$\kappa$B signaling to melanoma progression. Overall, repeated enrichment of **STAT5B**, **RELA**, and **PIK3R1** across pathways highlights coordinated dysregulation of inflammatory and cytokine signaling networks in amelanotic melanoma.

Enrichment analysis of **sample ACH-000070 (697)**, representing B-cell acute lymphoblastic leukemia (ALL) with the **t(1;19)(q23;p13.3) E2A–PBX1 (TCF3–PBX1)** translocation, revealed a strong disease-level association with **acute lymphocytic leukemia** (P-value = $7.37 \times 10^{-5}$, Adj.P = $8.10 \times 10^{-3}$) supported by genes including **IGF2**, **WBP1L**, **ELSPBP1**, **HOXD4**, **TCF3**, **THY1**, **SIRT1**, **GATA1**, and **PBX1**. Pathway enrichment further emphasized leukemia-relevant transcriptional programs: **Sudden Infant Death Syndrome (SIDS) Susceptibility Pathways** (WP706; P-value = $3.02 \times 10^{-4}$, Adj.P = $1.94 \times 10^{-2}$) through **TCF3**, **POU2F2**, **ESR2**, and **PBX1**; and the **Wnt/$\beta$-catenin Signaling Pathway in Leukemia** (WP3658; P-value = $1.29 \times 10^{-3}$, Adj.P = $4.13 \times 10^{-2}$) involving **AXIN2** and **TCF3**. Additional signals included the **Breast Cancer Pathway** (WP4262; P-value = $3.85 \times 10^{-3}$, Adj.P = $6.06 \times 10^{-2}$) with **E2F3**, **AXIN2**, and **ESR2**, and the broader **Wnt Signaling Pathway** (WP363; P-value = $5.10 \times 10^{-3}$, Adj.P = $6.06 \times 10^{-2}$) featuring **AXIN2** and **TCF3**. Collectively, these findings underscore recurrent involvement of the **TCF3–PBX1** fusion together with Wnt/$\beta$-catenin signaling and leukemogenic transcription factors, consistent with the molecular etiology of this ALL subtype.

Enrichment analysis of **sample ACH-000092 (NCI-H2452)**, representing pleural mesothelioma (metastasis; MESO), showed a modest disease-level association with **Lung Neoplasms** (P-value = $4.965 \times 10^{-3}$, Adj.P = $1.89 \times 10^{-1}$) supported by **MAP2K1**, **SLC26A2**, **AIFM1**, **SP1**, **ATG4B**, **SAT1**, **GLUL**, and **PPP1R9B**. Pathway analysis highlighted metabolic and growth-factor–linked programs: **Amino Acid metabolism** (WP3925; P-value = $1.20 \times 10^{-3}$, Adj.P = $5.63 \times 10^{-2}$) involving **BHMT**, **PDHA1**, and **GLUL**; the **Estrogen signaling pathway** (WP712; P-value = $1.27 \times 10^{-3}$, Adj.P = $5.63 \times 10^{-2}$) marked by **MAP2K1** and **SP1**; and **TGF-$\beta$ Signaling** (WP366; P-value = $3.47 \times 10^{-3}$, Adj.P = $1.15 \times 10^{-1}$) featuring **MAP2K1**, **SUMO1**, and **SP1**. Together, these results suggest coordinated amino acid metabolic rewiring and transcriptional control via MAPK–**SP1** and TGF-$\beta$ axes in the mesothelioma context.

Enrichment analysis of **sample ACH-000817 (RPMI 8226)**, representing plasma cell myeloma (primary; MM), revealed a highly significant association with **Multiple Myeloma** (P-value = $3.891 \times 10^{-6}$, Adj.P = $6.64 \times 10^{-4}$) supported by **ITGB1**, **KMT2D**, **CBX7**, **TNFRSF13B**, **IDH1**, **LAPTM5**, **HLA-C**, **TNFRSF10A**, **PIK3R2**, **KIR3DL1**, **BHLHA15**, **PIK3R1**, **MEFV**, **AURKA**, **NRAS**, **NUAK1**, **NPC1**, **BTK**, **KRAS**, **SGK1**, **HRAS**, **MAPK3**, and **FBXO9**. Pathway analysis further highlighted the **ErbB Signaling Pathway** (WP673; P-value = $8.47 \times 10^{-8}$, Adj.P = $1.03 \times 10^{-5}$) involving **MAPK10**, **CAMK2D**, **NRAS**, **PIK3R2**, **KRAS**, **PIK3R1**, **HRAS**, and **MAPK3**, indicating convergence of RAS–MAPK and PI3K effector cascades downstream of ErbB family receptors—features consistent with myeloma signaling dependencies and potential therapeutic vulnerabilities.

Enrichment analysis of **sample ACH-000649 (786-O)**, representing renal cell carcinoma (primary; KIRC), revealed a highly significant association with **Conventional (Clear Cell) Renal Cell Carcinoma** (P-value = $1.273 \times 10^{-9}$, Adj.P = $9.68 \times 10^{-7}$) supported by **TGFB1**, **IL4R**, **VCAM1**, **APAF1**, **OGG1**, **MSGN1**, **MITF**, **UNC5C**, **SMARCA2**, **TNF**, **POMC**, **FPGT**, **ORC2**, **ERBB2**, **ALDH1A1**, **PGK1**, **BIRC5**, **ZNF536**, **VHL**, and **CRYAB**. This gene set spans

hallmark features of clear cell RCC biology, including hypoxia/VHL-axis signaling (**VHL**, **PGK1**), apoptosis regulation (**APAF1**, **BIRC5**), growth factor and RTK pathways (**ERBB2**, **TGFB1**), immune–inflammatory components (**TNF**, **IL4R**, **VCAM1**), metabolic and oxidative stress responses (**ALDH1A1**, **CRYAB**), and genome maintenance/chromatin remodeling (**OGG1**, **ORC2**, **SMARCA2**), collectively underscoring a prototypical clear cell transcriptomic signature.

Enrichment analysis of **sample ACH-000018 (T24)**, representing bladder carcinoma (primary; BLCA), showed a borderline disease-level signal for **Cancer** (P-value $= 5.755 \times 10^{-2}$, Adj.P $= 1.02 \times 10^{-1}$) driven by **ZBTB16** and **RAF1**. Pathway-level analysis revealed significant enrichment of **Focal Adhesion** (WP306; P-value $= 1.17 \times 10^{-4}$, Adj.P $= 1.03 \times 10^{-2}$) involving **MAPK10**, **RAP1A**, **COL4A6**, and **RAF1**; the **ErbB Signaling Pathway** (WP673; P-value $= 2.20 \times 10^{-4}$, Adj.P $= 1.03 \times 10^{-2}$) featuring **MAPK10**, **CAMK2D**, and **RAF1**; and **Integrin-mediated Cell Adhesion** (WP185; P-value $= 2.99 \times 10^{-4}$, Adj.P $= 1.03 \times 10^{-2}$) with **MAPK10**, **RAP1A**, and **RAF1**. Collectively, recurrent involvement of **RAF1** together with **MAPK10** across adhesion and ErbB-axis pathways points to convergent RTK–MAPK and integrin signaling programs in this BLCA context.

Enrichment analysis of **sample ACH-000864 (COLO 684)**, representing endometrial adenocarcinoma (primary; UCEC), revealed a highly significant disease-level association with **Carcinoma, Small Cell** (P-value $= 1.064 \times 10^{-6}$, Adj.P $= 1.29 \times 10^{-4}$) supported by **SMARCB1**, **AKT3**, **MTOR**, and **SMARCA4**. Pathway analysis highlighted potent receptor tyrosine kinase and stemness programs: the **ErbB Signaling Pathway** (WP673; P-value $= 3.38 \times 10^{-9}$, Adj.P $= 4.90 \times 10^{-7}$) involving **JUN**, **CAMK2A**, **AKT3**, **PIK3R2**, and **MTOR**; the **EGF/EGFR Signaling Pathway** (WP437; P-value $= 6.19 \times 10^{-8}$, Adj.P $= 4.48 \times 10^{-6}$) through **JUN**, **GJA1**, **CAMK2A**, **PIK3R2**, and **MTOR**; and **ESC Pluripotency Pathways** (WP3931; P-value $= 1.03 \times 10^{-6}$, Adj.P $= 4.97 \times 10^{-5}$) marked by **JUN**, **AKT3**, **PIK3R2**, and **MTOR**. Collectively, recurrent involvement of **AKT3** and **MTOR**—together with **PIK3R2**, **JUN**, and **CAMK2A**—points to convergent EGFR/ErbB–PI3K–AKT–mTOR signaling and pluripotency-associated programs in this UCEC context.

### E.2 HUMAN EVALUDATION AND LLM AS JUDGE

To further assess the biological relevance of the generated subgraphs, we conducted a comprehensive evaluation involving both human domain experts and Large Language Models (LLMs). The human evaluation panel consisted of three bioinformaticians, denoted as $\mathbf{h}_1, \mathbf{h}_2$, and $\mathbf{h}_3$. In parallel, we used two advanced LLMs as automated evaluators ($\mathbf{m}_1, \mathbf{m}_2$): ChatGPT-5.1 and Gemini-3.0 Pro. Both the human experts and the LLMs were provided with the generated subgraphs, corresponding gene and pathway enrichment analysis results to facilitate the assessment of biological plausibility.

**Table 15:** Human Evaluation Scores

| Sample ID | TCGA Code | $\mathbf{h}_1$ | $\mathbf{h}_2$ | $\mathbf{h}_3$ | $\mathbf{m}_1$ | $\mathbf{m}_2$ | Mean $\pm$ Std |
|---|---|---|---|---|---|---|---|
| ACH-000860 | LUAD | 4 | 5 | 5 | 5 | 5 | $4.80 \pm 0.45$ |
| ACH-000054 | SRCA | 3 | 5 | 3 | 4 | 4 | $3.80 \pm 0.84$ |
| ACH-000001 | OV | 3 | 3 | 2 | 4 | 2 | $2.80 \pm 0.84$ |
| ACH-000219 | SKCM | 4 | 5 | 2 | 4 | 3 | $3.60 \pm 1.14$ |
| ACH-000070 | ALL | 5 | 5 | 5 | 4 | 5 | $4.80 \pm 0.45$ |
| ACH-000092 | MESO | 3 | 3 | 2 | 3 | 3 | $2.80 \pm 0.45$ |
| ACH-000817 | MM | 5 | 5 | 5 | 3 | 3 | $4.20 \pm 1.10$ |
| ACH-000649 | KIRC | 4 | 4 | 5 | 5 | 5 | $4.60 \pm 0.55$ |
| ACH-000018 | BLCA | 3 | 3 | 3 | 3 | 3 | $3.00 \pm 0.00$ |
| ACH-000864 | UCEC | 3 | 4 | 2 | 4 | 3 | $3.20 \pm 0.84$ |
| **Overall** | - | 3.7 | 4.2 | 3.4 | 3.9 | 3.6 | $\mathbf{3.76} \pm 0.80$ |

Each expert independently reviewed ten subgraph examples shown in Table 15 and assigned a score from 1 to 5 based on the degree of correspondence between the subgraph and the known biology of the associated TCGA cancer type. The scoring rubric was defined as follows:

- 5: Highly related — strong and clear match to hallmark pathways and well-established features of the cancer type.

- 4: Related — clear relationship to the cancer type but less comprehensive or slightly mixed.

- 3: Moderately related — some relevant elements present, but mixed with non-specific findings.
- 2: Not related — only weak or indirect connection to the cancer type.
- 1: Not related at all — no meaningful alignment with the biology of the cancer type.

As shown in Table 15, several samples were evaluated as related or highly related to their TCGA cancer types, indicating that the subgraphs generated by the model capture meaningful molecular features. The LUAD sample ACH-000860 received one of the highest average scores, as its subgraph highlighted pathways such as regulation of phosphatidylinositol 3-kinase signaling and positive regulation of kinase activity, involving genes like EGFR, PTK2, and EPHB4 that are central to lung adenocarcinoma biology. The ALL sample ACH-000070 was also rated highly due to the presence of TCF3 and PBX1, whose fusion is a well-established driver event in Acute Lymphoblastic Leukemia.

**Table 16:** Model Overall performance on Target-QA

| Model | Precision ↑ | Recall ↑ | F1 ↑ | Jaccard ↑ | Hit@5 ↑ | Hit@10 ↑ |
|---|---|---|---|---|---|---|
| M2T | 0.0016 | 0.0011 | 0.0013 | 0.0006 | 0.0000 | 0.0029 |
| GAT | $0.0006_{\pm 0.0000}$ | $0.0006_{\pm 0.0000}$ | $0.0006_{\pm 0.0000}$ | $0.0003_{\pm 0.0000}$ | $0.0000_{\pm 0.0000}$ | $0.0000_{\pm 0.0000}$ |
| L3 + Omics | $0.0071_{\pm 0.0032}$ | $0.0013_{\pm 0.0002}$ | $0.0021_{\pm 0.0002}$ | $0.0011_{\pm 0.0001}$ | $0.0032_{\pm 0.0055}$ | $0.0021_{\pm 0.0037}$ |
| L3 + Omics + KG | $0.0125_{\pm 0.0032}$ | $0.0029_{\pm 0.0003}$ | $0.0043_{\pm 0.0002}$ | $0.0022_{\pm 0.0001}$ | $0.0085_{\pm 0.0037}$ | $0.0122_{\pm 0.0033}$ |
| L3-FT(Med) + Omics | $0.0179_{\pm 0.0045}$ | $0.0133_{\pm 0.0064}$ | $0.0115_{\pm 0.0044}$ | $0.0059_{\pm 0.0023}$ | $0.0116_{\pm 0.0097}$ | $0.0122_{\pm 0.0072}$ |
| L3-FT(Med) + Omics + KG | $0.0158_{\pm 0.0030}$ | $0.0058_{\pm 0.0011}$ | $0.0074_{\pm 0.0016}$ | $0.0038_{\pm 0.0010}$ | $0.0106_{\pm 0.0048}$ | $0.0132_{\pm 0.0040}$ |
| L3-FT(QA) + Omics | $0.5250_{\pm 0.0282}$ | $0.4959_{\pm 0.0435}$ | $0.5094_{\pm 0.0365}$ | $0.3449_{\pm 0.0338}$ | $0.8889_{\pm 0.0168}$ | $0.8693_{\pm 0.0157}$ |
| L3-FT(QA) + Omics + KG | $0.5185_{\pm 0.0240}$ | $0.4908_{\pm 0.0402}$ | $0.5038_{\pm 0.0327}$ | $0.3393_{\pm 0.0298}$ | $0.8794_{\pm 0.0114}$ | $0.8529_{\pm 0.0153}$ |
| G-Retriever + pre-GAT | $0.4763_{\pm 0.0004}$ | $0.3929_{\pm 0.0063}$ | $0.4286_{\pm 0.0044}$ | $0.2757_{\pm 0.0038}$ | $0.8804_{\pm 0.0037}$ | $0.8550_{\pm 0.0046}$ |
| RoG | $0.5248_{\pm 0.0134}$ | $0.4726_{\pm 0.0445}$ | $0.4924_{\pm 0.0323}$ | $0.3338_{\pm 0.0267}$ | $0.8593_{\pm 0.0318}$ | $0.8450_{\pm 0.0350}$ |
| SubgraphRAG | $0.5280_{\pm 0.0044}$ | $0.4617_{\pm 0.0027}$ | $0.4860_{\pm 0.0033}$ | $0.3269_{\pm 0.0024}$ | $0.8624_{\pm 0.0120}$ | $0.8476_{\pm 0.0167}$ |
| GNN-RAG | $0.5258_{\pm 0.0126}$ | $0.4735_{\pm 0.0190}$ | $0.4935_{\pm 0.0168}$ | $0.3345_{\pm 0.0134}$ | $0.8656_{\pm 0.0302}$ | $0.8323_{\pm 0.0205}$ |
| GALAX | $\mathbf{0.5472_{\pm 0.0053}}$ | $0.5332_{\pm 0.0031}$ | $0.5399_{\pm 0.0041}$ | $0.3726_{\pm 0.0037}$ | $\mathbf{0.9249_{\pm 0.0048}}$ | $0.8815_{\pm 0.0033}$ |
| GALAX (Qwen2.5-7B) | $0.5445_{\pm 0.0114}$ | $\mathbf{0.5405_{\pm 0.0101}}$ | $\mathbf{0.5422_{\pm 0.0104}}$ | $\mathbf{0.3744_{\pm 0.0098}}$ | $0.9079_{\pm 0.0084}$ | $\mathbf{0.8841_{\pm 0.0126}}$ |

**Table 17:** Model performance on LUAD

| Model | Precision ↑ | Recall ↑ | F1 ↑ | Jaccard ↑ | Hit@5 ↑ | Hit@10 ↑ |
|---|---|---|---|---|---|---|
| M2T | 0.0020 | 0.0014 | 0.0017 | 0.0008 | 0.0000 | 0.0000 |
| GAT | $0.0000_{\pm 0.0000}$ | $0.0000_{\pm 0.0000}$ | $0.0000_{\pm 0.0000}$ | $0.0000_{\pm 0.0000}$ | $0.0000_{\pm 0.0000}$ | $0.0000_{\pm 0.0000}$ |
| L3 + Omics | $0.0079_{\pm 0.0137}$ | $0.0005_{\pm 0.0008}$ | $0.0009_{\pm 0.0016}$ | $0.0005_{\pm 0.0008}$ | $0.0095_{\pm 0.0165}$ | $0.0048_{\pm 0.0082}$ |
| L3 + Omics + KG | $0.0014_{\pm 0.0025}$ | $0.0010_{\pm 0.0016}$ | $0.0011_{\pm 0.0020}$ | $0.0006_{\pm 0.0010}$ | $0.0000_{\pm 0.0000}$ | $0.0000_{\pm 0.0000}$ |
| L3-FT(Med) + Omics | $0.0091_{\pm 0.0018}$ | $0.0105_{\pm 0.0044}$ | $0.0079_{\pm 0.0022}$ | $0.0040_{\pm 0.0011}$ | $0.0000_{\pm 0.0000}$ | $0.0000_{\pm 0.0000}$ |
| L3-FT(Med) + Omics + KG | $0.0081_{\pm 0.0071}$ | $0.0024_{\pm 0.0016}$ | $0.0025_{\pm 0.0002}$ | $0.0013_{\pm 0.0001}$ | $0.0095_{\pm 0.0165}$ | $0.0048_{\pm 0.0082}$ |
| L3-FT(CRISPR) + Omics | $0.5201_{\pm 0.0408}$ | $0.4905_{\pm 0.0532}$ | $0.5045_{\pm 0.0475}$ | $0.3396_{\pm 0.0433}$ | $0.8476_{\pm 0.0165}$ | $0.8667_{\pm 0.0218}$ |
| L3-FT(CRISPR) + Omics + KG | $0.5214_{\pm 0.0242}$ | $0.4952_{\pm 0.0432}$ | $0.5073_{\pm 0.0343}$ | $0.3416_{\pm 0.0314}$ | $0.7905_{\pm 0.0436}$ | $0.8048_{\pm 0.0541}$ |
| G-Retriever + pre-GAT | $0.4642_{\pm 0.0181}$ | $0.3881_{\pm 0.0264}$ | $0.4204_{\pm 0.0233}$ | $0.2671_{\pm 0.0188}$ | $0.8857_{\pm 0.0000}$ | $0.8524_{\pm 0.0165}$ |
| RoG | $0.5213_{\pm 0.0227}$ | $0.4562_{\pm 0.0848}$ | $0.4793_{\pm 0.0630}$ | $0.3228_{\pm 0.0503}$ | $0.8095_{\pm 0.0436}$ | $0.8238_{\pm 0.0218}$ |
| SubgraphRAG | $0.5123_{\pm 0.0105}$ | $0.4448_{\pm 0.0386}$ | $0.4684_{\pm 0.0279}$ | $0.3114_{\pm 0.0237}$ | $0.8190_{\pm 0.0165}$ | $0.8238_{\pm 0.0082}$ |
| GNN-RAG | $0.5334_{\pm 0.0225}$ | $0.5052_{\pm 0.0170}$ | $0.5165_{\pm 0.0161}$ | $0.3563_{\pm 0.0127}$ | $0.7905_{\pm 0.0719}$ | $0.7571_{\pm 0.0623}$ |
| GALAX | $0.5345_{\pm 0.0185}$ | $0.5157_{\pm 0.0043}$ | $0.5247_{\pm 0.0109}$ | $0.3581_{\pm 0.0101}$ | $\mathbf{0.9238_{\pm 0.0436}}$ | $\mathbf{0.8810_{\pm 0.0082}}$ |
| GALAX (Qwen2.5-7B) | $\mathbf{0.5475_{\pm 0.0019}}$ | $\mathbf{0.5462_{\pm 0.0111}}$ | $\mathbf{0.5465_{\pm 0.0050}}$ | $\mathbf{0.3778_{\pm 0.0046}}$ | $0.9048_{\pm 0.0165}$ | $0.8667_{\pm 0.0082}$ |

**Table 18:** Model performance on BRCA

| Model | Precision ↑ | Recall ↑ | F1 ↑ | Jaccard ↑ | Hit@5 ↑ | Hit@10 ↑ |
|---|---|---|---|---|---|---|
| M2T | 0.0000 | 0.0000 | 0.0000 | 0.0000 | 0.0000 | 0.0000 |
| GAT | $0.0033_{\pm 0.0000}$ | $0.0033_{\pm 0.0000}$ | $0.0033_{\pm 0.0000}$ | $0.0017_{\pm 0.0000}$ | $0.0000_{\pm 0.0000}$ | $0.0000_{\pm 0.0000}$ |
| L3 + Omics | $0.0020_{\pm 0.0035}$ | $0.0017_{\pm 0.0029}$ | $0.0018_{\pm 0.0032}$ | $0.0009_{\pm 0.0016}$ | $0.0000_{\pm 0.0000}$ | $0.0000_{\pm 0.0000}$ |
| L3 + Omics + KG | $0.0073_{\pm 0.0068}$ | $0.0033_{\pm 0.0029}$ | $0.0044_{\pm 0.0038}$ | $0.0022_{\pm 0.0019}$ | $0.0111_{\pm 0.0192}$ | $0.0056_{\pm 0.0096}$ |
| L3-FT(Med) + Omics | $0.0110_{\pm 0.0086}$ | $0.0106_{\pm 0.0075}$ | $0.0093_{\pm 0.0068}$ | $0.0047_{\pm 0.0035}$ | $0.0000_{\pm 0.0000}$ | $0.0111_{\pm 0.0192}$ |
| L3-FT(Med) + Omics + KG | $0.0149_{\pm 0.0057}$ | $0.0050_{\pm 0.0000}$ | $0.0071_{\pm 0.0012}$ | $0.0036_{\pm 0.0006}$ | $0.0000_{\pm 0.0000}$ | $0.0111_{\pm 0.0192}$ |
| L3-FT(CRISPR) + Omics | $0.5074_{\pm 0.0498}$ | $0.4856_{\pm 0.0570}$ | $0.4956_{\pm 0.0535}$ | $0.3336_{\pm 0.0491}$ | $\mathbf{0.8889_{\pm 0.0509}}$ | $0.8389_{\pm 0.0096}$ |
| L3-FT(CRISPR) + Omics + KG | $0.4856_{\pm 0.0395}$ | $0.4656_{\pm 0.0436}$ | $0.4751_{\pm 0.0412}$ | $0.3142_{\pm 0.0354}$ | $0.8778_{\pm 0.0192}$ | $0.8222_{\pm 0.0347}$ |
| G-Retriever + pre-GAT | $0.4414_{\pm 0.0099}$ | $0.3772_{\pm 0.0010}$ | $0.4062_{\pm 0.0034}$ | $0.2607_{\pm 0.0032}$ | $0.8667_{\pm 0.0000}$ | $\mathbf{0.8667_{\pm 0.0000}}$ |
| RoG | $0.4791_{\pm 0.0575}$ | $0.4311_{\pm 0.0721}$ | $0.4489_{\pm 0.0658}$ | $0.2999_{\pm 0.0528}$ | $0.7667_{\pm 0.0577}$ | $0.7611_{\pm 0.1110}$ |
| SubgraphRAG | $0.4708_{\pm 0.0317}$ | $0.3917_{\pm 0.0376}$ | $0.4196_{\pm 0.0339}$ | $0.2742_{\pm 0.0271}$ | $0.7556_{\pm 0.0839}$ | $0.7333_{\pm 0.1014}$ |
| GNN-RAG | $0.4787_{\pm 0.0453}$ | $0.4389_{\pm 0.0584}$ | $0.4543_{\pm 0.0531}$ | $0.3025_{\pm 0.0428}$ | $0.8444_{\pm 0.0385}$ | $0.8222_{\pm 0.0674}$ |
| GALAX | $\mathbf{0.5608_{\pm 0.0031}}$ | $\mathbf{0.5533_{\pm 0.0033}}$ | $\mathbf{0.5569_{\pm 0.0028}}$ | $\mathbf{0.3886_{\pm 0.0022}}$ | $\mathbf{0.8889_{\pm 0.0839}}$ | $0.8500_{\pm 0.0441}$ |
| GALAX (Qwen2.5-7B) | $0.5171_{\pm 0.0474}$ | $0.5206_{\pm 0.0419}$ | $0.5188_{\pm 0.0448}$ | $0.3532_{\pm 0.0392}$ | $0.8556_{\pm 0.0385}$ | $0.8000_{\pm 0.0764}$ |

**Table 19:** Model performance on COAD/READ

| Model | Precision ↑ | Recall ↑ | F1 ↑ | Jaccard ↑ | Hit@5 ↑ | Hit@10 ↑ |
|---|---|---|---|---|---|---|
| M2T | 0.0023 | 0.0017 | 0.0019 | 0.0010 | 0.0000 | 0.0000 |
| GAT | $0.0000_{\pm0.0000}$ | $0.0000_{\pm0.0000}$ | $0.0000_{\pm0.0000}$ | $0.0000_{\pm0.0000}$ | $0.0000_{\pm0.0000}$ | $0.0000_{\pm0.0000}$ |
| L3 + Omics | $0.0233_{\pm0.0252}$ | $0.0020_{\pm0.0020}$ | $0.0035_{\pm0.0033}$ | $0.0018_{\pm0.0017}$ | $0.0133_{\pm0.0231}$ | $0.0067_{\pm0.0115}$ |
| L3 + Omics + KG | $0.0055_{\pm0.0055}$ | $0.0033_{\pm0.0031}$ | $0.0041_{\pm0.0039}$ | $0.0021_{\pm0.0020}$ | $0.0000_{\pm0.0000}$ | $0.0000_{\pm0.0000}$ |
| L3-FT(Med) + Omics | $0.0300_{\pm0.0249}$ | $0.0220_{\pm0.0278}$ | $0.0199_{\pm0.0214}$ | $0.0104_{\pm0.0114}$ | $0.0267_{\pm0.0462}$ | $0.0333_{\pm0.0306}$ |
| L3-FT(Med) + Omics + KG | $0.0044_{\pm0.0038}$ | $0.0033_{\pm0.0031}$ | $0.0032_{\pm0.0028}$ | $0.0016_{\pm0.0014}$ | $0.0000_{\pm0.0000}$ | $0.0000_{\pm0.0000}$ |
| L3-FT(CRISPR) + Omics | $0.5011_{\pm0.0335}$ | $0.4727_{\pm0.0514}$ | $0.4862_{\pm0.0430}$ | $0.3230_{\pm0.0384}$ | $0.8400_{\pm0.0800}$ | $0.8267_{\pm0.0808}$ |
| L3-FT(CRISPR) + Omics + KG | $0.5116_{\pm0.0216}$ | $0.4807_{\pm0.0469}$ | $0.4950_{\pm0.0351}$ | $0.3301_{\pm0.0312}$ | $0.8800_{\pm0.0000}$ | $0.8333_{\pm0.0231}$ |
| G-Retriever + pre-GAT | $0.4417_{\pm0.0005}$ | $0.3680_{\pm0.0104}$ | $0.3995_{\pm0.0073}$ | $0.2504_{\pm0.0062}$ | $0.7200_{\pm0.0000}$ | $0.7667_{\pm0.0115}$ |
| RoG | $0.5184_{\pm0.0462}$ | $0.4627_{\pm0.0803}$ | $0.4850_{\pm0.0658}$ | $0.3254_{\pm0.0519}$ | $0.8533_{\pm0.0611}$ | $0.8600_{\pm0.0400}$ |
| SubgraphRAG | $0.5306_{\pm0.0066}$ | $0.4627_{\pm0.0854}$ | $0.4871_{\pm0.0556}$ | $0.3243_{\pm0.0473}$ | $\mathbf{0.9333_{\pm0.0231}}$ | $0.8533_{\pm0.0115}$ |
| GNN-RAG | $\mathbf{0.5383_{\pm0.0306}}$ | $0.4627_{\pm0.0463}$ | $0.4909_{\pm0.0404}$ | $0.3295_{\pm0.0352}$ | $0.8533_{\pm0.0462}$ | $\mathbf{0.8867_{\pm0.0115}}$ |
| GALAX | $0.5272_{\pm0.0143}$ | $0.5120_{\pm0.0087}$ | $\mathbf{0.5192_{\pm0.0103}}$ | $\mathbf{0.3512_{\pm0.0098}}$ | $0.9200_{\pm0.0693}$ | $0.8400_{\pm0.0200}$ |
| GALAX (Qwen2.5-7B) | $0.5177_{\pm0.0124}$ | $\mathbf{0.5187_{\pm0.0061}}$ | $0.5177_{\pm0.0083}$ | $0.3498_{\pm0.0076}$ | $0.9200_{\pm0.0800}$ | $0.8733_{\pm0.0503}$ |

**Table 20:** Model performance on PAAD

| Model | Precision ↑ | Recall ↑ | F1 ↑ | Jaccard ↑ | Hit@5 ↑ | Hit@10 ↑ |
|---|---|---|---|---|---|---|
| M2T | 0.0017 | 0.0013 | 0.0014 | 0.0008 | 0.0000 | 0.0000 |
| GAT | $0.0025_{\pm0.0000}$ | $0.0025_{\pm0.0000}$ | $0.0025_{\pm0.0000}$ | $0.0013_{\pm0.0000}$ | $0.0000_{\pm0.0000}$ | $0.0000_{\pm0.0000}$ |
| L3 + Omics | $0.0000_{\pm0.0000}$ | $0.0000_{\pm0.0000}$ | $0.0000_{\pm0.0000}$ | $0.0000_{\pm0.0000}$ | $0.0000_{\pm0.0000}$ | $0.0000_{\pm0.0000}$ |
| L3 + Omics + KG | $0.0139_{\pm0.0241}$ | $0.0008_{\pm0.0014}$ | $0.0016_{\pm0.0027}$ | $0.0008_{\pm0.0014}$ | $0.0167_{\pm0.0289}$ | $0.0083_{\pm0.0144}$ |
| L3-FT(Med) + Omics | $0.0051_{\pm0.0061}$ | $0.0050_{\pm0.0043}$ | $0.0039_{\pm0.0036}$ | $0.0020_{\pm0.0018}$ | $0.0167_{\pm0.0289}$ | $0.0083_{\pm0.0144}$ |
| L3-FT(Med) + Omics + KG | $0.0005_{\pm0.0008}$ | $0.0008_{\pm0.0014}$ | $0.0006_{\pm0.0010}$ | $0.0003_{\pm0.0005}$ | $0.0000_{\pm0.0000}$ | $0.0000_{\pm0.0000}$ |
| L3-FT(CRISPR) + Omics | $0.5649_{\pm0.0559}$ | $0.5358_{\pm0.0686}$ | $0.5492_{\pm0.0605}$ | $0.3811_{\pm0.0584}$ | $0.8000_{\pm0.0500}$ | $0.8417_{\pm0.0577}$ |
| L3-FT(CRISPR) + Omics + KG | $0.5499_{\pm0.0570}$ | $0.5183_{\pm0.0794}$ | $0.5330_{\pm0.0683}$ | $0.3657_{\pm0.0650}$ | $0.8333_{\pm0.0764}$ | $0.8333_{\pm0.0520}$ |
| G-Retriever + pre-GAT | $0.5594_{\pm0.0114}$ | $0.4567_{\pm0.0058}$ | $0.5016_{\pm0.0009}$ | $0.3387_{\pm0.0007}$ | $0.8500_{\pm0.0000}$ | $0.8500_{\pm0.0000}$ |
| RoG | $0.5196_{\pm0.1322}$ | $0.4842_{\pm0.1461}$ | $0.4973_{\pm0.1406}$ | $0.3462_{\pm0.1107}$ | $0.7667_{\pm0.1528}$ | $0.7833_{\pm0.1181}$ |
| SubgraphRAG | $\mathbf{0.5873_{\pm0.0050}}$ | $0.5308_{\pm0.0029}$ | $0.5518_{\pm0.0037}$ | $0.3869_{\pm0.0028}$ | $\mathbf{0.8833_{\pm0.0289}}$ | $\mathbf{0.8750_{\pm0.0000}}$ |
| GNN-RAG | $0.4923_{\pm0.0762}$ | $0.4283_{\pm0.0204}$ | $0.4522_{\pm0.0411}$ | $0.3068_{\pm0.0248}$ | $0.7667_{\pm0.1443}$ | $0.7917_{\pm0.1233}$ |
| GALAX | $0.5771_{\pm0.0174}$ | $\mathbf{0.5708_{\pm0.0118}}$ | $\mathbf{0.5739_{\pm0.0145}}$ | $\mathbf{0.4044_{\pm0.0154}}$ | $0.8333_{\pm0.0289}$ | $0.8500_{\pm0.0000}$ |
| GALAX (Qwen2.5-7B) | $0.5394_{\pm0.0417}$ | $0.5383_{\pm0.0427}$ | $0.5387_{\pm0.0422}$ | $0.3718_{\pm0.0402}$ | $0.8167_{\pm0.0289}$ | $0.8333_{\pm0.0144}$ |

**Table 21:** Model performance on GBM

| Model | Precision ↑ | Recall ↑ | F1 ↑ | Jaccard ↑ | Hit@5 ↑ | Hit@10 ↑ |
|---|---|---|---|---|---|---|
| M2T | 0.0000 | 0.0000 | 0.0000 | 0.0000 | 0.0000 | 0.0000 |
| GAT | $0.0000_{\pm0.0000}$ | $0.0000_{\pm0.0000}$ | $0.0000_{\pm0.0000}$ | $0.0000_{\pm0.0000}$ | $0.0000_{\pm0.0000}$ | $0.0000_{\pm0.0000}$ |
| L3 + Omics | $0.0056_{\pm0.0096}$ | $0.0011_{\pm0.0019}$ | $0.0019_{\pm0.0032}$ | $0.0009_{\pm0.0016}$ | $0.0000_{\pm0.0000}$ | $0.0000_{\pm0.0000}$ |
| L3 + Omics + KG | $0.0000_{\pm0.0000}$ | $0.0000_{\pm0.0000}$ | $0.0000_{\pm0.0000}$ | $0.0000_{\pm0.0000}$ | $0.0000_{\pm0.0000}$ | $0.0000_{\pm0.0000}$ |
| L3-FT(Med) + Omics | $0.0069_{\pm0.0045}$ | $0.0133_{\pm0.0088}$ | $0.0090_{\pm0.0059}$ | $0.0046_{\pm0.0030}$ | $0.0000_{\pm0.0000}$ | $0.0000_{\pm0.0000}$ |
| L3-FT(Med) + Omics + KG | $0.0107_{\pm0.0158}$ | $0.0056_{\pm0.0038}$ | $0.0064_{\pm0.0074}$ | $0.0032_{\pm0.0037}$ | $0.0000_{\pm0.0000}$ | $0.0000_{\pm0.0000}$ |
| L3-FT(CRISPR) + Omics | $0.5641_{\pm0.0310}$ | $0.5333_{\pm0.0462}$ | $0.5480_{\pm0.0387}$ | $0.3794_{\pm0.0364}$ | $0.9333_{\pm0.0667}$ | $0.9000_{\pm0.0333}$ |
| L3-FT(CRISPR) + Omics + KG | $0.5391_{\pm0.0213}$ | $0.5178_{\pm0.0164}$ | $0.5281_{\pm0.0181}$ | $0.3596_{\pm0.0165}$ | $\mathbf{0.9556_{\pm0.0385}}$ | $0.8444_{\pm0.0694}$ |
| G-Retriever + pre-GAT | $0.4906_{\pm0.0075}$ | $0.4411_{\pm0.0038}$ | $0.4642_{\pm0.0010}$ | $0.3037_{\pm0.0000}$ | $\mathbf{0.9556_{\pm0.0385}}$ | $0.8333_{\pm0.0577}$ |
| RoG | $\mathbf{0.5910_{\pm0.0281}}$ | $0.5078_{\pm0.1165}$ | $0.5380_{\pm0.0853}$ | $0.3734_{\pm0.0756}$ | $\mathbf{0.9556_{\pm0.0770}}$ | $\mathbf{0.9444_{\pm0.0509}}$ |
| SubgraphRAG | $0.5066_{\pm0.0597}$ | $0.4544_{\pm0.1001}$ | $0.4707_{\pm0.0884}$ | $0.3176_{\pm0.0729}$ | $0.7778_{\pm0.1925}$ | $0.7556_{\pm0.2117}$ |
| GNN-RAG | $0.5567_{\pm0.0222}$ | $0.5478_{\pm0.0212}$ | $0.5521_{\pm0.0181}$ | $0.3855_{\pm0.0141}$ | $0.8667_{\pm0.2309}$ | $0.8111_{\pm0.1540}$ |
| GALAX | $0.5527_{\pm0.0021}$ | $0.5422_{\pm0.0139}$ | $0.5473_{\pm0.0081}$ | $0.3769_{\pm0.0079}$ | $0.9111_{\pm0.0770}$ | $0.8667_{\pm0.0000}$ |
| GALAX (Qwen2.5-7B) | $0.5715_{\pm0.0228}$ | $\mathbf{0.5656_{\pm0.0269}}$ | $\mathbf{0.5685_{\pm0.0249}}$ | $\mathbf{0.3985_{\pm0.0236}}$ | $0.9111_{\pm0.0770}$ | $0.8556_{\pm0.0385}$ |

**Table 22:** Model performance on SARC

| Model | Precision ↑ | Recall ↑ | F1 ↑ | Jaccard ↑ | Hit@5 ↑ | Hit@10 ↑ |
|---|---|---|---|---|---|---|
| M2T | 0.0047 | 0.0033 | 0.0039 | 0.0019 | 0.0000 | 0.0000 |
| GAT | $0.0000_{\pm0.0000}$ | $0.0000_{\pm0.0000}$ | $0.0000_{\pm0.0000}$ | $0.0000_{\pm0.0000}$ | $0.0000_{\pm0.0000}$ | $0.0000_{\pm0.0000}$ |
| L3 + Omics | $0.0111_{\pm0.0192}$ | $0.0033_{\pm0.0058}$ | $0.0051_{\pm0.0089}$ | $0.0026_{\pm0.0045}$ | $0.0000_{\pm0.0000}$ | $0.0000_{\pm0.0000}$ |
| L3 + Omics + KG | $0.0271_{\pm0.0062}$ | $0.0111_{\pm0.0019}$ | $0.0157_{\pm0.0029}$ | $0.0080_{\pm0.0015}$ | $0.0222_{\pm0.0385}$ | $0.0222_{\pm0.0192}$ |
| L3-FT(Med) + Omics | $0.0689_{\pm0.0631}$ | $0.0167_{\pm0.0115}$ | $0.0229_{\pm0.0180}$ | $0.0119_{\pm0.0094}$ | $0.0222_{\pm0.0385}$ | $0.0667_{\pm0.0577}$ |
| L3-FT(Med) + Omics + KG | $0.0270_{\pm0.0108}$ | $0.0089_{\pm0.0014}$ | $0.0120_{\pm0.0009}$ | $0.0061_{\pm0.0005}$ | $0.0222_{\pm0.0385}$ | $0.0222_{\pm0.0192}$ |
| L3-FT(CRISPR) + Omics | $0.5347_{\pm0.0167}$ | $0.5089_{\pm0.0353}$ | $0.5211_{\pm0.0266}$ | $0.3547_{\pm0.0253}$ | $\mathbf{0.9111_{\pm0.0385}}$ | $0.8778_{\pm0.0192}$ |
| L3-FT(CRISPR) + Omics + KG | $0.5423_{\pm0.0207}$ | $0.5022_{\pm0.0539}$ | $0.5207_{\pm0.0383}$ | $0.3570_{\pm0.0344}$ | $0.8444_{\pm0.0385}$ | $0.8444_{\pm0.0385}$ |
| G-Retriever + pre-GAT | $0.4461_{\pm0.0308}$ | $0.3667_{\pm0.0115}$ | $0.4009_{\pm0.0055}$ | $0.2518_{\pm0.0051}$ | $0.8667_{\pm0.0000}$ | $\mathbf{0.9222_{\pm0.0192}}$ |
| RoG | $\mathbf{0.5533_{\pm0.0201}}$ | $0.4589_{\pm0.0740}$ | $0.4922_{\pm0.0433}$ | $0.3324_{\pm0.0375}$ | $0.8889_{\pm0.0192}$ | $0.8778_{\pm0.0192}$ |
| SubgraphRAG | $0.5365_{\pm0.0277}$ | $0.4456_{\pm0.0168}$ | $0.4772_{\pm0.0188}$ | $0.3175_{\pm0.0166}$ | $0.8889_{\pm0.1018}$ | $0.8667_{\pm0.0882}$ |
| GNN-RAG | $0.5485_{\pm0.0097}$ | $0.4778_{\pm0.0587}$ | $0.5036_{\pm0.0423}$ | $0.3426_{\pm0.0403}$ | $0.8889_{\pm0.0385}$ | $0.8889_{\pm0.0385}$ |
| GALAX | $0.5414_{\pm0.0143}$ | $0.5144_{\pm0.0168}$ | $0.5271_{\pm0.0157}$ | $0.3602_{\pm0.0165}$ | $\mathbf{0.9111_{\pm0.1018}}$ | $0.8444_{\pm0.0509}$ |
| GALAX (Qwen2.5-7B) | $0.5383_{\pm0.0096}$ | $\mathbf{0.5311_{\pm0.0084}}$ | $\mathbf{0.5347_{\pm0.0090}}$ | $\mathbf{0.3674_{\pm0.0088}}$ | $\mathbf{0.9111_{\pm0.1018}}$ | $0.8889_{\pm0.0694}$ |

**Table 23:** Model performance on OV

| Model | Precision ↑ | Recall ↑ | F1 ↑ | Jaccard ↑ | Hit@5 ↑ | Hit@10 ↑ |
|---|---|---|---|---|---|---|
| M2T | 0.0000 | 0.0000 | 0.0000 | 0.0000 | 0.0000 | 0.0000 |
| GAT | $0.0000_{\pm0.0000}$ | $0.0000_{\pm0.0000}$ | $0.0000_{\pm0.0000}$ | $0.0000_{\pm0.0000}$ | $0.0000_{\pm0.0000}$ | $0.0000_{\pm0.0000}$ |
| L3 + Omics | $0.0241_{\pm0.0251}$ | $0.0044_{\pm0.0051}$ | $0.0075_{\pm0.0085}$ | $0.0038_{\pm0.0043}$ | $0.0000_{\pm0.0000}$ | $0.0111_{\pm0.0192}$ |
| L3 + Omics + KG | $0.0660_{\pm0.0667}$ | $0.0111_{\pm0.0102}$ | $0.0183_{\pm0.0161}$ | $0.0093_{\pm0.0082}$ | $0.0000_{\pm0.0000}$ | $0.0889_{\pm0.0770}$ |
| L3-FT(Med) + Omics | $0.0356_{\pm0.0412}$ | $0.0100_{\pm0.0067}$ | $0.0102_{\pm0.0072}$ | $0.0051_{\pm0.0036}$ | $0.0444_{\pm0.0770}$ | $0.0222_{\pm0.0385}$ |
| L3-FT(Med) + Omics + KG | $0.0797_{\pm0.0496}$ | $0.0344_{\pm0.0280}$ | $0.0457_{\pm0.0375}$ | $0.0253_{\pm0.0226}$ | $0.0000_{\pm0.0000}$ | $0.0333_{\pm0.0333}$ |
| L3-FT(CRISPR) + Omics | $0.5469_{\pm0.0274}$ | $0.5167_{\pm0.0406}$ | $0.5307_{\pm0.0325}$ | $0.3631_{\pm0.0295}$ | $1.0000_{\pm0.0000}$ | $0.9111_{\pm0.0694}$ |
| L3-FT(CRISPR) + Omics + KG | $0.5439_{\pm0.0066}$ | $0.4989_{\pm0.0195}$ | $0.5200_{\pm0.0123}$ | $0.3542_{\pm0.0095}$ | $0.9556_{\pm0.0385}$ | $0.9444_{\pm0.0385}$ |
| G-Retriever + pre-GAT | $0.5413_{\pm0.0525}$ | $0.4033_{\pm0.0173}$ | $0.4575_{\pm0.0067}$ | $0.2991_{\pm0.0041}$ | $1.0000_{\pm0.0000}$ | $\mathbf{0.9667_{\pm0.0000}}$ |
| RoG | $0.5708_{\pm0.0158}$ | $0.5289_{\pm0.0523}$ | $0.5453_{\pm0.0343}$ | $0.3774_{\pm0.0317}$ | $0.9556_{\pm0.0385}$ | $0.9222_{\pm0.0385}$ |
| SubgraphRAG | $\mathbf{0.6143_{\pm0.0414}}$ | $0.4833_{\pm0.0384}$ | $0.5289_{\pm0.0215}$ | $0.3650_{\pm0.0185}$ | $0.9333_{\pm0.0667}$ | $0.9444_{\pm0.0385}$ |
| GNN-RAG | $0.5687_{\pm0.0300}$ | $0.4600_{\pm0.0145}$ | $0.5026_{\pm0.0061}$ | $0.3380_{\pm0.0036}$ | $0.8444_{\pm0.1540}$ | $0.8111_{\pm0.1540}$ |
| GALAX | $0.5489_{\pm0.0176}$ | $0.5344_{\pm0.0259}$ | $0.5413_{\pm0.0219}$ | $0.3714_{\pm0.0206}$ | $0.8444_{\pm0.0385}$ | $0.8667_{\pm0.0333}$ |
| GALAX (Qwen2.5-7B) | $0.5643_{\pm0.0190}$ | $\mathbf{0.5689_{\pm0.0550}}$ | $\mathbf{0.5654_{\pm0.0355}}$ | $\mathbf{0.3957_{\pm0.0349}}$ | $0.9778_{\pm0.0385}$ | $0.9556_{\pm0.0192}$ |

**Table 24:** Model performance on SKCM

| Model | Precision ↑ | Recall ↑ | F1 ↑ | Jaccard ↑ | Hit@5 ↑ | Hit@10 ↑ |
|---|---|---|---|---|---|---|
| M2T | 0.0055 | 0.0033 | 0.0041 | 0.0021 | 0.0000 | 0.0333 |
| GAT | $0.0000_{\pm0.0000}$ | $0.0000_{\pm0.0000}$ | $0.0000_{\pm0.0000}$ | $0.0000_{\pm0.0000}$ | $0.0000_{\pm0.0000}$ | $0.0000_{\pm0.0000}$ |
| L3 + Omics | $0.0000_{\pm0.0000}$ | $0.0000_{\pm0.0000}$ | $0.0000_{\pm0.0000}$ | $0.0000_{\pm0.0000}$ | $0.0000_{\pm0.0000}$ | $0.0000_{\pm0.0000}$ |
| L3 + Omics + KG | $0.0028_{\pm0.0049}$ | $0.0011_{\pm0.0019}$ | $0.0016_{\pm0.0028}$ | $0.0008_{\pm0.0014}$ | $0.0000_{\pm0.0000}$ | $0.0111_{\pm0.0192}$ |
| L3-FT(Med) + Omics | $0.0047_{\pm0.0041}$ | $0.0089_{\pm0.0069}$ | $0.0061_{\pm0.0052}$ | $0.0031_{\pm0.0026}$ | $0.0000_{\pm0.0000}$ | $0.0000_{\pm0.0000}$ |
| L3-FT(Med) + Omics + KG | $0.0000_{\pm0.0000}$ | $0.0000_{\pm0.0000}$ | $0.0000_{\pm0.0000}$ | $0.0000_{\pm0.0000}$ | $0.0000_{\pm0.0000}$ | $0.0000_{\pm0.0000}$ |
| L3-FT(CRISPR) + Omics | $0.5145_{\pm0.0437}$ | $0.4922_{\pm0.0455}$ | $0.5028_{\pm0.0443}$ | $0.3399_{\pm0.0401}$ | $0.8444_{\pm0.1388}$ | $0.8556_{\pm0.0385}$ |
| L3-FT(CRISPR) + Omics + KG | $0.4871_{\pm0.0529}$ | $0.4678_{\pm0.0626}$ | $0.4771_{\pm0.0579}$ | $0.3164_{\pm0.0514}$ | $0.8667_{\pm0.0667}$ | $0.8667_{\pm0.0667}$ |
| G-Retriever + pre-GAT | $0.5123_{\pm0.0545}$ | $0.4100_{\pm0.0545}$ | $0.4543_{\pm0.0258}$ | $0.2958_{\pm0.0222}$ | $0.9333_{\pm0.0000}$ | $0.8778_{\pm0.0192}$ |
| RoG | $0.5201_{\pm0.0255}$ | $0.4522_{\pm0.0681}$ | $0.4797_{\pm0.0520}$ | $0.3179_{\pm0.0461}$ | $0.8000_{\pm0.2309}$ | $0.8111_{\pm0.1895}$ |
| SubgraphRAG | $0.4929_{\pm0.0245}$ | $0.4222_{\pm0.0403}$ | $0.4502_{\pm0.0304}$ | $0.2929_{\pm0.0260}$ | $0.6889_{\pm0.2143}$ | $0.7222_{\pm0.2143}$ |
| GNN-RAG | $0.5298_{\pm0.0203}$ | $0.4956_{\pm0.0367}$ | $0.5092_{\pm0.0257}$ | $0.3447_{\pm0.0229}$ | $0.9556_{\pm0.0385}$ | $0.9222_{\pm0.0385}$ |
| GALAX | $0.5385_{\pm0.0116}$ | $0.5233_{\pm0.0200}$ | $0.5307_{\pm0.0160}$ | $0.3638_{\pm0.0152}$ | $1.0000_{\pm0.0000}$ | $0.9667_{\pm0.0000}$ |
| GALAX (Qwen2.5-7B) | $\mathbf{0.5651_{\pm0.0206}}$ | $\mathbf{0.5589_{\pm0.0280}}$ | $\mathbf{0.5619_{\pm0.0245}}$ | $\mathbf{0.3938_{\pm0.0213}}$ | $0.8889_{\pm0.0770}$ | $0.9111_{\pm0.0962}$ |

**Table 25:** Model performance on ESCA

| Model | Precision ↑ | Recall ↑ | F1 ↑ | Jaccard ↑ | Hit@5 ↑ | Hit@10 ↑ |
|---|---|---|---|---|---|---|
| M2T | 0.0000 | 0.0000 | 0.0000 | 0.0000 | 0.0000 | 0.0000 |
| GAT | $0.0000_{\pm0.0000}$ | $0.0000_{\pm0.0000}$ | $0.0000_{\pm0.0000}$ | $0.0000_{\pm0.0000}$ | $0.0000_{\pm0.0000}$ | $0.0000_{\pm0.0000}$ |
| L3 + Omics | $0.0000_{\pm0.0000}$ | $0.0000_{\pm0.0000}$ | $0.0000_{\pm0.0000}$ | $0.0000_{\pm0.0000}$ | $0.0000_{\pm0.0000}$ | $0.0000_{\pm0.0000}$ |
| L3 + Omics + KG | $0.0000_{\pm0.0000}$ | $0.0000_{\pm0.0000}$ | $0.0000_{\pm0.0000}$ | $0.0000_{\pm0.0000}$ | $0.0000_{\pm0.0000}$ | $0.0000_{\pm0.0000}$ |
| L3-FT(Med) + Omics | $0.0143_{\pm0.0036}$ | $0.0178_{\pm0.0038}$ | $0.0139_{\pm0.0021}$ | $0.0071_{\pm0.0011}$ | $0.0000_{\pm0.0000}$ | $0.0000_{\pm0.0000}$ |
| L3-FT(Med) + Omics + KG | $0.0000_{\pm0.0000}$ | $0.0000_{\pm0.0000}$ | $0.0000_{\pm0.0000}$ | $0.0000_{\pm0.0000}$ | $0.0000_{\pm0.0000}$ | $0.0000_{\pm0.0000}$ |
| L3-FT(CRISPR) + Omics | $0.5210_{\pm0.0590}$ | $0.4989_{\pm0.0691}$ | $0.5093_{\pm0.0637}$ | $0.3438_{\pm0.0577}$ | $0.9333_{\pm0.0667}$ | $0.9333_{\pm0.0000}$ |
| L3-FT(CRISPR) + Omics + KG | $0.5146_{\pm0.0187}$ | $0.4878_{\pm0.0051}$ | $0.5003_{\pm0.0074}$ | $0.3346_{\pm0.0059}$ | $0.9111_{\pm0.0385}$ | $0.9333_{\pm0.0000}$ |
| G-Retriever + pre-GAT | $0.5083_{\pm0.0142}$ | $0.4156_{\pm0.0038}$ | $0.4556_{\pm0.0076}$ | $0.2959_{\pm0.0064}$ | $0.9778_{\pm0.0385}$ | $0.9444_{\pm0.0385}$ |
| RoG | $0.5275_{\pm0.0188}$ | $0.4622_{\pm0.0534}$ | $0.4877_{\pm0.0400}$ | $0.3262_{\pm0.0358}$ | $0.9111_{\pm0.0770}$ | $0.9556_{\pm0.0385}$ |
| SubgraphRAG | $0.5373_{\pm0.0043}$ | $0.4744_{\pm0.0372}$ | $0.4990_{\pm0.0259}$ | $0.3385_{\pm0.0210}$ | $0.8889_{\pm0.0385}$ | $0.9333_{\pm0.0000}$ |
| GNN-RAG | $0.5405_{\pm0.0036}$ | $0.5067_{\pm0.0404}$ | $0.5198_{\pm0.0283}$ | $0.3548_{\pm0.0243}$ | $1.0000_{\pm0.0000}$ | $0.9778_{\pm0.0192}$ |
| GALAX | $\mathbf{0.5775_{\pm0.0204}}$ | $\mathbf{0.5667_{\pm0.0260}}$ | $\mathbf{0.5720_{\pm0.0233}}$ | $\mathbf{0.4031_{\pm0.0246}}$ | $0.9556_{\pm0.0385}$ | $0.9667_{\pm0.0333}$ |
| GALAX (Qwen2.5-7B) | $0.5706_{\pm0.0228}$ | $0.5567_{\pm0.0145}$ | $0.5634_{\pm0.0173}$ | $0.3936_{\pm0.0156}$ | $0.9333_{\pm0.1155}$ | $0.9333_{\pm0.0882}$ |

**Table 26:** Model performance on SCLC

| Model | Precision ↑ | Recall ↑ | F1 ↑ | Jaccard ↑ | Hit@5 ↑ | Hit@10 ↑ |
|---|---|---|---|---|---|---|
| M2T | 0.0000 | 0.0000 | 0.0000 | 0.0000 | 0.0000 | 0.0000 |
| GAT | $0.0000_{\pm0.0000}$ | $0.0000_{\pm0.0000}$ | $0.0000_{\pm0.0000}$ | $0.0000_{\pm0.0000}$ | $0.0000_{\pm0.0000}$ | $0.0000_{\pm0.0000}$ |
| L3 + Omics | $0.0000_{\pm0.0000}$ | $0.0000_{\pm0.0000}$ | $0.0000_{\pm0.0000}$ | $0.0000_{\pm0.0000}$ | $0.0000_{\pm0.0000}$ | $0.0000_{\pm0.0000}$ |
| L3 + Omics + KG | $0.0000_{\pm0.0000}$ | $0.0000_{\pm0.0000}$ | $0.0000_{\pm0.0000}$ | $0.0000_{\pm0.0000}$ | $0.0000_{\pm0.0000}$ | $0.0000_{\pm0.0000}$ |
| L3-FT(Med) + Omics | $0.0184_{\pm0.0212}$ | $0.0389_{\pm0.0448}$ | $0.0249_{\pm0.0286}$ | $0.0131_{\pm0.0153}$ | $0.0000_{\pm0.0000}$ | $0.0000_{\pm0.0000}$ |
| L3-FT(Med) + Omics + KG | $0.0000_{\pm0.0000}$ | $0.0000_{\pm0.0000}$ | $0.0000_{\pm0.0000}$ | $0.0000_{\pm0.0000}$ | $0.0000_{\pm0.0000}$ | $0.0000_{\pm0.0000}$ |
| L3-FT(CRISPR) + Omics | $0.5337_{\pm0.0349}$ | $0.4989_{\pm0.0479}$ | $0.5146_{\pm0.0378}$ | $0.3476_{\pm0.0341}$ | $0.8667_{\pm0.1155}$ | $0.8556_{\pm0.0694}$ |
| L3-FT(CRISPR) + Omics + KG | $0.4900_{\pm0.0597}$ | $0.4667_{\pm0.0677}$ | $0.4778_{\pm0.0638}$ | $0.3160_{\pm0.0555}$ | $0.9333_{\pm0.0667}$ | $0.8778_{\pm0.0694}$ |
| G-Retriever + pre-GAT | $0.4203_{\pm0.0263}$ | $0.3667_{\pm0.0115}$ | $0.3910_{\pm0.0180}$ | $0.2434_{\pm0.0140}$ | $0.8667_{\pm0.0000}$ | $0.8000_{\pm0.0000}$ |
| RoG | $0.4971_{\pm0.1497}$ | $0.4689_{\pm0.1375}$ | $0.4786_{\pm0.1399}$ | $0.3313_{\pm0.1074}$ | $0.7778_{\pm0.2694}$ | $0.8000_{\pm0.2028}$ |
| SubgraphRAG | $0.5516_{\pm0.0586}$ | $0.4578_{\pm0.1711}$ | $0.4917_{\pm0.1315}$ | $0.3346_{\pm0.1131}$ | $0.8889_{\pm0.1388}$ | $\mathbf{0.9111_{\pm0.1018}}$ |
| GNN-RAG | $0.5503_{\pm0.0732}$ | $0.4844_{\pm0.1771}$ | $0.5096_{\pm0.1386}$ | $0.3512_{\pm0.1203}$ | $0.8667_{\pm0.0000}$ | $0.8556_{\pm0.0192}$ |
| GALAX | $\mathbf{0.5850_{\pm0.0110}}$ | $\mathbf{0.5822_{\pm0.0102}}$ | $\mathbf{0.5836_{\pm0.0105}}$ | $\mathbf{0.4143_{\pm0.0102}}$ | $\mathbf{0.9778_{\pm0.0385}}$ | $0.9000_{\pm0.0333}$ |
| GALAX (Qwen2.5-7B) | $0.5706_{\pm0.0190}$ | $0.5667_{\pm0.0176}$ | $0.5686_{\pm0.0182}$ | $0.3988_{\pm0.0189}$ | $0.9556_{\pm0.0385}$ | $0.9000_{\pm0.0333}$ |

**Table 27:** Model performance on HNSC

| Model | Precision ↑ | Recall ↑ | F1 ↑ | Jaccard ↑ | Hit@5 ↑ | Hit@10 ↑ |
|---|---|---|---|---|---|---|
| M2T | 0.0000 | 0.0000 | 0.0000 | 0.0000 | 0.0000 | 0.0000 |
| GAT | $0.0000_{\pm0.0000}$ | $0.0000_{\pm0.0000}$ | $0.0000_{\pm0.0000}$ | $0.0000_{\pm0.0000}$ | $0.0000_{\pm0.0000}$ | $0.0000_{\pm0.0000}$ |
| L3 + Omics | $0.0000_{\pm0.0000}$ | $0.0000_{\pm0.0000}$ | $0.0000_{\pm0.0000}$ | $0.0000_{\pm0.0000}$ | $0.0000_{\pm0.0000}$ | $0.0000_{\pm0.0000}$ |
| L3 + Omics + KG | $0.0128_{\pm0.0222}$ | $0.0017_{\pm0.0029}$ | $0.0029_{\pm0.0051}$ | $0.0015_{\pm0.0026}$ | $0.0333_{\pm0.0577}$ | $0.0167_{\pm0.0289}$ |
| L3-FT(Med) + Omics | $0.0082_{\pm0.0036}$ | $0.0100_{\pm0.0050}$ | $0.0079_{\pm0.0016}$ | $0.0040_{\pm0.0008}$ | $0.0000_{\pm0.0000}$ | $0.0000_{\pm0.0000}$ |
| L3-FT(Med) + Omics + KG | $0.0000_{\pm0.0000}$ | $0.0000_{\pm0.0000}$ | $0.0000_{\pm0.0000}$ | $0.0000_{\pm0.0000}$ | $0.0000_{\pm0.0000}$ | $0.0000_{\pm0.0000}$ |
| L3-FT(CRISPR) + Omics | $0.4374_{\pm0.0214}$ | $0.4167_{\pm0.0208}$ | $0.4267_{\pm0.0208}$ | $0.2718_{\pm0.0174}$ | $\mathbf{1.0000_{\pm0.0000}}$ | $\mathbf{0.9667_{\pm0.0577}}$ |
| L3-FT(CRISPR) + Omics + KG | $0.4787_{\pm0.0445}$ | $0.4450_{\pm0.0676}$ | $0.4606_{\pm0.0565}$ | $0.3006_{\pm0.0470}$ | $0.9667_{\pm0.0577}$ | $0.9500_{\pm0.0000}$ |
| G-Retriever + pre-GAT | $0.4413_{\pm0.0030}$ | $0.3700_{\pm0.0000}$ | $0.4021_{\pm0.0009}$ | $0.2518_{\pm0.0008}$ | $\mathbf{1.0000_{\pm0.0000}}$ | $0.9500_{\pm0.0000}$ |
| RoG | $0.4718_{\pm0.0254}$ | $0.4400_{\pm0.0673}$ | $0.4529_{\pm0.0509}$ | $0.2952_{\pm0.0404}$ | $0.9667_{\pm0.0577}$ | $0.9500_{\pm0.0500}$ |
| SubgraphRAG | $0.4648_{\pm0.0120}$ | $0.4233_{\pm0.0635}$ | $0.4398_{\pm0.0450}$ | $0.2833_{\pm0.0355}$ | $\mathbf{1.0000_{\pm0.0000}}$ | $0.9333_{\pm0.0289}$ |
| GNN-RAG | $0.4655_{\pm0.0351}$ | $0.4317_{\pm0.0813}$ | $0.4455_{\pm0.0631}$ | $0.2890_{\pm0.0506}$ | $0.9000_{\pm0.0000}$ | $0.9000_{\pm0.0000}$ |
| GALAX | $0.4482_{\pm0.0229}$ | $0.4283_{\pm0.0208}$ | $0.4380_{\pm0.0218}$ | $0.2822_{\pm0.0177}$ | $0.9333_{\pm0.0577}$ | $\mathbf{0.9667_{\pm0.0289}}$ |
| GALAX (Qwen2.5-7B) | $\mathbf{0.4787_{\pm0.0082}}$ | $\mathbf{0.4667_{\pm0.0153}}$ | $\mathbf{0.4722_{\pm0.0070}}$ | $\mathbf{0.3095_{\pm0.0064}}$ | $\mathbf{1.0000_{\pm0.0000}}$ | $0.9000_{\pm0.0866}$ |

**Table 28:** Model performance on LUSC

| Model | Precision ↑ | Recall ↑ | F1 ↑ | Jaccard ↑ | Hit@5 ↑ | Hit@10 ↑ |
|---|---|---|---|---|---|---|
| M2T | 0.0000 | 0.0000 | 0.0000 | 0.0000 | 0.0000 | 0.0000 |
| GAT | $0.0000_{\pm0.0000}$ | $0.0000_{\pm0.0000}$ | $0.0000_{\pm0.0000}$ | $0.0000_{\pm0.0000}$ | $0.0000_{\pm0.0000}$ | $0.0000_{\pm0.0000}$ |
| L3 + Omics | $0.0222_{\pm0.0192}$ | $0.0067_{\pm0.0058}$ | $0.0103_{\pm0.0089}$ | $0.0052_{\pm0.0045}$ | $0.0000_{\pm0.0000}$ | $0.0000_{\pm0.0000}$ |
| L3 + Omics + KG | $0.0062_{\pm0.0068}$ | $0.0050_{\pm0.0050}$ | $0.0055_{\pm0.0058}$ | $0.0028_{\pm0.0029}$ | $0.0000_{\pm0.0000}$ | $0.0000_{\pm0.0000}$ |
| L3-FT(Med) + Omics | $0.0165_{\pm0.0106}$ | $0.0083_{\pm0.0058}$ | $0.0085_{\pm0.0010}$ | $0.0043_{\pm0.0005}$ | $0.0000_{\pm0.0000}$ | $0.0000_{\pm0.0000}$ |
| L3-FT(Med) + Omics + KG | $0.0094_{\pm0.0082}$ | $0.0033_{\pm0.0029}$ | $0.0049_{\pm0.0043}$ | $0.0025_{\pm0.0021}$ | $0.0000_{\pm0.0000}$ | $0.0000_{\pm0.0000}$ |
| L3-FT(CRISPR) + Omics | $0.5208_{\pm0.0525}$ | $0.4767_{\pm0.0810}$ | $0.4961_{\pm0.0677}$ | $0.3320_{\pm0.0618}$ | $0.9667_{\pm0.0577}$ | $0.9167_{\pm0.0577}$ |
| L3-FT(CRISPR) + Omics + KG | $0.4656_{\pm0.0253}$ | $0.4217_{\pm0.0584}$ | $0.4420_{\pm0.0427}$ | $0.2844_{\pm0.0355}$ | $0.9333_{\pm0.1155}$ | $0.9333_{\pm0.0577}$ |
| G-Retriever + pre-GAT | $0.4563_{\pm0.0579}$ | $0.3417_{\pm0.0318}$ | $0.3902_{\pm0.0418}$ | $0.2433_{\pm0.0313}$ | $\mathbf{1.0000_{\pm0.0000}}$ | $0.9000_{\pm0.0000}$ |
| RoG | $0.5038_{\pm0.0245}$ | $0.4550_{\pm0.0954}$ | $0.4717_{\pm0.0730}$ | $0.3123_{\pm0.0583}$ | $\mathbf{1.0000_{\pm0.0000}}$ | $0.9667_{\pm0.0289}$ |
| SubgraphRAG | $\mathbf{0.5360_{\pm0.0133}}$ | $0.4550_{\pm0.0433}$ | $0.4869_{\pm0.0223}$ | $0.3237_{\pm0.0186}$ | $\mathbf{1.0000_{\pm0.0000}}$ | $\mathbf{1.0000_{\pm0.0000}}$ |
| GNN-RAG | $0.4854_{\pm0.0257}$ | $0.4717_{\pm0.0375}$ | $0.4782_{\pm0.0319}$ | $0.3164_{\pm0.0263}$ | $\mathbf{1.0000_{\pm0.0000}}$ | $0.8833_{\pm0.0577}$ |
| GALAX | $0.5046_{\pm0.0833}$ | $0.4783_{\pm0.0808}$ | $0.4909_{\pm0.0819}$ | $0.3285_{\pm0.0726}$ | $\mathbf{1.0000_{\pm0.0000}}$ | $0.8833_{\pm0.0289}$ |
| GALAX (Qwen2.5-7B) | $0.5237_{\pm0.0212}$ | $\mathbf{0.5150_{\pm0.0180}}$ | $\mathbf{0.5193_{\pm0.0196}}$ | $\mathbf{0.3509_{\pm0.0178}}$ | $0.9333_{\pm0.0577}$ | $0.8500_{\pm0.0500}$ |

**Table 29:** Model performance on STAD

| Model | Precision ↑ | Recall ↑ | F1 ↑ | Jaccard ↑ | Hit@5 ↑ | Hit@10 ↑ |
|---|---|---|---|---|---|---|
| M2T | 0.0000 | 0.0000 | 0.0000 | 0.0000 | 0.0000 | 0.0000 |
| GAT | $0.0000_{\pm0.0000}$ | $0.0000_{\pm0.0000}$ | $0.0000_{\pm0.0000}$ | $0.0000_{\pm0.0000}$ | $0.0000_{\pm0.0000}$ | $0.0000_{\pm0.0000}$ |
| L3 + Omics | $0.0000_{\pm0.0000}$ | $0.0000_{\pm0.0000}$ | $0.0000_{\pm0.0000}$ | $0.0000_{\pm0.0000}$ | $0.0000_{\pm0.0000}$ | $0.0000_{\pm0.0000}$ |
| L3 + Omics + KG | $0.0000_{\pm0.0000}$ | $0.0000_{\pm0.0000}$ | $0.0000_{\pm0.0000}$ | $0.0000_{\pm0.0000}$ | $0.0000_{\pm0.0000}$ | $0.0000_{\pm0.0000}$ |
| L3-FT(Med) + Omics | $0.0057_{\pm0.0100}$ | $0.0017_{\pm0.0029}$ | $0.0026_{\pm0.0045}$ | $0.0013_{\pm0.0023}$ | $0.0000_{\pm0.0000}$ | $0.0167_{\pm0.0289}$ |
| L3-FT(Med) + Omics + KG | $0.0000_{\pm0.0000}$ | $0.0000_{\pm0.0000}$ | $0.0000_{\pm0.0000}$ | $0.0000_{\pm0.0000}$ | $0.0000_{\pm0.0000}$ | $0.0000_{\pm0.0000}$ |
| L3-FT(CRISPR) + Omics | $0.5528_{\pm0.0311}$ | $0.5200_{\pm0.0400}$ | $0.5353_{\pm0.0325}$ | $0.3684_{\pm0.0309}$ | $0.9333_{\pm0.0577}$ | $0.9333_{\pm0.0289}$ |
| L3-FT(CRISPR) + Omics + KG | $0.5305_{\pm0.0231}$ | $0.5000_{\pm0.0427}$ | $0.5144_{\pm0.0332}$ | $0.3486_{\pm0.0322}$ | $0.9000_{\pm0.0000}$ | $0.8667_{\pm0.0289}$ |
| G-Retriever + pre-GAT | $0.5574_{\pm0.0091}$ | $0.4883_{\pm0.0029}$ | $0.5193_{\pm0.0024}$ | $0.3550_{\pm0.0043}$ | $0.9000_{\pm0.0000}$ | $0.9000_{\pm0.0000}$ |
| RoG | $0.5099_{\pm0.0498}$ | $0.4950_{\pm0.0606}$ | $0.5020_{\pm0.0556}$ | $0.3392_{\pm0.0475}$ | $0.8000_{\pm0.2646}$ | $0.7167_{\pm0.2021}$ |
| SubgraphRAG | $0.5044_{\pm0.0443}$ | $0.4950_{\pm0.0606}$ | $0.4993_{\pm0.0531}$ | $0.3353_{\pm0.0437}$ | $0.7667_{\pm0.2309}$ | $0.7667_{\pm0.2309}$ |
| GNN-RAG | $0.5387_{\pm0.0068}$ | $0.5267_{\pm0.0058}$ | $0.5325_{\pm0.0047}$ | $0.3680_{\pm0.0037}$ | $\mathbf{1.0000_{\pm0.0000}}$ | $0.8167_{\pm0.0577}$ |
| GALAX | $0.5298_{\pm0.0331}$ | $0.4983_{\pm0.0404}$ | $0.5134_{\pm0.0371}$ | $0.3465_{\pm0.0338}$ | $0.9667_{\pm0.0577}$ | $0.8667_{\pm0.0289}$ |
| GALAX (Qwen2.5-7B) | $\mathbf{0.5731_{\pm0.0132}}$ | $\mathbf{0.5883_{\pm0.0029}}$ | $\mathbf{0.5803_{\pm0.0055}}$ | $\mathbf{0.4117_{\pm0.0057}}$ | $\mathbf{1.0000_{\pm0.0000}}$ | $\mathbf{0.9833_{\pm0.0289}}$ |

**Table 30:** Model performance on MB

| Model | Precision ↑ | Recall ↑ | F1 ↑ | Jaccard ↑ | Hit@5 ↑ | Hit@10 ↑ |
|---|---|---|---|---|---|---|
| M2T | 0.0000 | 0.0000 | 0.0000 | 0.0000 | 0.0000 | 0.0000 |
| GAT | $0.0000_{\pm0.0000}$ | $0.0000_{\pm0.0000}$ | $0.0000_{\pm0.0000}$ | $0.0000_{\pm0.0000}$ | $0.0000_{\pm0.0000}$ | $0.0000_{\pm0.0000}$ |
| L3 + Omics | $0.0333_{\pm0.0577}$ | $0.0067_{\pm0.0115}$ | $0.0111_{\pm0.0192}$ | $0.0056_{\pm0.0098}$ | $0.0000_{\pm0.0000}$ | $0.0000_{\pm0.0000}$ |
| L3 + Omics + KG | $0.0126_{\pm0.0218}$ | $0.0067_{\pm0.0115}$ | $0.0087_{\pm0.0151}$ | $0.0044_{\pm0.0076}$ | $0.0000_{\pm0.0000}$ | $0.0000_{\pm0.0000}$ |
| L3-FT(Med) + Omics | $0.0177_{\pm0.0215}$ | $0.0100_{\pm0.0100}$ | $0.0102_{\pm0.0089}$ | $0.0052_{\pm0.0045}$ | $0.0000_{\pm0.0000}$ | $0.0000_{\pm0.0000}$ |
| L3-FT(Med) + Omics + KG | $0.0282_{\pm0.0237}$ | $0.0167_{\pm0.0058}$ | $0.0192_{\pm0.0090}$ | $0.0097_{\pm0.0046}$ | $0.0667_{\pm0.1155}$ | $0.0333_{\pm0.0577}$ |
| L3-FT(CRISPR) + Omics | $0.5308_{\pm0.0327}$ | $0.5033_{\pm0.0551}$ | $0.5164_{\pm0.0440}$ | $0.3488_{\pm0.0394}$ | $0.8667_{\pm0.1155}$ | $0.8000_{\pm0.1000}$ |
| L3-FT(CRISPR) + Omics + KG | $0.5086_{\pm0.0633}$ | $0.5000_{\pm0.0608}$ | $0.5043_{\pm0.0620}$ | $0.3387_{\pm0.0569}$ | $0.7333_{\pm0.2309}$ | $0.7000_{\pm0.2000}$ |
| G-Retriever + pre-GAT | $\mathbf{0.5546_{\pm0.0421}}$ | $0.3600_{\pm0.0173}$ | $0.4365_{\pm0.0257}$ | $0.2794_{\pm0.0213}$ | $0.8000_{\pm0.0000}$ | $0.7000_{\pm0.0000}$ |
| RoG | $0.5530_{\pm0.0211}$ | $\mathbf{0.5400_{\pm0.0173}}$ | $\mathbf{0.5464_{\pm0.0192}}$ | $\mathbf{0.3761_{\pm0.0183}}$ | $\mathbf{1.0000_{\pm0.0000}}$ | $0.9000_{\pm0.0000}$ |
| SubgraphRAG | $0.5545_{\pm0.0119}$ | $0.3900_{\pm0.1212}$ | $0.4502_{\pm0.0737}$ | $0.2925_{\pm0.0632}$ | $0.8667_{\pm0.1155}$ | $0.9000_{\pm0.0000}$ |
| GNN-RAG | $0.5306_{\pm0.0267}$ | $0.4467_{\pm0.1097}$ | $0.4776_{\pm0.0606}$ | $0.3151_{\pm0.0512}$ | $0.9333_{\pm0.1155}$ | $\mathbf{0.9667_{\pm0.0577}}$ |
| GALAX | $0.4897_{\pm0.0304}$ | $0.4867_{\pm0.0351}$ | $0.4882_{\pm0.0328}$ | $0.3233_{\pm0.0286}$ | $\mathbf{1.0000_{\pm0.0000}}$ | $0.8333_{\pm0.0577}$ |
| GALAX (Qwen2.5-7B) | $0.5300_{\pm0.0346}$ | $0.5300_{\pm0.0346}$ | $0.5300_{\pm0.0346}$ | $0.3611_{\pm0.0325}$ | $0.8667_{\pm0.1155}$ | $0.9333_{\pm0.0577}$ |

**Table 31:** Model performance on ALL

| Model | Precision ↑ | Recall ↑ | F1 ↑ | Jaccard ↑ | Hit@5 ↑ | Hit@10 ↑ |
|---|---|---|---|---|---|---|
| M2T | 0.0082 | 0.0050 | 0.0062 | 0.0031 | 0.0000 | 0.0500 |
| GAT | $0.0000_{\pm 0.0000}$ | $0.0000_{\pm 0.0000}$ | $0.0000_{\pm 0.0000}$ | $0.0000_{\pm 0.0000}$ | $0.0000_{\pm 0.0000}$ | $0.0000_{\pm 0.0000}$ |
| L3 + Omics | $0.0333_{\pm 0.0577}$ | $0.0033_{\pm 0.0058}$ | $0.0061_{\pm 0.0105}$ | $0.0031_{\pm 0.0053}$ | $0.0667_{\pm 0.1155}$ | $0.0333_{\pm 0.0577}$ |
| L3 + Omics + KG | $0.0000_{\pm 0.0000}$ | $0.0000_{\pm 0.0000}$ | $0.0000_{\pm 0.0000}$ | $0.0000_{\pm 0.0000}$ | $0.0000_{\pm 0.0000}$ | $0.0000_{\pm 0.0000}$ |
| L3-FT(Med) + Omics | $0.0015_{\pm 0.0026}$ | $0.0033_{\pm 0.0058}$ | $0.0021_{\pm 0.0036}$ | $0.0010_{\pm 0.0018}$ | $0.0000_{\pm 0.0000}$ | $0.0000_{\pm 0.0000}$ |
| L3-FT(Med) + Omics + KG | $0.0607_{\pm 0.0528}$ | $0.0133_{\pm 0.0115}$ | $0.0219_{\pm 0.0189}$ | $0.0111_{\pm 0.0096}$ | $0.0000_{\pm 0.0000}$ | $0.0000_{\pm 0.0000}$ |
| L3-FT(CRISPR) + Omics | $0.5487_{\pm 0.0561}$ | $0.5400_{\pm 0.0624}$ | $0.5443_{\pm 0.0592}$ | $0.3754_{\pm 0.0569}$ | $\mathbf{1.0000}_{\pm 0.0000}$ | $0.9333_{\pm 0.1155}$ |
| L3-FT(CRISPR) + Omics + KG | $0.6010_{\pm 0.0608}$ | $0.5600_{\pm 0.1100}$ | $0.5786_{\pm 0.0868}$ | $0.4106_{\pm 0.0872}$ | $0.9333_{\pm 0.1155}$ | $0.9333_{\pm 0.1155}$ |
| G-Retriever + pre-GAT | $0.6285_{\pm 0.0043}$ | $0.5300_{\pm 0.0000}$ | $0.5750_{\pm 0.0018}$ | $0.4036_{\pm 0.0018}$ | $0.8000_{\pm 0.0000}$ | $0.8000_{\pm 0.0000}$ |
| RoG | $0.5596_{\pm 0.0699}$ | $0.5133_{\pm 0.1501}$ | $0.5326_{\pm 0.1168}$ | $0.3684_{\pm 0.1042}$ | $0.9333_{\pm 0.1155}$ | $0.9667_{\pm 0.0577}$ |
| SubgraphRAG | $0.6000_{\pm 0.0000}$ | $0.6000_{\pm 0.0000}$ | $0.6000_{\pm 0.0000}$ | $0.4286_{\pm 0.0000}$ | $\mathbf{1.0000}_{\pm 0.0000}$ | $\mathbf{1.0000}_{\pm 0.0000}$ |
| GNN-RAG | $0.5682_{\pm 0.0858}$ | $0.5200_{\pm 0.1609}$ | $0.5402_{\pm 0.1293}$ | $0.3770_{\pm 0.1186}$ | $0.9333_{\pm 0.1155}$ | $0.9667_{\pm 0.0577}$ |
| GALAX | $\mathbf{0.6860}_{\pm 0.0246}$ | $\mathbf{0.6700}_{\pm 0.0265}$ | $\mathbf{0.6779}_{\pm 0.0255}$ | $\mathbf{0.5131}_{\pm 0.0295}$ | $\mathbf{1.0000}_{\pm 0.0000}$ | $0.9333_{\pm 0.0577}$ |
| GALAX (Qwen2.5-7B) | $0.6000_{\pm 0.0520}$ | $0.6000_{\pm 0.0520}$ | $0.6000_{\pm 0.0520}$ | $0.4299_{\pm 0.0542}$ | $\mathbf{1.0000}_{\pm 0.0000}$ | $\mathbf{1.0000}_{\pm 0.0000}$ |

**Table 32:** Model performance on LGG

| Model | Precision ↑ | Recall ↑ | F1 ↑ | Jaccard ↑ | Hit@5 ↑ | Hit@10 ↑ |
|---|---|---|---|---|---|---|
| M2T | 0.0000 | 0.0000 | 0.0000 | 0.0000 | 0.0000 | 0.0000 |
| GAT | $0.0000_{\pm 0.0000}$ | $0.0000_{\pm 0.0000}$ | $0.0000_{\pm 0.0000}$ | $0.0000_{\pm 0.0000}$ | $0.0000_{\pm 0.0000}$ | $0.0000_{\pm 0.0000}$ |
| L3 + Omics | $0.0000_{\pm 0.0000}$ | $0.0000_{\pm 0.0000}$ | $0.0000_{\pm 0.0000}$ | $0.0000_{\pm 0.0000}$ | $0.0000_{\pm 0.0000}$ | $0.0000_{\pm 0.0000}$ |
| L3 + Omics + KG | $0.0053_{\pm 0.0092}$ | $0.0033_{\pm 0.0058}$ | $0.0041_{\pm 0.0071}$ | $0.0021_{\pm 0.0036}$ | $0.0000_{\pm 0.0000}$ | $0.0000_{\pm 0.0000}$ |
| L3-FT(Med) + Omics | $0.0070_{\pm 0.0121}$ | $0.0133_{\pm 0.0231}$ | $0.0092_{\pm 0.0159}$ | $0.0046_{\pm 0.0080}$ | $0.0000_{\pm 0.0000}$ | $0.0000_{\pm 0.0000}$ |
| L3-FT(Med) + Omics + KG | $0.0000_{\pm 0.0000}$ | $0.0000_{\pm 0.0000}$ | $0.0000_{\pm 0.0000}$ | $0.0000_{\pm 0.0000}$ | $0.0000_{\pm 0.0000}$ | $0.0000_{\pm 0.0000}$ |
| L3-FT(CRISPR) + Omics | $0.5000_{\pm 0.0500}$ | $0.4367_{\pm 0.0702}$ | $0.4649_{\pm 0.0545}$ | $0.3040_{\pm 0.0457}$ | $\mathbf{1.0000}_{\pm 0.0000}$ | $0.9667_{\pm 0.0577}$ |
| L3-FT(CRISPR) + Omics + KG | $0.5267_{\pm 0.0250}$ | $0.5067_{\pm 0.0115}$ | $0.5163_{\pm 0.0141}$ | $0.3480_{\pm 0.0128}$ | $\mathbf{1.0000}_{\pm 0.0000}$ | $0.9000_{\pm 0.1000}$ |
| G-Retriever + pre-GAT | $0.4846_{\pm 0.0537}$ | $0.4300_{\pm 0.0173}$ | $0.4540_{\pm 0.0126}$ | $0.2937_{\pm 0.0106}$ | $\mathbf{1.0000}_{\pm 0.0000}$ | $0.9000_{\pm 0.0000}$ |
| RoG | $\mathbf{0.5667}_{\pm 0.0462}$ | $\mathbf{0.5667}_{\pm 0.0462}$ | $\mathbf{0.5667}_{\pm 0.0462}$ | $\mathbf{0.3963}_{\pm 0.0458}$ | $\mathbf{1.0000}_{\pm 0.0000}$ | $0.9667_{\pm 0.0577}$ |
| SubgraphRAG | $0.5400_{\pm 0.0000}$ | $0.5400_{\pm 0.0000}$ | $0.5400_{\pm 0.0000}$ | $0.3699_{\pm 0.0000}$ | $\mathbf{1.0000}_{\pm 0.0000}$ | $\mathbf{1.0000}_{\pm 0.0000}$ |
| GNN-RAG | $0.5294_{\pm 0.0000}$ | $0.2700_{\pm 0.0000}$ | $0.3576_{\pm 0.0000}$ | $0.2177_{\pm 0.0000}$ | $0.8000_{\pm 0.0000}$ | $0.8000_{\pm 0.0000}$ |
| GALAX | $0.5433_{\pm 0.0252}$ | $0.5433_{\pm 0.0252}$ | $0.5433_{\pm 0.0252}$ | $0.3733_{\pm 0.0238}$ | $\mathbf{1.0000}_{\pm 0.0000}$ | $0.9000_{\pm 0.0000}$ |
| GALAX (Qwen2.5-7B) | $0.5501_{\pm 0.0576}$ | $0.5200_{\pm 0.0693}$ | $0.5345_{\pm 0.0639}$ | $0.3664_{\pm 0.0581}$ | $0.8667_{\pm 0.1155}$ | $0.8667_{\pm 0.0577}$ |

**Table 33:** Model performance on NB

| Model | Precision ↑ | Recall ↑ | F1 ↑ | Jaccard ↑ | Hit@5 ↑ | Hit@10 ↑ |
|---|---|---|---|---|---|---|
| M2T | 0.0000 | 0.0000 | 0.0000 | 0.0000 | 0.0000 | 0.0000 |
| GAT | $0.0000_{\pm 0.0000}$ | $0.0000_{\pm 0.0000}$ | $0.0000_{\pm 0.0000}$ | $0.0000_{\pm 0.0000}$ | $0.0000_{\pm 0.0000}$ | $0.0000_{\pm 0.0000}$ |
| L3 + Omics | $0.0000_{\pm 0.0000}$ | $0.0000_{\pm 0.0000}$ | $0.0000_{\pm 0.0000}$ | $0.0000_{\pm 0.0000}$ | $0.0000_{\pm 0.0000}$ | $0.0000_{\pm 0.0000}$ |
| L3 + Omics + KG | $0.0000_{\pm 0.0000}$ | $0.0000_{\pm 0.0000}$ | $0.0000_{\pm 0.0000}$ | $0.0000_{\pm 0.0000}$ | $0.0000_{\pm 0.0000}$ | $0.0000_{\pm 0.0000}$ |
| L3-FT(Med) + Omics | $0.0085_{\pm 0.0148}$ | $0.0167_{\pm 0.0289}$ | $0.0113_{\pm 0.0196}$ | $0.0057_{\pm 0.0100}$ | $0.0000_{\pm 0.0000}$ | $0.0000_{\pm 0.0000}$ |
| L3-FT(Med) + Omics + KG | $0.0000_{\pm 0.0000}$ | $0.0000_{\pm 0.0000}$ | $0.0000_{\pm 0.0000}$ | $0.0000_{\pm 0.0000}$ | $0.0000_{\pm 0.0000}$ | $0.0000_{\pm 0.0000}$ |
| L3-FT(CRISPR) + Omics | $0.5597_{\pm 0.0479}$ | $0.5233_{\pm 0.0651}$ | $0.5407_{\pm 0.0559}$ | $0.3718_{\pm 0.0523}$ | $0.8667_{\pm 0.2309}$ | $0.8667_{\pm 0.1528}$ |
| L3-FT(CRISPR) + Omics + KG | $0.5507_{\pm 0.0704}$ | $0.5067_{\pm 0.0723}$ | $0.5275_{\pm 0.0703}$ | $0.3603_{\pm 0.0667}$ | $0.9333_{\pm 0.1155}$ | $0.8667_{\pm 0.0577}$ |
| G-Retriever + pre-GAT | $0.4322_{\pm 0.0444}$ | $0.3933_{\pm 0.0404}$ | $0.4119_{\pm 0.0423}$ | $0.2599_{\pm 0.0341}$ | $\mathbf{1.0000}_{\pm 0.0000}$ | $\mathbf{0.9333}_{\pm 0.0577}$ |
| RoG | $\mathbf{0.5997}_{\pm 0.0551}$ | $0.4133_{\pm 0.0924}$ | $0.4817_{\pm 0.0400}$ | $0.3179_{\pm 0.0353}$ | $\mathbf{1.0000}_{\pm 0.0000}$ | $0.8667_{\pm 0.1155}$ |
| SubgraphRAG | $0.5679_{\pm 0.0551}$ | $0.4667_{\pm 0.0924}$ | $0.5048_{\pm 0.0400}$ | $0.3383_{\pm 0.0353}$ | $\mathbf{1.0000}_{\pm 0.0000}$ | $\mathbf{0.9333}_{\pm 0.1155}$ |
| GNN-RAG | $0.5810_{\pm 0.0165}$ | $\mathbf{0.5733}_{\pm 0.0231}$ | $\mathbf{0.5771}_{\pm 0.0198}$ | $\mathbf{0.4058}_{\pm 0.0197}$ | $0.7333_{\pm 0.2309}$ | $0.7333_{\pm 0.0577}$ |
| GALAX | $0.5533_{\pm 0.0115}$ | $0.5533_{\pm 0.0115}$ | $0.5533_{\pm 0.0115}$ | $0.3825_{\pm 0.0110}$ | $0.8667_{\pm 0.1155}$ | $0.8667_{\pm 0.0577}$ |
| GALAX (Qwen2.5-7B) | $0.5733_{\pm 0.0462}$ | $\mathbf{0.5733}_{\pm 0.0462}$ | $0.5733_{\pm 0.0462}$ | $0.4028_{\pm 0.0446}$ | $0.8000_{\pm 0.0000}$ | $0.8667_{\pm 0.0577}$ |

**Table 34:** Model performance on MESO

| Model | Precision ↑ | Recall ↑ | F1 ↑ | Jaccard ↑ | Hit@5 ↑ | Hit@10 ↑ |
|---|---|---|---|---|---|---|
| M2T | 0.0070 | 0.0050 | 0.0058 | 0.0029 | 0.0000 | 0.0000 |
| GAT | $0.0000_{\pm 0.0000}$ | $0.0000_{\pm 0.0000}$ | $0.0000_{\pm 0.0000}$ | $0.0000_{\pm 0.0000}$ | $0.0000_{\pm 0.0000}$ | $0.0000_{\pm 0.0000}$ |
| L3 + Omics | $0.0000_{\pm 0.0000}$ | $0.0000_{\pm 0.0000}$ | $0.0000_{\pm 0.0000}$ | $0.0000_{\pm 0.0000}$ | $0.0000_{\pm 0.0000}$ | $0.0000_{\pm 0.0000}$ |
| L3 + Omics + KG | $0.0000_{\pm 0.0000}$ | $0.0000_{\pm 0.0000}$ | $0.0000_{\pm 0.0000}$ | $0.0000_{\pm 0.0000}$ | $0.0000_{\pm 0.0000}$ | $0.0000_{\pm 0.0000}$ |
| L3-FT(Med) + Omics | $0.0190_{\pm 0.0201}$ | $0.0167_{\pm 0.0208}$ | $0.0133_{\pm 0.0121}$ | $0.0067_{\pm 0.0061}$ | $0.0000_{\pm 0.0000}$ | $0.0000_{\pm 0.0000}$ |
| L3-FT(Med) + Omics + KG | $0.0192_{\pm 0.0228}$ | $0.0100_{\pm 0.0100}$ | $0.0104_{\pm 0.0052}$ | $0.0053_{\pm 0.0026}$ | $0.0000_{\pm 0.0000}$ | $0.0000_{\pm 0.0000}$ |
| L3-FT(CRISPR) + Omics | $0.5182_{\pm 0.0806}$ | $0.4833_{\pm 0.0874}$ | $0.4995_{\pm 0.0808}$ | $0.3354_{\pm 0.0720}$ | $0.9333_{\pm 0.1155}$ | $0.8667_{\pm 0.1528}$ |
| L3-FT(CRISPR) + Omics + KG | $0.5417_{\pm 0.0419}$ | $0.5233_{\pm 0.0416}$ | $0.5322_{\pm 0.0402}$ | $0.3633_{\pm 0.0374}$ | $0.8667_{\pm 0.1155}$ | $0.9000_{\pm 0.0000}$ |
| G-Retriever + pre-GAT | $0.4721_{\pm 0.0072}$ | $0.3733_{\pm 0.0462}$ | $0.4161_{\pm 0.0324}$ | $0.2630_{\pm 0.0255}$ | $\mathbf{1.0000}_{\pm 0.0000}$ | $0.8000_{\pm 0.0000}$ |
| RoG | $0.4476_{\pm 0.0755}$ | $0.3900_{\pm 0.1375}$ | $0.4137_{\pm 0.1135}$ | $0.2651_{\pm 0.0893}$ | $0.8667_{\pm 0.2309}$ | $0.8000_{\pm 0.0000}$ |
| SubgraphRAG | $0.4421_{\pm 0.0000}$ | $0.4200_{\pm 0.0000}$ | $0.4308_{\pm 0.0000}$ | $0.2745_{\pm 0.0000}$ | $0.6000_{\pm 0.0000}$ | $0.8000_{\pm 0.0000}$ |
| GNN-RAG | $0.3750_{\pm 0.0000}$ | $0.2400_{\pm 0.0000}$ | $0.2927_{\pm 0.0000}$ | $0.1714_{\pm 0.0000}$ | $\mathbf{1.0000}_{\pm 0.0000}$ | $0.8000_{\pm 0.0000}$ |
| GALAX | $\mathbf{0.5933}_{\pm 0.0231}$ | $\mathbf{0.5933}_{\pm 0.0231}$ | $\mathbf{0.5933}_{\pm 0.0231}$ | $\mathbf{0.4221}_{\pm 0.0236}$ | $\mathbf{1.0000}_{\pm 0.0000}$ | $0.8000_{\pm 0.0000}$ |
| GALAX (Qwen2.5-7B) | $0.5621_{\pm 0.0310}$ | $0.5533_{\pm 0.0462}$ | $0.5576_{\pm 0.0388}$ | $0.3872_{\pm 0.0367}$ | $\mathbf{1.0000}_{\pm 0.0000}$ | $0.9333_{\pm 0.0577}$ |

**Table 35:** Model performance on LIHC

| Model | Precision ↑ | Recall ↑ | F1 ↑ | Jaccard ↑ | Hit@5 ↑ | Hit@10 ↑ |
|---|---|---|---|---|---|---|
| M2T | 0.0000 | 0.0000 | 0.0000 | 0.0000 | 0.0000 | 0.0000 |
| GAT | $0.0100_{\pm0.0000}$ | $0.0100_{\pm0.0000}$ | $0.0100_{\pm0.0000}$ | $0.0050_{\pm0.0000}$ | $0.0000_{\pm0.0000}$ | $0.0000_{\pm0.0000}$ |
| L3 + Omics | $0.0000_{\pm0.0000}$ | $0.0000_{\pm0.0000}$ | $0.0000_{\pm0.0000}$ | $0.0000_{\pm0.0000}$ | $0.0000_{\pm0.0000}$ | $0.0000_{\pm0.0000}$ |
| L3 + Omics + KG | $0.0000_{\pm0.0000}$ | $0.0000_{\pm0.0000}$ | $0.0000_{\pm0.0000}$ | $0.0000_{\pm0.0000}$ | $0.0000_{\pm0.0000}$ | $0.0000_{\pm0.0000}$ |
| L3-FT(Med) + Omics | $0.0031_{\pm0.0027}$ | $0.0067_{\pm0.0058}$ | $0.0043_{\pm0.0037}$ | $0.0021_{\pm0.0019}$ | $0.0000_{\pm0.0000}$ | $0.0000_{\pm0.0000}$ |
| L3-FT(Med) + Omics + KG | $0.0000_{\pm0.0000}$ | $0.0000_{\pm0.0000}$ | $0.0000_{\pm0.0000}$ | $0.0000_{\pm0.0000}$ | $0.0000_{\pm0.0000}$ | $0.0000_{\pm0.0000}$ |
| L3-FT(CRISPR) + Omics | $0.4239_{\pm0.0343}$ | $0.3967_{\pm0.0513}$ | $0.4096_{\pm0.0435}$ | $0.2582_{\pm0.0341}$ | $0.7333_{\pm0.1155}$ | $0.7000_{\pm0.1732}$ |
| L3-FT(CRISPR) + Omics + KG | $0.4552_{\pm0.0568}$ | $0.4467_{\pm0.0635}$ | $0.4509_{\pm0.0600}$ | $0.2924_{\pm0.0512}$ | $0.8000_{\pm0.0000}$ | $0.7667_{\pm0.0577}$ |
| G-Retriever + pre-GAT | $0.3598_{\pm0.0387}$ | $0.3000_{\pm0.0520}$ | $0.3270_{\pm0.0467}$ | $0.1961_{\pm0.0340}$ | $0.8000_{\pm0.0000}$ | $0.7667_{\pm0.1155}$ |
| RoG | $0.4205_{\pm0.1173}$ | $0.3733_{\pm0.1856}$ | $0.3903_{\pm0.1596}$ | $0.2507_{\pm0.1247}$ | $0.9333_{\pm0.1155}$ | $0.8000_{\pm0.1732}$ |
| SubgraphRAG | $0.4738_{\pm0.1320}$ | $0.4267_{\pm0.2136}$ | $0.4436_{\pm0.1843}$ | $0.2964_{\pm0.1437}$ | $0.9333_{\pm0.1155}$ | $0.8000_{\pm0.1732}$ |
| GNN-RAG | $0.4233_{\pm0.0289}$ | $0.4233_{\pm0.0289}$ | $0.4233_{\pm0.0289}$ | $0.2688_{\pm0.0230}$ | $1.0000_{\pm0.0000}$ | $0.8333_{\pm0.0577}$ |
| GALAX | $\mathbf{0.4900_{\pm0.0100}}$ | $\mathbf{0.4900_{\pm0.0100}}$ | $\mathbf{0.4900_{\pm0.0100}}$ | $0.3245_{\pm0.0088}$ | $1.0000_{\pm0.0000}$ | $\mathbf{0.9000_{\pm0.0000}}$ |
| GALAX (Qwen2.5-7B) | $0.4362_{\pm0.0033}$ | $0.4333_{\pm0.0058}$ | $0.4348_{\pm0.0045}$ | $0.2778_{\pm0.0037}$ | $0.8667_{\pm0.1155}$ | $0.8667_{\pm0.0577}$ |

**Table 36:** Model performance on LAML

| Model | Precision ↑ | Recall ↑ | F1 ↑ | Jaccard ↑ | Hit@5 ↑ | Hit@10 ↑ |
|---|---|---|---|---|---|---|
| M2T | 0.0164 | 0.0100 | 0.0124 | 0.0062 | 0.0000 | 0.0000 |
| GAT | $0.0000_{\pm0.0000}$ | $0.0000_{\pm0.0000}$ | $0.0000_{\pm0.0000}$ | $0.0000_{\pm0.0000}$ | $0.0000_{\pm0.0000}$ | $0.0000_{\pm0.0000}$ |
| L3 + Omics | $0.0000_{\pm0.0000}$ | $0.0000_{\pm0.0000}$ | $0.0000_{\pm0.0000}$ | $0.0000_{\pm0.0000}$ | $0.0000_{\pm0.0000}$ | $0.0000_{\pm0.0000}$ |
| L3 + Omics + KG | $0.1576_{\pm0.1412}$ | $0.0200_{\pm0.0173}$ | $0.0354_{\pm0.0307}$ | $0.0182_{\pm0.0158}$ | $0.1333_{\pm0.2309}$ | $0.2000_{\pm0.1732}$ |
| L3-FT(Med) + Omics | $0.1136_{\pm0.0630}$ | $0.0400_{\pm0.0173}$ | $0.0495_{\pm0.0008}$ | $0.0254_{\pm0.0004}$ | $0.2000_{\pm0.3464}$ | $0.1333_{\pm0.1528}$ |
| L3-FT(Med) + Omics + KG | $0.1085_{\pm0.0867}$ | $0.0267_{\pm0.0058}$ | $0.0377_{\pm0.0221}$ | $0.0193_{\pm0.0114}$ | $0.2000_{\pm0.3464}$ | $0.2000_{\pm0.1732}$ |
| L3-FT(CRISPR) + Omics | $0.4758_{\pm0.0251}$ | $0.3733_{\pm0.0115}$ | $0.4180_{\pm0.0082}$ | $0.2642_{\pm0.0066}$ | $0.8667_{\pm0.2309}$ | $0.9000_{\pm0.1000}$ |
| L3-FT(CRISPR) + Omics + KG | $\mathbf{0.4988_{\pm0.0700}}$ | $0.4500_{\pm0.0600}$ | $\mathbf{0.4721_{\pm0.0572}}$ | $\mathbf{0.3102_{\pm0.0481}}$ | $0.8667_{\pm0.2309}$ | $0.8333_{\pm0.1155}$ |
| G-Retriever + pre-GAT | $0.3719_{\pm0.0373}$ | $0.2400_{\pm0.0100}$ | $0.2912_{\pm0.0100}$ | $0.1705_{\pm0.0081}$ | $1.0000_{\pm0.0000}$ | $0.9000_{\pm0.0000}$ |
| RoG | $0.4228_{\pm0.0149}$ | $0.3467_{\pm0.1097}$ | $0.3725_{\pm0.0685}$ | $0.2303_{\pm0.0506}$ | $1.0000_{\pm0.0000}$ | $0.8667_{\pm0.1155}$ |
| SubgraphRAG | $0.4141_{\pm0.0000}$ | $0.4100_{\pm0.0000}$ | $0.4121_{\pm0.0000}$ | $0.2595_{\pm0.0000}$ | $1.0000_{\pm0.0000}$ | $0.8000_{\pm0.0000}$ |
| GNN-RAG | $0.3972_{\pm0.0934}$ | $0.3200_{\pm0.1559}$ | $0.3511_{\pm0.1311}$ | $0.2183_{\pm0.1015}$ | $1.0000_{\pm0.0000}$ | $0.8333_{\pm0.0577}$ |
| GALAX | $0.4730_{\pm0.0901}$ | $0.4333_{\pm0.0306}$ | $0.4510_{\pm0.0554}$ | $0.2923_{\pm0.0467}$ | $0.9333_{\pm0.1155}$ | $0.8667_{\pm0.2309}$ |
| GALAX (Qwen2.5-7B) | $0.4596_{\pm0.0437}$ | $\mathbf{0.4567_{\pm0.0462}}$ | $0.4581_{\pm0.0449}$ | $0.2979_{\pm0.0385}$ | $1.0000_{\pm0.0000}$ | $1.0000_{\pm0.0000}$ |

**Table 37:** Model performance on DLBC

| Model | Precision ↑ | Recall ↑ | F1 ↑ | Jaccard ↑ | Hit@5 ↑ | Hit@10 ↑ |
|---|---|---|---|---|---|---|
| M2T | 0.0000 | 0.0000 | 0.0000 | 0.0000 | 0.0000 | 0.0000 |
| GAT | $0.0000_{\pm0.0000}$ | $0.0000_{\pm0.0000}$ | $0.0000_{\pm0.0000}$ | $0.0000_{\pm0.0000}$ | $0.0000_{\pm0.0000}$ | $0.0000_{\pm0.0000}$ |
| L3 + Omics | $0.0000_{\pm0.0000}$ | $0.0000_{\pm0.0000}$ | $0.0000_{\pm0.0000}$ | $0.0000_{\pm0.0000}$ | $0.0000_{\pm0.0000}$ | $0.0000_{\pm0.0000}$ |
| L3 + Omics + KG | $0.0000_{\pm0.0000}$ | $0.0000_{\pm0.0000}$ | $0.0000_{\pm0.0000}$ | $0.0000_{\pm0.0000}$ | $0.0000_{\pm0.0000}$ | $0.0000_{\pm0.0000}$ |
| L3-FT(Med) + Omics | $0.0220_{\pm0.0249}$ | $0.0367_{\pm0.0404}$ | $0.0275_{\pm0.0308}$ | $0.0141_{\pm0.0159}$ | $0.0000_{\pm0.0000}$ | $0.0000_{\pm0.0000}$ |
| L3-FT(Med) + Omics + KG | $0.0375_{\pm0.0385}$ | $0.0067_{\pm0.0058}$ | $0.0111_{\pm0.0097}$ | $0.0056_{\pm0.0049}$ | $0.0667_{\pm0.1155}$ | $0.0333_{\pm0.0577}$ |
| L3-FT(CRISPR) + Omics | $0.3985_{\pm0.0669}$ | $0.3900_{\pm0.0721}$ | $0.3942_{\pm0.0695}$ | $0.2470_{\pm0.0530}$ | $0.9333_{\pm0.1155}$ | $0.8333_{\pm0.0577}$ |
| L3-FT(CRISPR) + Omics + KG | $0.4200_{\pm0.0230}$ | $0.4100_{\pm0.0173}$ | $0.4149_{\pm0.0200}$ | $0.2619_{\pm0.0158}$ | $0.8667_{\pm0.1155}$ | $0.7000_{\pm0.1000}$ |
| G-Retriever + pre-GAT | $0.3196_{\pm0.0307}$ | $0.2767_{\pm0.0058}$ | $0.2963_{\pm0.0168}$ | $0.1740_{\pm0.0115}$ | $1.0000_{\pm0.0000}$ | $0.8000_{\pm0.0000}$ |
| RoG | $0.4424_{\pm0.0239}$ | $0.4367_{\pm0.0289}$ | $0.4395_{\pm0.0264}$ | $0.2819_{\pm0.0219}$ | $0.8667_{\pm0.1155}$ | $0.7667_{\pm0.1155}$ |
| SubgraphRAG | $0.4286_{\pm0.0000}$ | $0.4200_{\pm0.0000}$ | $0.4242_{\pm0.0000}$ | $0.2692_{\pm0.0000}$ | $0.8000_{\pm0.0000}$ | $0.7000_{\pm0.0000}$ |
| GNN-RAG | $0.4433_{\pm0.0231}$ | $0.4433_{\pm0.0231}$ | $0.4433_{\pm0.0231}$ | $0.2850_{\pm0.0192}$ | $1.0000_{\pm0.0000}$ | $0.9000_{\pm0.0000}$ |
| GALAX | $0.4400_{\pm0.0000}$ | $0.4400_{\pm0.0000}$ | $0.4400_{\pm0.0000}$ | $0.2821_{\pm0.0000}$ | $0.8667_{\pm0.1155}$ | $0.6333_{\pm0.2309}$ |
| GALAX (Qwen2.5-7B) | $\mathbf{0.4764_{\pm0.0055}}$ | $\mathbf{0.4700_{\pm0.0000}}$ | $\mathbf{0.4732_{\pm0.0027}}$ | $\mathbf{0.3099_{\pm0.0023}}$ | $0.8000_{\pm0.0000}$ | $0.9000_{\pm0.0000}$ |

**Table 38:** Model performance on MM

| Model | Precision ↑ | Recall ↑ | F1 ↑ | Jaccard ↑ | Hit@5 ↑ | Hit@10 ↑ |
|---|---|---|---|---|---|---|
| M2T | 0.0000 | 0.0000 | 0.0000 | 0.0000 | 0.0000 | 0.0000 |
| GAT | $0.0000_{\pm0.0000}$ | $0.0000_{\pm0.0000}$ | $0.0000_{\pm0.0000}$ | $0.0000_{\pm0.0000}$ | $0.0000_{\pm0.0000}$ | $0.0000_{\pm0.0000}$ |
| L3 + Omics | $0.0000_{\pm0.0000}$ | $0.0000_{\pm0.0000}$ | $0.0000_{\pm0.0000}$ | $0.0000_{\pm0.0000}$ | $0.0000_{\pm0.0000}$ | $0.0000_{\pm0.0000}$ |
| L3 + Omics + KG | $0.0000_{\pm0.0000}$ | $0.0000_{\pm0.0000}$ | $0.0000_{\pm0.0000}$ | $0.0000_{\pm0.0000}$ | $0.0000_{\pm0.0000}$ | $0.0000_{\pm0.0000}$ |
| L3-FT(Med) + Omics | $0.0275_{\pm0.0296}$ | $0.0167_{\pm0.0208}$ | $0.0156_{\pm0.0149}$ | $0.0079_{\pm0.0075}$ | $0.0667_{\pm0.1155}$ | $0.0333_{\pm0.0577}$ |
| L3-FT(Med) + Omics + KG | $0.1091_{\pm0.0630}$ | $0.0200_{\pm0.0173}$ | $0.0329_{\pm0.0027}$ | $0.0167_{\pm0.0014}$ | $0.1333_{\pm0.1155}$ | $0.1333_{\pm0.1155}$ |
| L3-FT(CRISPR) + Omics | $0.4001_{\pm0.0022}$ | $0.3800_{\pm0.0265}$ | $0.3894_{\pm0.0134}$ | $0.2418_{\pm0.0103}$ | $0.9333_{\pm0.1155}$ | $0.9000_{\pm0.0000}$ |
| L3-FT(CRISPR) + Omics + KG | $0.4753_{\pm0.0613}$ | $0.4367_{\pm0.0306}$ | $0.4549_{\pm0.0447}$ | $0.2952_{\pm0.0377}$ | $0.8000_{\pm0.0000}$ | $0.7333_{\pm0.0577}$ |
| G-Retriever + pre-GAT | $0.4514_{\pm0.0890}$ | $0.3667_{\pm0.0808}$ | $0.4046_{\pm0.0850}$ | $0.2561_{\pm0.0690}$ | $0.9333_{\pm0.1155}$ | $0.9000_{\pm0.0000}$ |
| RoG | $0.4900_{\pm0.0615}$ | $0.3833_{\pm0.1550}$ | $0.4246_{\pm0.1182}$ | $0.2745_{\pm0.0997}$ | $0.9333_{\pm0.1155}$ | $0.8667_{\pm0.0577}$ |
| SubgraphRAG | $0.4515_{\pm0.0122}$ | $0.3033_{\pm0.0289}$ | $0.3620_{\pm0.0175}$ | $0.2211_{\pm0.0130}$ | $0.9333_{\pm0.1155}$ | $0.8667_{\pm0.0577}$ |
| GNN-RAG | $0.5166_{\pm0.0376}$ | $0.5133_{\pm0.0404}$ | $0.5150_{\pm0.0390}$ | $0.3474_{\pm0.0359}$ | $1.0000_{\pm0.0000}$ | $0.8333_{\pm0.0577}$ |
| GALAX | $\mathbf{0.5858_{\pm0.0564}}$ | $\mathbf{0.5700_{\pm0.0819}}$ | $\mathbf{0.5775_{\pm0.0695}}$ | $\mathbf{0.4082_{\pm0.0680}}$ | $1.0000_{\pm0.0000}$ | $\mathbf{0.9333_{\pm0.0577}}$ |
| GALAX (Qwen2.5-7B) | $0.5000_{\pm0.0917}$ | $0.5000_{\pm0.0917}$ | $0.5000_{\pm0.0917}$ | $0.3367_{\pm0.0834}$ | $0.9333_{\pm0.1155}$ | $0.9000_{\pm0.1000}$ |

**Table 39:** Model performance on KIRC

| Model | Precision ↑ | Recall ↑ | F1 ↑ | Jaccard ↑ | Hit@5 ↑ | Hit@10 ↑ |
|---|---|---|---|---|---|---|
| M2T | 0.0000 | 0.0000 | 0.0000 | 0.0000 | 0.0000 | 0.0000 |
| GAT | $0.0000_{\pm 0.0000}$ | $0.0000_{\pm 0.0000}$ | $0.0000_{\pm 0.0000}$ | $0.0000_{\pm 0.0000}$ | $0.0000_{\pm 0.0000}$ | $0.0000_{\pm 0.0000}$ |
| L3 + Omics | $0.0000_{\pm 0.0000}$ | $0.0000_{\pm 0.0000}$ | $0.0000_{\pm 0.0000}$ | $0.0000_{\pm 0.0000}$ | $0.0000_{\pm 0.0000}$ | $0.0000_{\pm 0.0000}$ |
| L3 + Omics + KG | $0.0000_{\pm 0.0000}$ | $0.0000_{\pm 0.0000}$ | $0.0000_{\pm 0.0000}$ | $0.0000_{\pm 0.0000}$ | $0.0000_{\pm 0.0000}$ | $0.0000_{\pm 0.0000}$ |
| L3-FT(Med) + Omics | $0.0000_{\pm 0.0000}$ | $0.0000_{\pm 0.0000}$ | $0.0000_{\pm 0.0000}$ | $0.0000_{\pm 0.0000}$ | $0.0000_{\pm 0.0000}$ | $0.0000_{\pm 0.0000}$ |
| L3-FT(Med) + Omics + KG | $0.0000_{\pm 0.0000}$ | $0.0000_{\pm 0.0000}$ | $0.0000_{\pm 0.0000}$ | $0.0000_{\pm 0.0000}$ | $0.0000_{\pm 0.0000}$ | $0.0000_{\pm 0.0000}$ |
| L3-FT(CRISPR) + Omics | $0.5458_{\pm 0.0450}$ | $0.5333_{\pm 0.0513}$ | $0.5394_{\pm 0.0478}$ | $0.3703_{\pm 0.0452}$ | $0.8667_{\pm 0.2309}$ | $0.8667_{\pm 0.1155}$ |
| L3-FT(CRISPR) + Omics + KG | $0.4897_{\pm 0.0182}$ | $0.4800_{\pm 0.0265}$ | $0.4847_{\pm 0.0219}$ | $0.3201_{\pm 0.0192}$ | $0.8667_{\pm 0.1155}$ | $0.9000_{\pm 0.0000}$ |
| G-Retriever + pre-GAT | $0.4160_{\pm 0.0315}$ | $0.3733_{\pm 0.0058}$ | $0.3933_{\pm 0.0171}$ | $0.2449_{\pm 0.0133}$ | $0.6000_{\pm 0.0000}$ | $0.8000_{\pm 0.0000}$ |
| RoG | $0.6207_{\pm 0.0617}$ | $\mathbf{0.5867}_{\pm \mathbf{0.0751}}$ | $\mathbf{0.6031}_{\pm \mathbf{0.0689}}$ | $\mathbf{0.4340}_{\pm \mathbf{0.0687}}$ | $\mathbf{1.0000}_{\pm \mathbf{0.0000}}$ | $\mathbf{1.0000}_{\pm \mathbf{0.0000}}$ |
| SubgraphRAG | $\mathbf{0.6412}_{\pm \mathbf{0.0261}}$ | $0.5300_{\pm 0.1237}$ | $0.5714_{\pm 0.1237}$ | $0.4067_{\pm 0.1160}$ | $\mathbf{1.0000}_{\pm \mathbf{0.0000}}$ | $0.9333_{\pm 0.1155}$ |
| GNN-RAG | $0.5898_{\pm 0.0350}$ | $0.5733_{\pm 0.0635}$ | $0.5812_{\pm 0.0499}$ | $0.4108_{\pm 0.0486}$ | $\mathbf{1.0000}_{\pm \mathbf{0.0000}}$ | $\mathbf{1.0000}_{\pm \mathbf{0.0000}}$ |
| GALAX | $0.5797_{\pm 0.0251}$ | $0.5700_{\pm 0.0265}$ | $0.5748_{\pm 0.0254}$ | $0.4036_{\pm 0.0247}$ | $0.9333_{\pm 0.1155}$ | $0.9000_{\pm 0.1732}$ |
| GALAX (Qwen2.5-7B) | $0.5786_{\pm 0.1114}$ | $0.5700_{\pm 0.1044}$ | $0.5742_{\pm 0.1076}$ | $0.4079_{\pm 0.1019}$ | $0.8667_{\pm 0.1155}$ | $0.9333_{\pm 0.0577}$ |

**Table 40:** Model performance on THCA

| Model | Precision ↑ | Recall ↑ | F1 ↑ | Jaccard ↑ | Hit@5 ↑ | Hit@10 ↑ |
|---|---|---|---|---|---|---|
| M2T | 0.0000 | 0.0000 | 0.0000 | 0.0000 | 0.0000 | 0.0000 |
| GAT | $0.0000_{\pm 0.0000}$ | $0.0000_{\pm 0.0000}$ | $0.0000_{\pm 0.0000}$ | $0.0000_{\pm 0.0000}$ | $0.0000_{\pm 0.0000}$ | $0.0000_{\pm 0.0000}$ |
| L3 + Omics | $0.0000_{\pm 0.0000}$ | $0.0000_{\pm 0.0000}$ | $0.0000_{\pm 0.0000}$ | $0.0000_{\pm 0.0000}$ | $0.0000_{\pm 0.0000}$ | $0.0000_{\pm 0.0000}$ |
| L3 + Omics + KG | $0.1500_{\pm 0.1500}$ | $0.0200_{\pm 0.0173}$ | $0.0348_{\pm 0.0303}$ | $0.0179_{\pm 0.0155}$ | $0.1333_{\pm 0.2309}$ | $0.1000_{\pm 0.1732}$ |
| L3-FT(Med) + Omics | $0.0439_{\pm 0.0532}$ | $0.0167_{\pm 0.0058}$ | $0.0200_{\pm 0.0119}$ | $0.0101_{\pm 0.0061}$ | $0.0667_{\pm 0.1155}$ | $0.0333_{\pm 0.0577}$ |
| L3-FT(Med) + Omics + KG | $0.0156_{\pm 0.0145}$ | $0.0300_{\pm 0.0265}$ | $0.0205_{\pm 0.0187}$ | $0.0104_{\pm 0.0095}$ | $0.0000_{\pm 0.0000}$ | $0.0333_{\pm 0.0577}$ |
| L3-FT(CRISPR) + Omics | $0.5495_{\pm 0.0165}$ | $0.5367_{\pm 0.0153}$ | $0.5430_{\pm 0.0158}$ | $0.3728_{\pm 0.0148}$ | $0.8667_{\pm 0.1155}$ | $0.8333_{\pm 0.0577}$ |
| L3-FT(CRISPR) + Omics + KG | $0.5818_{\pm 0.0490}$ | $0.5333_{\pm 0.0416}$ | $0.5556_{\pm 0.0365}$ | $0.3853_{\pm 0.0353}$ | $0.9333_{\pm 0.1155}$ | $0.9000_{\pm 0.1000}$ |
| G-Retriever + pre-GAT | $0.6237_{\pm 0.0849}$ | $0.3833_{\pm 0.0231}$ | $0.4718_{\pm 0.0097}$ | $0.3088_{\pm 0.0083}$ | $0.5333_{\pm 0.1155}$ | $0.7333_{\pm 0.1155}$ |
| RoG | $0.5696_{\pm 0.0350}$ | $0.5367_{\pm 0.0635}$ | $0.5524_{\pm 0.0499}$ | $0.3827_{\pm 0.0486}$ | $\mathbf{1.0000}_{\pm \mathbf{0.0000}}$ | $\mathbf{1.0000}_{\pm \mathbf{0.0000}}$ |
| SubgraphRAG | $0.5495_{\pm 0.0000}$ | $0.5000_{\pm 0.0000}$ | $0.5236_{\pm 0.0000}$ | $0.3546_{\pm 0.0000}$ | $\mathbf{1.0000}_{\pm \mathbf{0.0000}}$ | $\mathbf{1.0000}_{\pm \mathbf{0.0000}}$ |
| GNN-RAG | $\mathbf{0.6465}_{\pm \mathbf{0.0350}}$ | $0.4400_{\pm 0.1386}$ | $0.5127_{\pm 0.0782}$ | $0.3473_{\pm 0.0731}$ | $0.9333_{\pm 0.1155}$ | $0.8333_{\pm 0.0577}$ |
| GALAX | $0.6076_{\pm 0.0331}$ | $\mathbf{0.6033}_{\pm \mathbf{0.0289}}$ | $\mathbf{0.6054}_{\pm \mathbf{0.0308}}$ | $\mathbf{0.4346}_{\pm \mathbf{0.0313}}$ | $\mathbf{1.0000}_{\pm \mathbf{0.0000}}$ | $\mathbf{1.0000}_{\pm \mathbf{0.0000}}$ |
| GALAX (Qwen2.5-7B) | $0.5790_{\pm 0.0350}$ | $0.4967_{\pm 0.0751}$ | $0.5312_{\pm 0.0356}$ | $0.3622_{\pm 0.0333}$ | $0.8000_{\pm 0.2000}$ | $0.8333_{\pm 0.0577}$ |

**Table 41:** Model performance on BLCA

| Model | Precision ↑ | Recall ↑ | F1 ↑ | Jaccard ↑ | Hit@5 ↑ | Hit@10 ↑ |
|---|---|---|---|---|---|---|
| M2T | 0.0000 | 0.0000 | 0.0000 | 0.0000 | 0.0000 | 0.0000 |
| GAT | $0.0000_{\pm 0.0000}$ | $0.0000_{\pm 0.0000}$ | $0.0000_{\pm 0.0000}$ | $0.0000_{\pm 0.0000}$ | $0.0000_{\pm 0.0000}$ | $0.0000_{\pm 0.0000}$ |
| L3 + Omics | $0.0000_{\pm 0.0000}$ | $0.0000_{\pm 0.0000}$ | $0.0000_{\pm 0.0000}$ | $0.0000_{\pm 0.0000}$ | $0.0000_{\pm 0.0000}$ | $0.0000_{\pm 0.0000}$ |
| L3 + Omics + KG | $0.0000_{\pm 0.0000}$ | $0.0000_{\pm 0.0000}$ | $0.0000_{\pm 0.0000}$ | $0.0000_{\pm 0.0000}$ | $0.0000_{\pm 0.0000}$ | $0.0000_{\pm 0.0000}$ |
| L3-FT(Med) + Omics | $0.0127_{\pm 0.0111}$ | $0.0267_{\pm 0.0231}$ | $0.0172_{\pm 0.0149}$ | $0.0087_{\pm 0.0076}$ | $0.0000_{\pm 0.0000}$ | $0.0000_{\pm 0.0000}$ |
| L3-FT(Med) + Omics + KG | $0.0222_{\pm 0.0385}$ | $0.0033_{\pm 0.0058}$ | $0.0058_{\pm 0.0100}$ | $0.0029_{\pm 0.0051}$ | $0.0000_{\pm 0.0000}$ | $0.0333_{\pm 0.0577}$ |
| L3-FT(CRISPR) + Omics | $0.5538_{\pm 0.0498}$ | $0.5000_{\pm 0.0200}$ | $0.5250_{\pm 0.0301}$ | $0.3563_{\pm 0.0279}$ | $0.7333_{\pm 0.1155}$ | $0.8000_{\pm 0.0000}$ |
| L3-FT(CRISPR) + Omics + KG | $0.5617_{\pm 0.0584}$ | $0.5367_{\pm 0.0231}$ | $0.5484_{\pm 0.0376}$ | $0.3784_{\pm 0.0356}$ | $0.8667_{\pm 0.2309}$ | $0.8667_{\pm 0.0577}$ |
| G-Retriever + pre-GAT | $0.5025_{\pm 0.0481}$ | $0.4733_{\pm 0.0058}$ | $0.4870_{\pm 0.0253}$ | $0.3221_{\pm 0.0223}$ | $0.6000_{\pm 0.0000}$ | $0.6333_{\pm 0.0577}$ |
| RoG | $0.5633_{\pm 0.0231}$ | $0.5633_{\pm 0.0231}$ | $0.5633_{\pm 0.0231}$ | $0.3924_{\pm 0.0226}$ | $\mathbf{1.0000}_{\pm \mathbf{0.0000}}$ | $\mathbf{0.9667}_{\pm \mathbf{0.0577}}$ |
| SubgraphRAG | $0.5457_{\pm 0.0075}$ | $0.4633_{\pm 0.1501}$ | $0.4922_{\pm 0.1001}$ | $0.3302_{\pm 0.0851}$ | $0.9333_{\pm 0.1155}$ | $0.9000_{\pm 0.1732}$ |
| GNN-RAG | $0.5270_{\pm 0.1006}$ | $0.4900_{\pm 0.1646}$ | $0.5056_{\pm 0.1376}$ | $0.3456_{\pm 0.1177}$ | $0.6667_{\pm 0.5774}$ | $0.6000_{\pm 0.5196}$ |
| GALAX | $0.5247_{\pm 0.0081}$ | $0.5033_{\pm 0.0289}$ | $0.5133_{\pm 0.0115}$ | $0.3453_{\pm 0.0104}$ | $\mathbf{1.0000}_{\pm \mathbf{0.0000}}$ | $0.9000_{\pm 0.1000}$ |
| GALAX (Qwen2.5-7B) | $\mathbf{0.5852}_{\pm \mathbf{0.0091}}$ | $\mathbf{0.5733}_{\pm \mathbf{0.0115}}$ | $\mathbf{0.5791}_{\pm \mathbf{0.0015}}$ | $\mathbf{0.4076}_{\pm \mathbf{0.0015}}$ | $0.9333_{\pm 0.1155}$ | $0.9000_{\pm 0.0000}$ |

**Table 42:** Model performance on UCEC

| Model | Precision ↑ | Recall ↑ | F1 ↑ | Jaccard ↑ | Hit@5 ↑ | Hit@10 ↑ |
|---|---|---|---|---|---|---|
| M2T | 0.0000 | 0.0000 | 0.0000 | 0.0000 | 0.0000 | 0.0000 |
| GAT | $0.0000_{\pm 0.0000}$ | $0.0000_{\pm 0.0000}$ | $0.0000_{\pm 0.0000}$ | $0.0000_{\pm 0.0000}$ | $0.0000_{\pm 0.0000}$ | $0.0000_{\pm 0.0000}$ |
| L3 + Omics | $0.0000_{\pm 0.0000}$ | $0.0000_{\pm 0.0000}$ | $0.0000_{\pm 0.0000}$ | $0.0000_{\pm 0.0000}$ | $0.0000_{\pm 0.0000}$ | $0.0000_{\pm 0.0000}$ |
| L3 + Omics + KG | $0.0000_{\pm 0.0000}$ | $0.0000_{\pm 0.0000}$ | $0.0000_{\pm 0.0000}$ | $0.0000_{\pm 0.0000}$ | $0.0000_{\pm 0.0000}$ | $0.0000_{\pm 0.0000}$ |
| L3-FT(Med) + Omics | $0.0000_{\pm 0.0000}$ | $0.0000_{\pm 0.0000}$ | $0.0000_{\pm 0.0000}$ | $0.0000_{\pm 0.0000}$ | $0.0000_{\pm 0.0000}$ | $0.0000_{\pm 0.0000}$ |
| L3-FT(Med) + Omics + KG | $0.0000_{\pm 0.0000}$ | $0.0000_{\pm 0.0000}$ | $0.0000_{\pm 0.0000}$ | $0.0000_{\pm 0.0000}$ | $0.0000_{\pm 0.0000}$ | $0.0000_{\pm 0.0000}$ |
| L3-FT(CRISPR) + Omics | $\mathbf{0.5293}_{\pm \mathbf{0.0394}}$ | $0.5133_{\pm 0.0379}$ | $0.5211_{\pm 0.0379}$ | $0.3530_{\pm 0.0344}$ | $0.7333_{\pm 0.1155}$ | $0.7000_{\pm 0.1000}$ |
| L3-FT(CRISPR) + Omics + KG | $0.4672_{\pm 0.0472}$ | $0.4533_{\pm 0.0231}$ | $0.4600_{\pm 0.0346}$ | $0.2991_{\pm 0.0296}$ | $0.8000_{\pm 0.2000}$ | $0.8000_{\pm 0.1000}$ |
| G-Retriever + pre-GAT | $0.4517_{\pm 0.0318}$ | $0.4000_{\pm 0.0173}$ | $0.4242_{\pm 0.0237}$ | $0.2694_{\pm 0.0193}$ | $\mathbf{1.0000}_{\pm \mathbf{0.0000}}$ | $0.9000_{\pm 0.0000}$ |
| RoG | $0.4508_{\pm 0.0506}$ | $0.4233_{\pm 0.0982}$ | $0.4355_{\pm 0.0772}$ | $0.2803_{\pm 0.0614}$ | $0.6667_{\pm 0.5774}$ | $0.4667_{\pm 0.4041}$ |
| SubgraphRAG | $0.4800_{\pm 0.0000}$ | $0.4800_{\pm 0.0000}$ | $0.4800_{\pm 0.0000}$ | $0.3158_{\pm 0.0000}$ | $\mathbf{1.0000}_{\pm \mathbf{0.0000}}$ | $0.7000_{\pm 0.0000}$ |
| GNN-RAG | $0.4777_{\pm 0.0739}$ | $0.4433_{\pm 0.1159}$ | $0.4589_{\pm 0.0975}$ | $0.3011_{\pm 0.0795}$ | $0.6000_{\pm 0.5292}$ | $0.4667_{\pm 0.4163}$ |
| GALAX | $0.5280_{\pm 0.0555}$ | $\mathbf{0.5233}_{\pm \mathbf{0.0635}}$ | $\mathbf{0.5256}_{\pm \mathbf{0.0596}}$ | $\mathbf{0.3579}_{\pm \mathbf{0.0536}}$ | $0.8000_{\pm 0.0000}$ | $\mathbf{0.9000}_{\pm \mathbf{0.0000}}$ |
| GALAX (Qwen2.5-7B) | $0.4848_{\pm 0.0391}$ | $0.5033_{\pm 0.0231}$ | $0.4938_{\pm 0.0313}$ | $0.3282_{\pm 0.0280}$ | $0.9333_{\pm 0.1155}$ | $0.8333_{\pm 0.1155}$ |

**Table 43:** Model performance on PRAD

| Model | Precision ↑ | Recall ↑ | F1 ↑ | Jaccard ↑ | Hit@5 ↑ | Hit@10 ↑ |
|---|---|---|---|---|---|---|
| M2T | 0.0000 | 0.0000 | 0.0000 | 0.0000 | 0.0000 | 0.0000 |
| GAT | $0.0000_{\pm 0.0000}$ | $0.0000_{\pm 0.0000}$ | $0.0000_{\pm 0.0000}$ | $0.0000_{\pm 0.0000}$ | $0.0000_{\pm 0.0000}$ | $0.0000_{\pm 0.0000}$ |
| L3 + Omics | $0.0111_{\pm 0.0192}$ | $0.0033_{\pm 0.0058}$ | $0.0051_{\pm 0.0089}$ | $0.0026_{\pm 0.0045}$ | $0.0000_{\pm 0.0000}$ | $0.0000_{\pm 0.0000}$ |
| L3 + Omics + KG | $0.0000_{\pm 0.0000}$ | $0.0000_{\pm 0.0000}$ | $0.0000_{\pm 0.0000}$ | $0.0000_{\pm 0.0000}$ | $0.0000_{\pm 0.0000}$ | $0.0000_{\pm 0.0000}$ |
| L3-FT(Med) + Omics | $0.0145_{\pm 0.0251}$ | $0.0033_{\pm 0.0058}$ | $0.0054_{\pm 0.0094}$ | $0.0027_{\pm 0.0047}$ | $0.0000_{\pm 0.0000}$ | $0.0000_{\pm 0.0000}$ |
| L3-FT(Med) + Omics + KG | $0.0176_{\pm 0.0261}$ | $0.0067_{\pm 0.0058}$ | $0.0078_{\pm 0.0083}$ | $0.0039_{\pm 0.0042}$ | $0.0000_{\pm 0.0000}$ | $0.0333_{\pm 0.0577}$ |
| L3-FT(CRISPR) + Omics | $0.6021_{\pm 0.0558}$ | $0.5600_{\pm 0.0400}$ | $0.5796_{\pm 0.0401}$ | $0.4088_{\pm 0.0392}$ | $\mathbf{1.0000}_{\pm \mathbf{0.0000}}$ | $0.9333_{\pm 0.1155}$ |
| L3-FT(CRISPR) + Omics + KG | $0.6032_{\pm 0.0191}$ | $0.5867_{\pm 0.0115}$ | $0.5947_{\pm 0.0102}$ | $0.4232_{\pm 0.0103}$ | $0.9333_{\pm 0.1155}$ | $0.9000_{\pm 0.1000}$ |
| G-Retriever + pre-GAT | $0.4749_{\pm 0.0041}$ | $0.4433_{\pm 0.0231}$ | $0.4584_{\pm 0.0141}$ | $0.2974_{\pm 0.0120}$ | $\mathbf{1.0000}_{\pm \mathbf{0.0000}}$ | $\mathbf{1.0000}_{\pm \mathbf{0.0000}}$ |
| RoG | $0.5833_{\pm 0.0404}$ | $0.5833_{\pm 0.0404}$ | $0.5833_{\pm 0.0404}$ | $0.4125_{\pm 0.0410}$ | $\mathbf{1.0000}_{\pm \mathbf{0.0000}}$ | $0.9333_{\pm 0.0577}$ |
| SubgraphRAG | $0.5931_{\pm 0.1258}$ | $0.4500_{\pm 0.0964}$ | $0.5042_{\pm 0.0691}$ | $0.3389_{\pm 0.0607}$ | $0.6667_{\pm 0.5774}$ | $0.6333_{\pm 0.5508}$ |
| GNN-RAG | $0.6140_{\pm 0.0138}$ | $0.6100_{\pm 0.0173}$ | $0.6120_{\pm 0.0156}$ | $0.4411_{\pm 0.0163}$ | $\mathbf{1.0000}_{\pm \mathbf{0.0000}}$ | $\mathbf{1.0000}_{\pm \mathbf{0.0000}}$ |
| GALAX | $0.5152_{\pm 0.1398}$ | $0.4733_{\pm 0.0751}$ | $0.4923_{\pm 0.1036}$ | $0.3308_{\pm 0.0934}$ | $0.9333_{\pm 0.1155}$ | $0.9667_{\pm 0.0577}$ |
| GALAX (Qwen2.5-7B) | $\mathbf{0.6195}_{\pm \mathbf{0.0292}}$ | $\mathbf{0.6133}_{\pm \mathbf{0.0289}}$ | $\mathbf{0.6164}_{\pm \mathbf{0.0290}}$ | $\mathbf{0.4459}_{\pm \mathbf{0.0300}}$ | $\mathbf{1.0000}_{\pm \mathbf{0.0000}}$ | $0.9000_{\pm 0.0000}$ |

