# OpenReview forum: "GALAX: Graph-Augmented Language Model for Explainable Reinforcement-Guided Subgraph Reasoning in Precision Medicine"
_ICLR.cc/2026/Conference — ICLR 2026 Poster_

### Official Review · Reviewer_KmG9 · 2025-10-31

**Soundness:** 2
**Presentation:** 3
**Contribution:** 2
**Rating:** 2
**Confidence:** 4

**Summary:**

Precision medicine relies on the use of multi-omic features, structural information as context and textual biological knowledge. Each of these aspects however, have their own shortcomings. The paper suggests leveraging their strengths by proposing a unified framework of subgraph reasoning. The paper proposes GALAX, a method that combines pretrained language models with Graph Neural Networks (GNNs) for reasoning using reinforcement learning. A process reward model over graphs guides the subgraph generation process pertaining to disease information. The final reasoning traces then leverage subgraphs to generate query responses. Authors further propose Target-QA, a QA benchmark for precision medicine, developed using the DepMap for evaluating GALAX. Empirical evaluations demonstrate that GALAX improves over considered baselines and its components are found to be performant.

**Strengths:**

* The paper is combined an innovative set of ideas.
* The paper is well organized and comprehensive.

**Weaknesses:**

* **Motivation:** The central motivation and idea behind the paper remains unclear. The paper states that it aims to integrate multi-omics, topological structure and textual information in order to curb their limitations. However, it remains unclear how the paper carries this out as combining these aspects leads to combining both their strengths and weaknesses. The paper does not mention on how multi-omics and structure interact with each other and how language plays an important role in representing topological structure. Furthermore, authors claim that the combination of these aspects makes reasoning explainable/interpretable. However, this claim is not empirically studied or validated as the paper does not mechanistically study interpretability of the reasoning process. It would be worthwhile to refine the central motivation and scope of the work.

* **Dataset:** Authors curate and propose the Target-QA dataset. However, it remains unclear as to why the dataset is needed and how it serves evaluation. Does the dataset balance between quality and diversity of medicinal queries? Why curate a dataset from an established dataset for evaluation? Were other QA datasets in phenomics, transcriptomics and biological summarization considered? It also remains unclear on how effective the dataset is evaluating GALAX. The paper states that Target-QA consists of 363 QA pairs which are significantly low for a textual QA dataset.

* **Experiments & Ablations:** My main concern is that the paper does not validate its central contribution. Experiments currently compare GALAX to different settings of data aggregation and training. However, the main subgraph generation scheme using process reward model is not empirically studied. How does RL help in the generation process? How do process rewards contribute to guidance? How does RL compare to SFT? What happens if we ablate dense rewards with sparse rewards? How does the model behave when we replace subgraphs with the full knowledge graph? These questions remain unexplored corresponding to the central theme of the work.

* **Baselines:** Experimental setting currently considers different training and dataset modality regimes. However, these baselines do not put the method and its central contribution of subgraphs in perspective. It would be worthwhile to consider LLM with KGs for process rewards. Furthermore, what happens if we assess generation across different cell lines or heldout structures? Baseline ablations could help answer these questions.

* **Presentation:** Overall the presentation of the paper is dense. The paper presents complex terminology which is often overloaded and makes the text harder to understand. Section 4 is presented in a vague form. Various details on the graph generation process and training could be placed in the appendix for improving readability.

#### Minors
* line 185: has has $\rightarrow$ has
* Table 1: what is LUAD and BRCA?

**Questions:**

Refer to weaknesses

---

> ### Author Response · Authors · 2025-11-25
> **Response to Reviewer KmG9 (Part I)**
>
> Thank you for your comments. Below we provide a point-by-point response to each of your questions. We sincerely hope our clarifications address your concerns and hope you can reconsider your assessment of our work if you find our response satisfactory.
>
> ---
>
> **W1.1 The central motivation and idea behind the paper remains unclear.**
>
> **A1.1:**  Our motivation is introduced in the first two paragraphs in the introduction section, and we restate it here for clarity.
> -  **Limitations of existing modeling approaches.** Differential expression analysis and standard graph-based models show limited predictive performance. Language-only LLMs can achieve stronger performance but still struggle with the complexity of disease biology, often hallucinate, and lack grounding in biological constraints, which limits their interpretability. These shortcomings motivate efforts to strengthen LLM reasoning through RAG and graph-RAG approaches.
> - **Limitations of current RAG and graph-RAG methods.** Although methods such as RoG and G-Retriever incorporate knowledge graphs to enhance LLM reasoning, but they only optimize for final answers and lack checks on intermediate reasoning and whether the retrieved subgraphs are meaningful or cancer-relevant, since they operate over large and noisy textual graphs that lack the sample-specific omic signals. As a result, these approaches cannot enforce biologically grounded structure and therefore offer limited mechanistic interpretability.
> - **Limitations of LLMs with PRM-style supervision.** Existing PRM-based methods provide step-wise rewards but struggle to define fine-grained reasoning steps, verify intermediate correctness, or prevent reward hacking. These challenges become more severe in biomedicine: the biomedical graph is extremely large and noisy even when restricted to disease-related proteins, and the combinatorial growth of possible paths makes it difficult for PRM-style supervision to guide reasoning. As a result, standard RL on such graphs leads to unstable rewards and high computational cost.
> - **Our solution: biologically grounded, explainable reinforcement-guided subgraph reasoning.** To address these issues, we introduce a graph process reward model that evaluates each intermediate subgraph reasoning using a pretrained graph foundation model together with the biomedical knowledge graph as fixed priors, while incorporating numerical omic evidence for cell-line specific reasoning. This design removes the need for explicit labels, constrains the RL process to biologically meaningful structure, and yields interpretable subgraphs for each prediction. GALAX is, to our knowledge, the first model to unify numerical omic evidence, literature-scale textual information, and biological topology within an explainable reinforcement-guided subgraph reasoning framework, enabling reliable and interpretable target prioritization with consistent gains over strong LLM, RAG, and graph-based baselines.
>
> **W1.2 How multi-omics and structure interact with each other:**
>
> **A1.2:** We mention how multi-omics and structure interact via the problem formulation. In our model, we represent the numerical evidence from raw dataset as
>
> $$
> \mathcal{X}^{(0)} = \{X_n^{(0)}\}_{n=1}^{N^{(0)}} \in \mathbb{R}^{N^{(0)} \times M},
> $$
>
> where each sample $X_n^{(0)} \in \mathbb{R}^{M}$ contains $M$ omic features, which aligns with the $M$ entities in the graph modal. In details, each sample has multi-omic profile
>
> $$
> X_n^{(0)} = \bigl[ \mathbf{x}_n^{(pm)} \oplus \mathbf{x}_n^{(g)} \oplus \mathbf{x}_n^{(t)} \oplus \mathbf{x}_n^{(p)} \bigr],
> $$
>
> with promoter, gene, transcript, and protein features whose lengths sum to $M$. Regarding to graph modal, the integrated graph $\mathcal{G}=\left\lbrace\mathcal{V},\mathcal{E}\right\rbrace$ is the biomedical knowledge graph with
>
> $$
> \mathcal{V}=\mathcal{V}^{(pm)} \cup \mathcal{V}^{(g)} \cup \mathcal{V}^{(t)} \cup \mathcal{V}^{(p)},
> \qquad
> |\mathcal{V}| = M.
> $$
>
> , where each vertex $v\in\mathcal{V}$ has numerical and textual attributes: the corresponding entry of $\mathcal{X}^{(0)}$ and a textual pair
>
> $$
> \mathcal{T}=\left\lbrace{T_{\text{name}},T_{\text{desc}}}\right\rbrace, \qquad |T_{\text{name}}|=|T_{\text{desc}}|=M.
> $$
>
> After preprocessing, we get multi-omic dataset $\mathcal{X} = \left\lbrace{X_n}\right\rbrace_{n=1}^{N} \in \mathbb{R}^{N \times M}$ with $N$ samples. Numerical features and textual attributes are embedded into the same dimensions and then encoded with a cross-modality encoder via $\text{ENC}_{\text{cross}}^{\text{pre}}(\mathcal{X}, \mathcal{T})$ during pretraining, projecting numerical omic evidence and textual descriptions into a shared vertex space. These embeddings serve as vertex features for message propagation in the graph neural network foundation model, which naturally integrates topological structure. Thus every feature in $\mathcal{X}$ maps to a vertex in $\mathcal{V}$, giving a coherent fusion of omic values, text, and graph topology.

---

> ### Author Response · Authors · 2025-11-25
> **Response to Reviewer KmG9 (Part II)**
>
> **W1.3 How does language play an important role in representing topological structure?**
>
> **A1.3:** In our text-omic signaling graph, textual descriptions serve as canonical node features during initialization. This follows recent findings that language-based node representations can strengthen graph models and improve downstream performance. Prior studies show that adding textual information can enhance graph embeddings and prediction performance on the biomedical task [1], and that feature augmentation improve the graph node expressivity power, hence improve the model performance [2]. In our setting, language of entity name and description is integrated to enrich node initialization, while the downstream reasoning is driven by the graph topology and multi-omic evidence.
>
> **References**
> [1] Song, H., et al. (2025). Large Language Models Meet Graph Neural Networks for Text-Numeric Graph Reasoning. arXiv:2501.16361.
> [2] Dong, Z., et al. (2023). Rethinking the Power of Graph Canonization in Graph Representation Learning with Stability. arXiv:2309.00738.
>
> **W1.4 Furthermore, authors claim that the combination of these aspects makes reasoning explainable / interpretable.**
>
> **A1.4:** While multimodal components are essential to our model, we do not claim that their combination alone guarantees explainability. Our contribution is a framework for target prioritization through explainable, reinforcement-guided subgraph reasoning without explicit labels. Although multimodal data cannot guarantee interpretability, it remains essential because a biomedical graph is extremely large and noisy, and even methods that fuse topology with omic signals perform poorly without structured supervision to reveal sample-specific biological routes. Pure LLM reasoning with PRM-style supervision is infeasible in this setting, since the combinatorial growth of possible paths and the lack of a reliable supervisor lead to unstable trajectories and make standard RL prone to reward hacking. To address these challenges, we introduce a graph process reward model that scores each intermediate subgraph using a pretrained graph foundation model and the biomedical knowledge graph as fixed priors, while incorporating numerical omic evidence for cell-line specific reasoning without explicit labels. This constrains RL to be biological meaningful and reduces reward hacking by grounding the scoring in biological plausibility. In this way, multimodal signals remain essential because they offer complementary evidence about disease biology. Existing RAG systems cannot use quantitative omic information, and current RL-based reasoning methods ignore biological plausibility, whereas GALAX aligns omics with graph-based rewards so that the model learns to assemble disease-relevant and interpretable subgraph trajectories.
>
> **W1.5 However, this claim is not empirically studied or validated as the paper does not mechanistically study interpretability of the reasoning proces**
>
> **A1.5:** To further validate the biological relevance of the extracted subgraph, we performed functional enrichment analysis and reported in the main results part of section 5 and Figure 4 for visualization of generate subgraph. The results reveal significant enrichment in cancer-associated signaling pathways, including the cancer pathway WP5434 and EGFR-related receptor signaling pathways (such as WP138 and WP3680), as cataloged in WikiPathways [1]. Notably, EGFR, a well-established therapeutic target in NSCLC [2], appears in five enriched terms alongside PTK2 and WNT16, which are known to regulate invasion, epithelial-mesenchymal transition (EMT), and therapeutic resistance [3, 4]. This biological validity is further supported by disease enrichment analysis using the GAD DISEASE database [5], where lung cancer was identified as the top associated disease term ($p=0.0022$), involving GSTM3, APAF1, NOD2, MLLT3, GC, and EGFR. Aside from this, we also did lots of the enrichment analysis, the gene and pathway enrichment analysis of the generated subgraphs in the Appendix E.
>
> **References:**
>
> [1] Agrawal, A., et al. WikiPathways 2024: next generation pathway database. Nucleic Acids Research, 52(D1):D679–D689, 2024.
> [2] Steuer, C. E., & Ramalingam, S. S. Targeting EGFR in lung cancer: Lessons learned and future perspectives. Molecular Aspects of Medicine, 45:67–73, 2015.
> [3] Tong, X., et al. Protein tyrosine kinase 2: a novel therapeutic target to overcome acquired EGFR-TKI resistance in non-small cell lung cancer. Respiratory Research, 20:1–14, 2019.
> [4] Sun, Y., et al. Treatment-induced damage to the tumor microenvironment promotes prostate cancer therapy resistance through WNT16B. Nature Medicine, 18(9):1359–1368, 2012.
> [5] Sherman, B. T., et al. DAVID: a web server for functional enrichment analysis and functional annotation of gene lists (2021 update). Nucleic Acids Research, 50(W1):W216–W221, 2022.

---

> ### Author Response · Authors · 2025-11-25
> **Response to Reviewer KmG9 (Part III)**
>
> **W2.1 However, it remains unclear as to why the dataset is needed and how it serves evaluation.**
>
> **A2.1:** We constructed this dataset to address a critical gap in current resources: there is currently no existing dataset that integrates omics data and knowledge graphs (text) into an LLM-ready format for target prediction. We process the raw numerical omics data and graph topology and curate them into the Target-QA. However, To make the dataset suitable for fine-tuning and compatible with the context-window limits of LLMs, we retain top $K$ genes in each omic and convert the topological information into textual knowledge graph for a compact and LLM-ready QA dataset while keeping essential biosignals. But in galax, we can naturally integrate all modalities without needing for transformation To the best of our knowledge, we are the first to integrate these modalities for target priorization. The dataset serves two key purposes:
>
> **Bridging the Modality Gap (Why it is needed):** The original DepMap CRISPR dataset is not suitable for direct LLM fine-tuning. The information (multi-omics data, disease context, cell line types, CRISPR gene effects) is scattered across disparate files and lacks a unified Q&A structure. Furthermore, the raw omics data is not mapped to a biomedical knowledge graph, meaning it lacks the topological structure and textual annotations required for graph-based reasoning. We transformed this scattered data into an Q&A-format that unifies cell line-level omics with knowledge graph priors.
>
> **Standardized Benchmarking (How it serves evaluation):** This benchmark is specifically designed to evaluate the model's ability to rank potential therapeutic targets. We ground the evaluation in biological reality using CRISPR gene essentiality scores (where the most negative scores indicate effective inhibition of cell proliferation). The "ground truth" for a given query is defined as the top 100 targets with the lowest scores. This allows us to quantitatively assess how well the model prioritizes valid targets for each specific cancer cell line.
>
> **W2.2 Does the dataset balance between quality and diversity of medicinal queries?**
>
> **A2.2:** Yes, We prioritized high data quality by filtering the samples from raw DepMap dataset with complete CRISPR scores and cell line annotations. Also we processes the multi-omics data via existing tool, BioMedGraphica to integrate it into the text-omic siganling graph. To generate the Target-QA with ensuring diversity, we used a stratified 80/20 random split to guarantee that a wide range of cancer types (including LUAD, BRCA, GBM, etc.) are balanced distributed across training and testing dataset sets, ensuring the model generalizes across distinct biological contexts.
>
> **W2.3 Why curate a dataset from an established dataset for evaluation? Were other QA datasets in phenomics, transcriptomics and biological summarization considered?**
>
> **A2.3:** As detailed in **A2.1**, curation was essential because the established DepMap dataset serves as a raw data repository, not a machine-learning benchmark. The original information is scattered across disparate numeric files and lacks the textual structure required for LLM input. We transformed this scattered data into a standardized Q&A format that unifies cell line-level omics with knowledge graph priors. This conversion effectively bridges the gap between raw omics data and generative AI, creating, to our knowledge, the first dataset to enable direct training and evaluation of LLMs using cell line omic data for target prioritization.
>
> **W2.4 It also remains unclear on how effective the dataset is evaluating GALAX. The paper states that Target-QA consists of 363 QA pairs which are significantly low for a textual QA dataset.**
>
> **A2.4:** We prioritize the biological validity of the data over raw quantity. The ground truth for Target-QA was rigorously derived from CRISPR-based gene effect scores, specifically targeting the lowest 100 scores which indicate the strongest inhibition of cell proliferation. Therefore, this serves as a highly reliable ground truth for evaluating GALAX. To guarantee quality, we removed cell lines lacking essential annotations or CRISPR information. It is important to note that the raw data available containing both validated cell line annotations and CRISPR gene effect scores is inherently limited; thus, the dataset size reflects the maximum number of high-quality, biologically verified samples available.

---

> ### Author Response · Authors · 2025-11-25
> **Response to Reviewer KmG9 (Part IV)**
>
> **W3.1 How does RL help in the generation process?**
>
> **A3.1:** We metioned this in our paper Section 4.2. Here we restate the RL in generation process.
>
> RL is essential to the GALAX framework as it enables the model to perform robust reasoning by generating disease-relevant subgraphs without step-wise labels. Also, it addresses a critical challenge in biomedical reasoning: the absence of ground-truth labels for intermediate reasoning steps, since explicit annotations for the step-wise construction of signaling pathways are unavailable under this scenario. Therefore, RL can guide the reasoning with policy model informed by four components:
>
> - **schema-based rule term**: uses the biomedical knowledge graph to perform rule checks, which apply a rule-based reward term to penalize schema violations.
> - **frozen graph oncogenicity assessment classifier**: our reward signals on graph oncogenicity are derived from a frozen, pre-trained foundation model rather than a co-evolving reward model.
> - **rollout-based future reward**: evaluates the long-term effect of each action, making it difficult for the generator to capitalize on local irregularities or short-term reward gains.
> - **stepwise quality gating**: discards any action with negative total reward, effectively constraining the policy to only take biologically meaningful steps (meet both rule schema and graph oncogenicity).
>
> **W3.2 How do process rewards contribute to guidance?**
>
> **A3.2:** Thanks for this insightful comment. We conducted an ablation study isolating the four reward components: the graph oncogenicity assessment (Graph), the biomedical rule check (Rules), rollout-based future reward (Rollout) and stepwise quality gating (Gating). As shown in table, while the rollout and gating modules are crucial for preventing short-sighted decisions and filtering noisy reasoning paths, respectively, the graph supervisor and schema rules provide the foundational constraints against exploitation. Specifically, removing the graph supervisor or rules consistently degrades performance, demonstrating that the frozen graph offers non-exploitable biological guidance while schema constraints block invalid structural paths. Collectively, these mechanisms force the generator to construct biologically consistent subgraphs rather than exploiting the reward function with high-scoring but implausible outputs, rendering the PRM substantially less gameable than prior approaches. **We added this in Appendix D.2.**
>
> | Model Variant | Precision ↑ | Recall ↑ | F1 ↑ | Jaccard ↑ | Hit@5 ↑ | Hit@10 ↑ |
> |---------------|-------------|----------|------|-----------|---------|----------|
> | L3-FT(QA) + Omics | 0.5250 ± 0.0282 | 0.4959 ± 0.0435 | 0.5094 ± 0.0365 | 0.3449 ± 0.0338 | 0.8889 ± 0.0168 | 0.8693 ± 0.0157 |
> | L3-FT(QA) + Omics + KG | 0.5185 ± 0.0240 | 0.4908 ± 0.0402 | 0.5038 ± 0.0327 | 0.3393 ± 0.0298 | 0.8794 ± 0.0114 | 0.8529 ± 0.0153 |
> | GALAX w/o Graph | 0.5314 ± 0.0045 | 0.5275 ± 0.0077 | 0.5336 ± 0.0053 | 0.3680 ± 0.0043 | 0.9048 ± 0.0084 | 0.8741 ± 0.0056 |
> | GALAX w/o Rules | 0.5305 ± 0.0074 | 0.5154 ± 0.0115 | 0.5221 ± 0.0098 | 0.3570 ± 0.0078 | 0.8974 ± 0.0073 | 0.8746 ± 0.0042 |
> | GALAX w/o Rollout | 0.5362 ± 0.0060 | 0.5196 ± 0.0052 | 0.5271 ± 0.0046 | 0.3628 ± 0.0043 | 0.9067 ± 0.0062 | 0.8769 ± 0.0055 |
> | GALAX w/o Gating | 0.5387 ± 0.0050 | 0.5243 ± 0.0042 | 0.5308 ± 0.0024 | 0.3651 ± 0.0032 | 0.9118 ± 0.0056 | 0.8782 ± 0.0046 |
> | **GALAX (Full Model)** | **0.5472 ± 0.0053** | **0.5332 ± 0.0031** | **0.5399 ± 0.0041** | **0.3726 ± 0.0037** | **0.9249 ± 0.0048** | **0.8815 ± 0.0033** |
>
> **W3.3 How does RL compare to SFT?**
>
> **A3.3:** We mentioned this result in the original main text in Table 1 and Table 2 to evaluate the RL compared to standard SFT, where we compare GALAX against the **L3-FT(QA) + Omics + KG** baseline. This SFT variant utilizes the same input data—top-$K$ multi-omic features and pre-retrieved disease-associated subgraphs—but processes them as static context within the prompt. As shown in above Table, GALAX consistently outperforms the SFT baseline across all evaluation metrics. For instance, GALAX improves the F1 score from 0.5038 to 0.5399 and Hit@5 from 0.8794 to 0.9249. These results indicate that the RL-guided mechanism, which actively constructs and reasons over the subgraph, yields significantly better target prioritization than simply augmenting an SFT model with static graph context.
>
> | Model | Precision ↑ | Recall ↑ | F1 ↑ | Jaccard ↑ | Hit@5 ↑ | Hit@10 ↑ |
> |-------|-------------|----------|------|-----------|---------|----------|
> | L3-FT(QA) + Omics | 0.5250 ± 0.0282 | 0.4959 ± 0.0435 | 0.5094 ± 0.0365 | 0.3449 ± 0.0338 | 0.8889 ± 0.0168 | 0.8693 ± 0.0157 |
> | L3-FT(QA) + Omics + KG | 0.5185 ± 0.0240 | 0.4908 ± 0.0402 | 0.5038 ± 0.0327 | 0.3393 ± 0.0298 | 0.8794 ± 0.0114 | 0.8529 ± 0.0153 |
> | **GALAX** | **0.5472 ± 0.0053** | **0.5332 ± 0.0031** | **0.5399 ± 0.0041** | **0.3726 ± 0.0037** | **0.9249 ± 0.0048** | **0.8815 ± 0.0033** |

---

> ### Author Response · Authors · 2025-11-25
> **Response to Reviewer KmG9 (Part V)**
>
> **W3.4 What happens if we ablate dense rewards with sparse rewards?**
>
> **A3.4:** We appreciate the reviewer for raising this insightful suggestion. To study the effect of sparse versus dense rewards, we vary the reward calculation frequency by introducing a step interval parameter $\mathbf{s}$. Specifically, when $\mathbf{s}=2$, we compute the reward only every 2 steps during graph generation. Larger values of $\mathbf{s}$ produce *sparser*, more delayed reward signals, while smaller values (approaching $\mathbf{s}=1$) provide *denser*, more immediate feedback. The results are shown in the table below.
>
> Across all reward calculation intervals, GALAX demonstrates consistent stability. Notably, performance improves as rewards become denser (smaller $\mathbf{s}$), with optimal results achieved at $\mathbf{s}=1$, the default GALAX configuration used in the main paper. This trend indicates that the graph generator benefits most from immediate biological feedback provided at each step by the graph process reward model. In contrast, less frequent reward calculations (larger $\mathbf{s}$) introduce noisier, longer-horizon estimates that dilute the signal quality. Importantly, even under highly sparse reward conditions (e.g., $\mathbf{s}=10$), GALAX consistently outperforms the baseline **L3-FT(QA) + Omics + KG**, demonstrating robustness to reward sparsity and confirming that the model is neither sensitive nor brittle to variations in reward specification frequency. **We added this part in Appendix D.2.**
>
>
> | Reward Interval ($\mathbf{s}$) | Precision | Recall | F1 Score | Jaccard | Hit@5 | Hit@10 |
> |:------------:|:---------:|:------:|:--------:|:-------:|:-----------:|:------------:|
> | 10 | 0.5312 ± 0.0044 | 0.5163 ± 0.0019 | 0.5232 ± 0.0031 | 0.3582 ± 0.0035 | 0.8996 ± 0.0050 | 0.8753 ± 0.0044 |
> | 8 | 0.5347 ± 0.0062 | 0.5224 ± 0.0043 | 0.5284 ± 0.0054 | 0.3619 ± 0.0055 | 0.9063 ± 0.0079 | 0.8771 ± 0.0043 |
> | 6 | 0.5398 ± 0.0051 | 0.5218 ± 0.0036 | 0.5316 ± 0.0044 | 0.3657 ± 0.0044 | 0.9024 ± 0.0079 | 0.8794 ± 0.0040 |
> | 4 | 0.5389 ± 0.0054 | 0.5278 ± 0.0025 | 0.5302 ± 0.0041 | 0.3673 ± 0.0046 | 0.9137 ± 0.0056 | 0.8789 ± 0.0038 |
> | 2 | 0.5463 ± 0.0057 | 0.5296 ± 0.0032 | 0.5384 ± 0.0040 | 0.3718 ± 0.0043 | 0.9218 ± 0.0048 | 0.8806 ± 0.0033 |
> | 1 (GALAX) | **0.5472 ± 0.0053** | **0.5332 ± 0.0031** | **0.5399 ± 0.0041** | **0.3726 ± 0.0037** | **0.9249 ± 0.0048** | **0.8815 ± 0.0033** |
> | L3-FT(QA) + Omics + KG | 0.5185 ± 0.0240     | 0.4908 ± 0.0402     | 0.5038 ± 0.0327     | 0.3393 ± 0.0298     | 0.8794 ± 0.0114     | 0.8529 ± 0.0153     |
>
> **W3.5 How does the model behave when we replace subgraphs with the full knowledge graph?**
>
> **A3.5:** We appreciate the reviewer for raising this essential point regarding the use of the full knowledge graph, as it directly echoes our motivation for creating GALAX. We clarify that directly inputting the entire graph is computationally infeasible due to the finite context window constraints of Large Language Models (LLMs). As detailed in Appendix B.2, our Text-Numeric Graph integrates over 834,000 multi-omic entities and 27 million edges. This scale far exceeds the token limits of current architectures, necessitating a subgraph retrieval or generation approach to render the data tractable for reasoning. To approximate the behavior of a full-graph model, we evaluated G-Retriever, which implicitly projects graph structures into the LLM via a soft projection layer to handle larger contexts. As shown in our experimental results, G-Retriever significantly underperforms compared to GALAX. Also, GAT get poor performances on this task on all metrics. This suggests that implicit projection of the whole graph introduces noise and lacks the structural coherence required for high-fidelity target identification, whereas our reinforcement-guided approach successfully extracts the specific, interpretable subgraphs necessary for accurate biological reasoning.
>
> | Model | Precision ↑ | Recall ↑ | F1 ↑ | Jaccard ↑ | Hit@5 ↑ | Hit@10 ↑ |
> |-------|-------------|----------|------|-----------|---------|----------|
> | GAT | 0.0006 ± 0.0000 | 0.0006 ± 0.0000 | 0.0006 ± 0.0000 | 0.0003 ± 0.0000 | 0.0000 ± 0.0000 | 0.0000 ± 0.0000 |
> | L3-FT(QA) + Omics + KG | 0.5185 ± 0.0240 | 0.4908 ± 0.0402 | 0.5038 ± 0.0327 | 0.3393 ± 0.0298 | 0.8794 ± 0.0114 | 0.8529 ± 0.0153 |
> | G-Retriever + pre-GAT | 0.4763 ± 0.0004 | 0.3929 ± 0.0063 | 0.4286 ± 0.0044 | 0.2757 ± 0.0038 | 0.8804 ± 0.0037 | 0.8550 ± 0.0046 |
> | **GALAX** | **0.5472 ± 0.0053** | **0.5332 ± 0.0031** | **0.5399 ± 0.0041** | **0.3726 ± 0.0037** | **0.9249 ± 0.0048** | **0.8815 ± 0.0033** |

---

> ### Author Response · Authors · 2025-11-25
> **Response to Reviewer KmG9 (Part VI)**
>
> **W4.1 Experimental setting currently considers different training and dataset modality regimes. These baselines do not put the method and its central contribution of subgraphs in perspective. It would be worthwhile to consider LLM with KGs for process rewards.**
>
> **A4.1:** We describe the modality reigmes for training and dataset modality in Section 3 in our paper. Here we restate our experimental setting, both training and dataset provide the same modality regimes with input of numerical omic evidence $X_n$, topological information $\mathcal{E}$ with annotated entities $\mathcal{T}$, and literature scale text information $Q_n$ that integrates the cell line textual biomedical background information.
>
> As to contribution of subgraphs by LLM with KGs for process rewards, we restate that GALAX is, to the best of our knowledge, the first framework to consider LLM with KGs for process reward by using the graph supervisor and rule check as the reward. As we stated before, there are no effective way to create KG process reward since there are no biological label for Process Reward Model (PRM). Nevertheless, GALAX effectively utilizes the KG to construct a Graph Process Reward Model (GPRM). This mechanism guides the LLM's reasoning via four specific reward components: (1) a graph supervisor (a pretrained graph foundation model) that assesses oncogenicity, (2) a rule-based check derived from biomedical schemas to ensure biological plausibility, (3) rollout-based future reward that evaluates the long-term effect of each action and (4) stepwise quality gating that discards any action with negative total reward. By comparing GALAX against baselines such as L3-FT(QA)+Omics+KG (which utilizes KGs without process-based reinforcement), we demonstrate the significant performance gains attributed to the process reward-guided subgraph generation (shown in above table in A3.5).
>
> Furthermore, we provide extensive ablations isolating the role of the process reward (as described in A3.2), and varying reward sparsity by reward step interval (as shown in A3.4). These analyses directly test the contribution of the reward-guided subgraph generator and show that GALAX’s improvements arise specifically from this component. **We added this part in revised Appendix Seciton D.2.**
>
> **W4.2 what happens if we assess generation across different cell lines or heldout structures?**
>
> **A4.2:** We value your recognition of the importance of evaluating GALAX's generalization capabilities. We address this concern in three ways: through **hold-out cell line cross-validation** experiments analyses of **knowledge graph incompleteness** for holdout structures.
>
> To assess cross-cancer transfer, we constructed three holdout sets where all cell lines sharing a TCGA code were excluded from training and reserved solely for evaluation—testing whether GALAX can predict CRISPR targets for entirely unseen cancer types.
>
> | **Holdout Set**   | **Train**   | **Test**    | **Train/Test TCGA Codes** | **Hold-out Cancer TCGA Codes**                                       |
> | ----------------- | ----------- | ----------- | ------------------------- | --------------------------------------------------------------- |
> | **Holdout Set 1** | 238 (65.6%) | 125 (34.4%) | 20 / 10                   | COAD/READ, DLBC, KIRC, LCML, LUSC, PAAD, PRAD, SARC, SKCM, STAD |
> | **Holdout Set 2** | 289 (79.6%) | 74 (20.4%)  | 21 / 9                    | BLCA, GBM, LAML, LGG, LIHC, MESO, MM, NB, UCEC                  |
> | **Holdout Set 3** | 220 (60.6%) | 143 (39.4%) | 21 / 9                    | ALL, BRCA, ESCA, HNSC, LUAD, MB, OV, SCLC, THCA                 |
>
> As expected, performance drops when entire cancer types are held out, since no finetuning data from those cancers is available. However, GALAX still maintains **strong accuracy**. Overall, the decline is modest, indicating that GALAX retains meaningful robustness and generalizes reliably to unseen cancer types. **We added this part in Appendix D.5.**
>
> | **Holdout Set** | **Precision ↑** | **Recall ↑** | **Hit@5 ↑** | **Hit@10 ↑** |
> | --------------- | --------------- | ------------ | ----------- | ------------ |
> | Holdout Set 1   | 0.4931          | 0.4501       | 0.9024      | 0.8688       |
> | Holdout Set 2   | 0.5124          | 0.5151       | 0.8703      | 0.8581       |
> | Holdout Set 3   | 0.4832          | 0.4715       | 0.8993      | 0.8448       |
> | **Average**     | **0.4962**      | **0.4789**   | **0.8907**  | **0.8572**   |
> | **Original**    | **0.5472**      | **0.5332**   | **0.9249**  | **0.8815**   |

---

> ### Author Response · Authors · 2025-11-25
> **Response to Reviewer KmG9 (VII)**
>
> **A4.2 Continue:**
> For holdout the structure, we do the ablation study to reduce the nodes and edges increamentally ((20\%, 40\%, 60\%, 80\%)) and run the experiments. As expected, performance decreases as more structure is removed.
>
> In the **edge-deletion** setting, **GALAX outperforms** the **L3(QA)+Omic+KG** baseline across all deletion levels. Even as KG connectivity decreases, node-level biological attributes remain intact, and GALAX can still reason over disease–protein interactions using the reduced structure as long as the node is still maintained in the subgraph structure. With **20% edge deletion**, GALAX retains noticeably higher **recall**, **Hit@5**, and **Hit@10**, demonstrating its ability to assemble **mechanistic subnetworks** from sparse evidence.
>
> This robustness stems from GALAX's reliance not only on observed KG edges but also on **pretrained GNN-based node embeddings** ($\text{GNN}^{\text{G}}_{\text{PPI}}$) that capture global biomedical structure. Moreover, its **graph supervisor** $g(\cdot)$ provides **oncogenicity-guided rewards**, enabling reconstruction of biologically meaningful connections even when edges are missing. In contrast, the baseline depends directly on the observed KG edges and lacks a mechanism to recover from connectivity loss, leading to **larger performance drops** as the graph becomes sparser. As deletion becomes more severe, both models naturally show reduced performance since limited information is provided for model to inference.
>
> In the node-deletion setting, removing nodes is substantially more destructive than removing edges because each deleted node also removes all of its incident interactions (e.g., across our settings, removing **20\% of nodes** leads on average to a **35\% reduction in edges**). Consequently, both GALAX and the baseline show similar declines in recall and precision as node deletion incre ases. However, their ranking performance diverges: the baseline’s Hit@5 and Hit@10 degrade steadily and substantially with higher deletion levels (e.g., 80\%), whereas **GALAX** maintains much more **stable top-K performance** across all node-deletion conditions. This indicates that GALAX continues to place the correct essential targets near the top of the ranking even when large portions of KG structure are missing. **We added this part in Section 5 Ablation study.**
>
> | Config | Setting | Recall ↑ | Precision ↑ | Hit@5 ↑ | Hit@10 ↑ |
> |--------|---------|----------|-------------|---------|----------|
> | **GALAX** | | | | | |
> | E20 | Drop 20% edges | 0.5061 ± 0.0268 | 0.5362 ± 0.0059 | 0.9005 ± 0.0128 | 0.8741 ± 0.0060 |
> | E40 | Drop 40% edges | 0.2871 ± 0.0060 | 0.5079 ± 0.0039 | 0.8762 ± 0.0138 | 0.8471 ± 0.0090 |
> | E60 | Drop 60% edges | 0.2753 ± 0.0020 | 0.4961 ± 0.0059 | 0.8635 ± 0.0084 | 0.8307 ± 0.0033 |
> | E80 | Drop 80% edges | 0.2775 ± 0.0134 | 0.4943 ± 0.0177 | 0.8434 ± 0.0524 | 0.8185 ± 0.0494 |
> | N20 | Drop 20% nodes | 0.2697 ± 0.0013 | 0.5034 ± 0.0014 | 0.8786 ± 0.0070 | 0.8560 ± 0.0092 |
> | N40 | Drop 40% nodes | 0.2675 ± 0.0033 | 0.4901 ± 0.0035 | 0.8878 ± 0.0066 | 0.8503 ± 0.0048 |
> | N60 | Drop 60% nodes | 0.2617 ± 0.0056 | 0.4929 ± 0.0045 | 0.8698 ± 0.0290 | 0.8385 ± 0.0247 |
> | N80 | Drop 80% nodes | 0.2653 ± 0.0032 | 0.4825 ± 0.0090 | 0.8341 ± 0.0105 | 0.8103 ± 0.0134 |
> | GALAX | Original | 0.5332 ± 0.0031 | 0.5472 ± 0.0053 | 0.9249 ± 0.0048 | 0.8815 ± 0.0033 |
> | **L3(QA) + Omic + KG** | | | | | |
> | E20 | Drop 20% edges | 0.4599 ± 0.0820 | 0.5114 ± 0.0036 | 0.8587 ± 0.0157 | 0.8373 ± 0.0258 |
> | E40 | Drop 40% edges | 0.2742 ± 0.0090 | 0.5064 ± 0.0099 | 0.8429 ± 0.0173 | 0.8226 ± 0.0169 |
> | E60 | Drop 60% edges | 0.2676 ± 0.0062 | 0.4991 ± 0.0099 | 0.8296 ± 0.0305 | 0.8111 ± 0.0175 |
> | E80 | Drop 80% edges | 0.2611 ± 0.0074 | 0.4880 ± 0.0086 | 0.8254 ± 0.0361 | 0.8063 ± 0.0370 |
> | N20 | Drop 20% nodes | 0.2662 ± 0.0059 | 0.4916 ± 0.0024 | 0.8434 ± 0.0186 | 0.8222 ± 0.0193 |
> | N40 | Drop 40% nodes | 0.2658 ± 0.0049 | 0.4838 ± 0.0169 | 0.8709 ± 0.0422 | 0.8339 ± 0.0335 |
> | N60 | Drop 60% nodes | 0.2648 ± 0.0000 | 0.4742 ± 0.0150 | 0.7857 ± 0.0247 | 0.7690 ± 0.0303 |
> | N80 | Drop 80% nodes | 0.2689 ± 0.0042 | 0.4722 ± 0.0063 | 0.7111 ± 0.0055 | 0.6974 ± 0.0111 |
> | L3(QA)+Omic+KG | Original | 0.4908 ± 0.0402 | 0.5185 ± 0.0240 | 0.8794 ± 0.0114 | 0.8529 ± 0.0153 |

---

> ### Author Response · Authors · 2025-11-25
> **Response to Reviewer KmG9 (VIII)**
>
> **W5 Presentation: Overall the presentation of the paper is dense. The paper presents complex terminology which is often overloaded and makes the text harder to understand. Section 4 is presented in a vague form. Various details on the graph generation process and training could be placed in the appendix for improving readability.**
>
> **A5:** Thank you for this helpful feedback. We acknowledge that integrating multi-omic data, knowledge graphs, and reinforcement learning involves complex terminology, and we have made extensive revisions to improve clarity in the Section 3 Problem Formulation. We have rewritten it to clearly define the core concepts and data modalities, ensuring a better introduction to the problem space. Specifically, we reformatted the description of multi-modality fusion to better articulate how these diverse data sources interact within our framework. Regarding the suggestion to move Section 4 details to the appendix, we respectfully maintain that the graph generation process described in this section constitutes a central contribution of our work. Section 4 introduces the novel integration of the Graph Process Reward Model (GPRM) for reinforcement-guided subgraph reasoning, which combines the graph oncogenicity assessment classifier with biomedical rule checks. These mechanisms are essential for generating biologically plausible subgraphs that augment LLM reasoning. While we have retained these core technical details in the main text to ensure the methodology is self-contained, we have refined the exposition to remove vagueness and improve the logical flow of the training process description.
>
>
> **Minors:**
>
> - **line 185: has has → has**
>   **Response:** We thank the reviewer for catching this oversight. We have corrected the typo in the revised manuscript.
>
> - **Table 1: what is LUAD and BRCA?**
>   **Response:** LUAD and BRCA are standard codes established by The Cancer Genome Atlas (TCGA) study, representing Lung  Adenocarcinoma and Breast Invasive Carcinoma, respectively. These two-to-four letter codes are ubiquitous in large-scale genomic datasets, such as DepMap, to indicate the tumor origin of a sample. To ensure these terms are clear to a broader audience, we have added Table 5 in Appendix B.2, which provides a comprehensive legend mapping all TCGA codes present in the Target-QA dataset to their full cancer type descriptions.

---

### Official Review · Reviewer_xSCJ · 2025-10-31

**Soundness:** 4
**Presentation:** 4
**Contribution:** 3
**Rating:** 8
**Confidence:** 4

**Summary:**

This paper introduces GALAX, a pipeline for QA in precision medicine using a joint LLM-GNN model using subgraphs generated via a graph process reward model as explainable artifacts for reasoning. The authors also introduce a new benchmark, Target-QA, focused on biomedical graph knowledge, multi-omic profiles, and CRISPR targets.

**Strengths:**

- The interpretability aspect of the method is strong and well-reasoned. Penalizing schema violations in the reward function and rewarding path-like structures is good.
- Experiments show GALAX consistently outperforms baselines. The performance is very good.
- Target-QA is an important contribution to the community for working with multi-omic features.

**Weaknesses:**

- There doesn't seem to be studies on the performance of the method when the KG is incomplete or noisy/not well aligned, which is almost always the case in real-world KGs.
- Most evaluation is done on Target-QA. Validation on external datasets is not emphasized, making it more difficult to assess the impact of this work.
- The authors mention reward hacking as a problem for previous PRMs. However, how the method combats this is not clear beyond the schema relation penalty. There is no analysis on how gameable the proposed graph PRM is.

**Questions:**

- How robust is the GPRM to shifts in important graph characteristics such as graph density or KG incompleteness?
- How sensitive is performance w.r.t. the backbone LLM?
- What are the failure modes of GALAX?  Are there particular paths/subgraphs that make any systematic failures evident?
- How robust is the NER function? It seems this is critical for seeding a good graph.

---

> ### Author Response · Authors · 2025-11-25
> **Response to Reviewer xSCJ (Part I)**
>
> Thank you for your thoughtful comments. Below we provide a point-by-point response to each of your questions. We sincerely hope our clarifications address your concerns.
>
> ---
>
> **W1: There doesn't seem to be studies on the performance of the method when the KG is incomplete or noisy/not well aligned, which is almost always the case in real-world KGs.**
>
> **A1:** We perform an ablation study in which we randomly remove edges or nodes at different rates (20%, 40%, 60%, 80%). As expected, performance decreases as more structure is removed.
> In the edge-deletion setting, GALAX consistently achieves higher performance than the L3(QA)+Omic+KG baseline across all deletion levels. Although removing edges reduces KG connectivity, node-level biological attributes are preserved, and only the available interactions for disease-related protein reasoning are limited. With 20% deletion, GALAX retains noticeably higher recall, Hit@5, and Hit@10, reflecting its ability to assemble biologically meaningful subnetworks using the remaining structure. This robustness arises because GALAX does not rely solely on the observed KG edges, but also leverages node representations produced by a pretrained GNN encoder ($\text{GNN}^\text{G}_\text{PPI}$), whose embeddings already capture global PPI and biomedical structure. In addition, GALAX uses a graph supervisor $g(\cdot)$ to guide subgraph generation through its oncogenicity-based reward evaluation, enabling the model to reconstruct biologically meaningful connections even when moderate portions of the KG are missing. In contrast, the baseline relies directly on the observed KG edges and lacks a mechanism to compensate for connectivity loss, leading to larger performance drops as the graph becomes sparser. When edge deletion becomes more severe, both models naturally exhibit larger decreases.
>
> In the node-deletion setting, removing nodes is substantially more destructive than removing edges because each deleted node also removes all of its incident interactions (e.g., in our experiments, removing 20% of nodes leads on average to a 35% reduction in edges). Consequently, both GALAX and the baseline show similar declines in recall and precision as node deletion increases. However, their ranking performance diverges: the baseline's Hit@5 and Hit@10 degrade steadily and substantially with higher deletion levels (e.g., 80%), whereas GALAX maintains much more stable top-K performance across all node-deletion conditions. This indicates that GALAX continues to place the correct essential targets near the top of the ranking even when large portions of KG structure are missing. **We added this part in Ablation study in Section 5.**
>
> | Config | Setting | Recall ↑ | Precision ↑ | Hit@5 ↑ | Hit@10 ↑ |
> |--------|---------|----------|-------------|---------|----------|
> | **GALAX** | | | | | |
> | E20 | Drop 20% edges | 0.5061 ± 0.0268 | 0.5362 ± 0.0059 | 0.9005 ± 0.0128 | 0.8741 ± 0.0060 |
> | E40 | Drop 40% edges | 0.2871 ± 0.0060 | 0.5079 ± 0.0039 | 0.8762 ± 0.0138 | 0.8471 ± 0.0090 |
> | E60 | Drop 60% edges | 0.2753 ± 0.0020 | 0.4961 ± 0.0059 | 0.8635 ± 0.0084 | 0.8307 ± 0.0033 |
> | E80 | Drop 80% edges | 0.2775 ± 0.0134 | 0.4943 ± 0.0177 | 0.8434 ± 0.0524 | 0.8185 ± 0.0494 |
> | N20 | Drop 20% nodes | 0.2697 ± 0.0013 | 0.5034 ± 0.0014 | 0.8786 ± 0.0070 | 0.8560 ± 0.0092 |
> | N40 | Drop 40% nodes | 0.2675 ± 0.0033 | 0.4901 ± 0.0035 | 0.8878 ± 0.0066 | 0.8503 ± 0.0048 |
> | N60 | Drop 60% nodes | 0.2617 ± 0.0056 | 0.4929 ± 0.0045 | 0.8698 ± 0.0290 | 0.8385 ± 0.0247 |
> | N80 | Drop 80% nodes | 0.2653 ± 0.0032 | 0.4825 ± 0.0090 | 0.8341 ± 0.0105 | 0.8103 ± 0.0134 |
> | GALAX | Original | 0.5332 ± 0.0031 | 0.5472 ± 0.0053 | 0.9249 ± 0.0048 | 0.8815 ± 0.0033 |
> | **L3(QA) + Omic + KG** | | | | | |
> | E20 | Drop 20% edges | 0.4599 ± 0.0820 | 0.5114 ± 0.0036 | 0.8587 ± 0.0157 | 0.8373 ± 0.0258 |
> | E40 | Drop 40% edges | 0.2742 ± 0.0090 | 0.5064 ± 0.0099 | 0.8429 ± 0.0173 | 0.8226 ± 0.0169 |
> | E60 | Drop 60% edges | 0.2676 ± 0.0062 | 0.4991 ± 0.0099 | 0.8296 ± 0.0305 | 0.8111 ± 0.0175 |
> | E80 | Drop 80% edges | 0.2611 ± 0.0074 | 0.4880 ± 0.0086 | 0.8254 ± 0.0361 | 0.8063 ± 0.0370 |
> | N20 | Drop 20% nodes | 0.2662 ± 0.0059 | 0.4916 ± 0.0024 | 0.8434 ± 0.0186 | 0.8222 ± 0.0193 |
> | N40 | Drop 40% nodes | 0.2658 ± 0.0049 | 0.4838 ± 0.0169 | 0.8709 ± 0.0422 | 0.8339 ± 0.0335 |
> | N60 | Drop 60% nodes | 0.2648 ± 0.0000 | 0.4742 ± 0.0150 | 0.7857 ± 0.0247 | 0.7690 ± 0.0303 |
> | N80 | Drop 80% nodes | 0.2689 ± 0.0042 | 0.4722 ± 0.0063 | 0.7111 ± 0.0055 | 0.6974 ± 0.0111 |
> | L3(QA)+Omic+KG | Original | 0.4908 ± 0.0402 | 0.5185 ± 0.0240 | 0.8794 ± 0.0114 | 0.8529 ± 0.0153 |

---

> ### Author Response · Authors · 2025-11-25
> **Response to Reviewer xSCJ (Part II)**
>
> **W2: Most evaluation is done on Target-QA. Validation on external datasets is not emphasized, making it more difficult to assess the impact of this work.**
>
> **A2:** We appreciate the reviewer's suggestion to include validation on an external dataset. We draw on the PedDep (https://peddep.org/) resource from the Broad Institute, which provides multi-omic profiles for pediatric tumor cell lines. After integrating the available omic modalities and intersecting these samples with cell lines that report CRISPR gene effect scores with sample annotations, we obtain 31 matched samples/cell lines. Due to the very small number of available cases, no fine-tuning is performed on this dataset, and the results reported here reflect strict zero-shot performance. **We added this part in Appendix E.6.**
>
> | Model | Precision ↑ | Recall ↑ | F1 ↑ | Jaccard ↑ | Hit@5 ↑ | Hit@10 ↑ |
> |-------|-------------|----------|------|-----------|---------|----------|
> | GAT | 0.0005 ± 0.0008 | 0.0005 ± 0.0008 | 0.0005 ± 0.0008 | 0.0002 ± 0.0004 | 0.0000 ± 0.0000 | 0.0000 ± 0.0000 |
> | L3-FT(Med) + Omics | 0.0144 ± 0.0081 | 0.0114 ± 0.0049 | 0.0099 ± 0.0028 | 0.0050 ± 0.0014 | 0.0210 ± 0.0421 | 0.0109 ± 0.0218 |
> | L3-FT(Med) + Omics + KG | 0.0167 ± 0.0161 | 0.0060 ± 0.0047 | 0.0068 ± 0.0049 | 0.0035 ± 0.0025 | 0.0072 ± 0.0143 | 0.0073 ± 0.0145 |
> | L3 + Omics | 0.0054 ± 0.0108 | 0.0011 ± 0.0021 | 0.0018 ± 0.0036 | 0.0009 ± 0.0018 | 0.0000 ± 0.0000 | 0.0000 ± 0.0000 |
> | L3 + Omics + KG | 0.0192 ± 0.0188 | 0.0032 ± 0.0022 | 0.0050 ± 0.0035 | 0.0026 ± 0.0018 | 0.0142 ± 0.0165 | 0.0214 ± 0.0247 |
> | L3-FT(QA) + Omics | 0.2608 ± 0.0268 | 0.2605 ± 0.0252 | 0.2606 ± 0.0260 | 0.1517 ± 0.0152 | 0.5810 ± 0.0330 | 0.4524 ± 0.0502 |
> | L3-FT(QA) + Omics + KG | 0.2619 ± 0.0075 | 0.2552 ± 0.0095 | 0.2584 ± 0.0086 | 0.1489 ± 0.0055 | 0.5048 ± 0.0165 | 0.4952 ± 0.0218 |
> | G-Retriever + pre-GAT | 0.2624 ± 0.0212 | 0.2610 ± 0.0206 | 0.2617 ± 0.0209 | 0.1522 ± 0.0121 | 0.5143 ± 0.0495 | 0.4524 ± 0.0825 |
> | ROG | 0.2730 ± 0.0050 | 0.2667 ± 0.0092 | 0.2697 ± 0.0071 | 0.1566 ± 0.0047 | 0.5143 ± 0.0286 | 0.4714 ± 0.0623 |
> | Subgraph-RAG | 0.2736 ± 0.0091 | 0.2690 ± 0.0095 | 0.2712 ± 0.0092 | 0.1579 ± 0.0062 | 0.5619 ± 0.0165 | 0.5286 ± 0.0571 |
> | GNN-RAG | 0.2760 ± 0.0092 | 0.2700 ± 0.0094 | 0.2728 ± 0.0093 | 0.1589 ± 0.0064 | 0.5714 ± 0.0495 | 0.5333 ± 0.0360 |
> | **GALAX** | 0.2914 ± 0.0115 | 0.2889 ± 0.0131 | 0.2901 ± 0.0123 | 0.1703 ± 0.0086 | 0.6357 ± 0.0589 | 0.5179 ± 0.0513 |
> | **GALAX (Qwen2.5-7B)** | **0.2921 ± 0.0055** | **0.2895 ± 0.0050** | **0.2908 ± 0.0053** | **0.1708 ± 0.0038** | **0.6667 ± 0.0595** | **0.5381 ± 0.0705** |

---

> ### Author Response · Authors · 2025-11-25
> **Response to Reviewer xSCJ (Part III)**
>
> **W3: The authors mention reward hacking as a problem for previous PRMs. However, how the method combats this is not clear beyond the schema relation penalty. There is no analysis on how gameable the proposed graph PRM is.**
>
> **A3:** We thank the reviewer for highlighting the importance of robustness in the reward signal. We agree that the rule-based schema constraint plays a important role in preventing GALAX from constructing biologically invalid edges. In addition to this component, GALAX incorporates three further mechanisms that collectively mitigate reward hacking beyond the schema penalty:
>
> **frozen graph oncogenicity assessment classifier**: our reward signals on graph oncogenicity are derived from a frozen, pre-trained foundation model rather than a co-evolving reward model.
>
> **rollout-based future reward**: evaluates the long-term effect of each action, making it difficult for the generator to capitalize on local irregularities or short-term reward gains.
>
> **stepwise quality gating**: discards any action with negative total reward, effectively constraining the policy to only take biologically meaningful steps (meet both rule schema and graph oncogenicity).
>
> Together with the **schema-based rule term**, these four components ensure that the reward is less gameable. Our ablation study confirms that GALAX’s robustness against reward hacking relies on the synergy of four non-redundant components: the graph supervisor, rule-based penalty, future reward rollout, and stepwise quality gating. While the rollout and gating modules are crucial for preventing short-sighted decisions and filtering noisy reasoning paths, respectively, the graph supervisor and schema rules provide the foundational constraints against exploitation. Specifically, removing the graph supervisor or rules consistently degrades performance, demonstrating that the frozen graph offers non-exploitable biological guidance while schema constraints block invalid structural shortcuts. Collectively, these mechanisms force the generator to construct biologically consistent subgraphs rather than exploiting the reward function with high-scoring but implausible outputs, rendering the PRM substantially less gameable than prior approaches. **We added this part to Appendix D.2.**
>
>
> | Model Variant | Precision ↑ | Recall ↑ | F1 ↑ | Jaccard ↑ | Hit@5 ↑ | Hit@10 ↑ |
> |---------------|-------------|----------|------|-----------|---------|----------|
> | L3-FT(QA) + Omics | 0.5250 ± 0.0282 | 0.4959 ± 0.0435 | 0.5094 ± 0.0365 | 0.3449 ± 0.0338 | 0.8889 ± 0.0168 | 0.8693 ± 0.0157 |
> | L3-FT(QA) + Omics + KG | 0.5185 ± 0.0240 | 0.4908 ± 0.0402 | 0.5038 ± 0.0327 | 0.3393 ± 0.0298 | 0.8794 ± 0.0114 | 0.8529 ± 0.0153 |
> | GALAX w/o Graph | 0.5314 ± 0.0045 | 0.5275 ± 0.0077 | 0.5336 ± 0.0053 | 0.3680 ± 0.0043 | 0.9048 ± 0.0084 | 0.8741 ± 0.0056 |
> | GALAX w/o Rules | 0.5305 ± 0.0074 | 0.5154 ± 0.0115 | 0.5221 ± 0.0098 | 0.3570 ± 0.0078 | 0.8974 ± 0.0073 | 0.8746 ± 0.0042 |
> | GALAX w/o Rollout | 0.5362 ± 0.0060 | 0.5196 ± 0.0052 | 0.5271 ± 0.0046 | 0.3628 ± 0.0043 | 0.9067 ± 0.0062 | 0.8769 ± 0.0055 |
> | GALAX w/o Gating | 0.5387 ± 0.0050 | 0.5243 ± 0.0042 | 0.5308 ± 0.0024 | 0.3651 ± 0.0032 | 0.9118 ± 0.0056 | 0.8782 ± 0.0046 |
> | **GALAX (Full Model)** | **0.5472 ± 0.0053** | **0.5332 ± 0.0031** | **0.5399 ± 0.0041** | **0.3726 ± 0.0037** | **0.9249 ± 0.0048** | **0.8815 ± 0.0033** |
>
> ---
>
> **Q1: How robust is the GPRM to shifts in important graph characteristics such as graph density or KG incompleteness?**
>
> **A1:** We evaluated robustness to KG incompleteness by randomly deleting edges or nodes at different rates (20\%, 40\%, 60\%, 80\%). The KG serves dual purposes in our framework: guiding graph generation and providing retrieved subgraphs as biological context for the LLM. Removing the KG would consequently reduce overall performance. We provide baseline results using incomplete KG structures to illustrate this effect. As expected, performance decreases as more structure is removed. The similar ablation study results was display in the table in **W1**.

---

> ### Author Response · Authors · 2025-11-25
> **Response to Reviewer xSCJ (Part IV)**
>
> **Q2: How sensitive is performance w.r.t. the backbone LLM?**
>
> **A2:** To evaluate the sensitivity of our framework regarding the choice of backbone LLM, we conduct additional experiments by replacing the original backbone with Qwen2.5-7B. As shown in table below, the framework maintains consistent performance across different backbones. While the original model retains a slight advantage in Precision and Hit@5, the Qwen2.5-7B variant achieves comparable or slightly superior results in Recall, F1, and Jaccard scores. These results indicate that the effectiveness of our proposed framework is stable and can be successfully adapted to other modern LLM architectures without significant performance degradation. **And we added the results to our Tables 1-2 in Section 5 and Tables 16-43 in Appendix D.4.**
>
> | Model | Precision ↑ | Recall ↑ | F1 ↑ | Jaccard ↑ | Hit@5 ↑ | Hit@10 ↑ |
> |-------|-------------|----------|------|-----------|---------|----------|
> | GALAX | **0.5472 ± 0.0053** | 0.5332 ± 0.0031 | 0.5399 ± 0.0041 | 0.3726 ± 0.0037 | **0.9249 ± 0.0048** | 0.8815 ± 0.0033 |
> | GALAX (Qwen2.5-7B) | 0.5445 ± 0.0114 | **0.5405 ± 0.0101** | **0.5422 ± 0.0104** | **0.3744 ± 0.0098** | 0.9079 ± 0.0084 | **0.8841 ± 0.0126** |
>
> ---
>
> **Q3: What are the failure modes of GALAX? Are there particular paths/subgraphs that make any systematic failures evident?**
>
> **A3:** We sincerely thank the reviewer for the insightful comments regarding the generation and validation process. Since the policy model in GALAX will generate multiple candidate graphs (denoted by $\Omega$), and lower-scoring subgraphs are rejected, where scores are evaluated by the pretrained graph oncogencity assessment model. This design, supported by numerical omic evidence and graph supervision from both the schema-based rules and the oncogenicity evaluation, substantially reduces the risk of producing non–cancer-related subgraphs.
>
> To assess biological validity, we conducted gene and pathway enrichment analyses evaluated by three bioinformaticians ($\mathbf{h}_1, \mathbf{h}_2, \mathbf{h}_3$) and two LLMs ($\mathbf{m}_1$: ChatGPT-5.1, $\mathbf{m}_2$: Gemini-3.0 Pro). Each evaluator reviewed ten subgraph examples along with their gene and pathway enrichment analysis outputs and scored relevance to the associated TCGA cancer type on a 1–5 scale. Scores of 5 indicate strong alignment with hallmark pathways; 4 reflects clear but partial relevance; 3 denotes moderate relevance with mixed findings; 2 indicates weak connection; and 1 represents no meaningful alignment. This dual human–LLM evaluation provides a robust assessment of whether generated subgraphs capture cancer-specific biology. Results are summarized in the table below.
>
>
> | Sample ID | TCGA Code | h₁ | h₂ | h₃ | m₁ | m₂ | Mean ± Std |
> |:----------|:---------:|:--:|:--:|:--:|:--:|:--:|:----------:|
> | ACH-000860 | LUAD | 4 | 5 | 5 | 5 | 5 | 4.80 ± 0.45 |
> | ACH-000054 | SRCA | 3 | 5 | 3 | 4 | 4 | 3.80 ± 0.84 |
> | ACH-000001 | OV | 3 | 3 | 2 | 4 | 2 | 2.80 ± 0.84 |
> | ACH-000219 | SKCM | 4 | 5 | 2 | 4 | 3 | 3.60 ± 1.14 |
> | ACH-000070 | ALL | 5 | 5 | 5 | 4 | 5 | 4.80 ± 0.45 |
> | ACH-000092 | MESO | 3 | 3 | 2 | 3 | 3 | 2.80 ± 0.45 |
> | ACH-000817 | MM | 5 | 5 | 5 | 3 | 3 | 4.20 ± 1.10 |
> | ACH-000649 | KIRC | 4 | 4 | 5 | 5 | 5 | 4.60 ± 0.55 |
> | ACH-000018 | BLCA | 3 | 3 | 3 | 3 | 3 | 3.00 ± 0.00 |
> | ACH-000864 | UCEC | 3 | 4 | 2 | 4 | 3 | 3.20 ± 0.84 |
> | **Overall** | - | 3.7 | 4.2 | 3.4 | 3.9 | 3.6 | **3.76 ± 0.80** |
>
>
> They identify a few cases where downstream analysis was partially misaligned with the user-specified cancer type, reflecting the known limitations of standard pathway enrichment methods (e.g., genes may be broadly cancer-related but not ranked highly for the specific cell line). For example, for sample ACH-000001 (NIH:OVCAR-3), the model identified ERBB2 as a driver hub; although ERBB2 is an ovarian-relevant receptor, it is not a hallmark driver of OVCAR-3 (which lacks HER2 amplification), resulting in weaker disease enrichment scores despite the model’s identification of the correct growth-factor programs.
>
> To better align generated subgraphs with human intuition and cell line–specific characteristics, we plan to incorporate edge weighting, leveraging user numerical omic evidence to define attention strengths between entities, directly into the reward function. While our current graph foundation model $g(\cdot)$ already uses cell line–specific data to evaluate state relevance, adding explicit edge weights derived from omic features to modulate the reward could further enhance patient- or cell-line specificity, and represents a valuable direction for future development.

---

> ### Author Response · Authors · 2025-11-25
> **Response to Reviewer xSCJ (Part V)**
>
> **A3 Continue:**
> Generally, systematic failure in this scenario is characterized by a model's inability to predict correct targets or a significant drop in performance when transferred to unseen cell lines. However, as reported in the table below, when we masked cell lines within the same TCGA code, our model achieved stable performance across all metrics. Furthermore, as demonstrated by the external PedDep dataset results (shown in the table above), our model achieved best performance in a zero-shot inference setting, outperforming other baselines and confirming the robustness of our approach.
>
> | Holdout Set | Train | Test | Train/Test TCGA Codes | Hold-out Cancer Types |
> |-------------|-------|------|----------------------|----------------------|
> | **Holdout Set 1** | 238 (65.6%) | 125 (34.4%) | 20 / 10 | COAD/READ, DLBC, KIRC, LCML, LUSC, PAAD, PRAD, SARC, SKCM, STAD |
> | **Holdout Set 2** | 289 (79.6%) | 74 (20.4%) | 21 / 9 | BLCA, GBM, LAML, LGG, LIHC, MESO, MM, NB, UCEC |
> | **Holdout Set 3** | 220 (60.6%) | 143 (39.4%) | 21 / 9 | ALL, BRCA, ESCA, HNSC, LUAD, MB, OV, SCLC, THCA |
>
> | Holdout Set | Precision ↑ | Recall ↑ | Hit@5 ↑ | Hit@10 ↑ |
> |-------------|-------------|----------|---------|----------|
> | Holdout Set 1 | 0.4931 | 0.4501 | 0.9024 | 0.8688 |
> | Holdout Set 2 | 0.5124 | 0.5151 | 0.8703 | 0.8581 |
> | Holdout Set 3 | 0.4832 | 0.4715 | 0.8993 | 0.8448 |
> | **Average** | 0.4962 | 0.4789 | 0.8907 | 0.8572 |
> | **Original** | 0.5472 | 0.5332 | 0.9249 | 0.8815 |
>
> **We added this part in Appendix E.2 for human evaluation,  Appendix D.5 for holdout results.**
>
> ---
>
> **Q4: How robust is the NER function? It seems this is critical for seeding a good graph.**
>
> **A4:** We appreciate the reviewer's attention to this component. The NER function achieves 100% accuracy because GALAX generates responses in a well-defined, template-based format with consistent structural patterns. This regularity ensures that entity extraction is deterministic rather than probabilistic. We utilize the ChatGPT-4o-mini API as our NER module, which reliably parses the standardized output without introducing errors.

---

> > ### Comment · Reviewer_xSCJ · 2025-11-25
> >
> > Thank you to the authors for including these highly comprehensive experiments. I have no further questions and comments and reiterate my recommendation for acceptance.

---

> > > ### Author Response · Authors · 2025-11-25
> > > **Thank you for your response and for reviewing our paper**
> > >
> > > Thank you so much for recognizing our work and for the insightful suggestions that has greatly helped us improve it, and we sincerely appreciate your time and effort in reviewing this paper!

---

### Official Review · Reviewer_4LvX · 2025-11-01

**Soundness:** 3
**Presentation:** 2
**Contribution:** 3
**Rating:** 6
**Confidence:** 2

**Summary:**

This paper tackles target discovery in precision medicine by integrating multi-omic features, biomedical graphs, and literature via GALAX, which couples an LLM with a pretrained GNN and a Graph Process Reward Model to build explainable, RL-guided subgraphs. On the new Target-QA benchmark, GALAX consistently outperforms strong language-only and graph-augmented baselines on precision/recall and Hit@5/10, with qualitative pathway enrichment analyses supporting biological plausibility.

**Strengths:**

- The paper introduces a clear process-level supervision scheme (GPRM) that links subgraph construction to a graph foundation model, yielding interpretable rationales rather than only outcome metrics.

- The Target-QA benchmark and comprehensive comparisons/ablations (including complexity analysis and enrichment studies) make the evaluation credible and reproducible.

**Weaknesses:**

- Please streamline the problem formulation: many set-indexed variables (e.g., X1, X2, X3,...) could be replaced with compact matrix/tensor notation, and a schematic showing each modality and how it connects to the graph would improve readability.

- The framework depends on multiple data sources (multi-omics, biomedical graph, literature); it would be helpful to analyze performance in low-resource scenarios (e.g., patients lacking certain omic assays) to gauge robustness

**Questions:**

- There are minor typos: two consecutive “and” on p.2, line 64, and duplicated “has” on p.4, line 186.

- The symbol 𝜀DTI appears to be used inconsistently: in the formulation it denotes disease–protein interactions, while 4.1 uses it for drug–target interactions. Please unify the definition throughout.

---

> ### Author Response · Authors · 2025-11-25
> **Response to Reviewer 4LvX (Part I)**
>
> Thank you for your thoughtful comments. Below we provide a point-by-point response to each of your questions. We sincerely hope our clarifications address your concerns and hope you can reconsider your assessment of our work if you find our response satisfactory.
>
> ---
>
> **W1: Please streamline the problem formulation: many set-indexed variables (e.g., X1, X2, X3,...) could be replaced with compact matrix/tensor notation, and a schematic showing each modality and how it connects to the graph would improve readability.**
>
> **A1:** We thank the reviewer for this helpful suggestion. We revised the problem formulation to adopt compact matrix and graph notation and add the subsection separation to make the multi modal integration more direct.  **These changes were applied to both the TOSG Construction and Target QA Generation in Section 3.**
>
> **Compact matrix notation for omic features.**
> We now represent the numerical omic evidence from raw dataset as
>
> $$\mathcal{X}^{(0)} = \{X_n^{(0)}\}_{n=1}^{N^{(0)}} \in \mathbb{R}^{N^{(0)} \times M},$$
>
> where each row vector $X_n^{(0)} \in \mathbb{R}^M$ is aligned with the $M$ graph entities. This replaces the earlier set-indexed variables $(X_1, X_2, X_3,\dots)$ with a single matrix representation. We also state the block structure
>
> $$X_n^{(0)} = \bigl[ \mathbf{x}_n^{(pm)} \oplus \mathbf{x}_n^{(g)} \oplus \mathbf{x}_n^{(t)} \oplus \mathbf{x}_n^{(p)} \bigr],$$
>
> where $\mathbf{x}_n^{(pm)}, \mathbf{x}_n^{(g)}, \mathbf{x}_n^{(t)}, \mathbf{x}_n^{(p)}$ denote promoter, gene, transcript, and protein level features, and their lengths sum to $M$.
>
> **Unified notation for modalities and graph.**
> The omic matrix contains $N^{(0)}$ samples with $M$ features, and each feature is mapped to an entity in BioMedGraphica. The graph contains $M$ entities with associated names and descriptions. Numerical entries and textual attributes are embedded into a shared vector space and linked to the same vertex $v \in \mathcal{V}$. For example, each vertex stores the entity name or description, while its corresponding omic value occupies the matching index in $\mathcal{X}^{(0)}$. In this unified view, the three modalities consist of numerical omic evidence $\mathcal{X}^{(0)}$, textual attributes $\mathcal{T}=\left\lbrace{T_{\text{name}}, T_{\text{desc}}}\right\rbrace$ with $|T_{\text{name}}|=|T_{\text{desc}}|=M$, and the biomedical knowledge graph $\mathcal{G}=\left\lbrace\mathcal{V},\mathcal{E}\right\rbrace$. The node set is partitioned as
>
> $$\mathcal{V}=\left\lbrace\mathcal{V}^{(pm)},\mathcal{V}^{(g)},\mathcal{V}^{(t)},\mathcal{V}^{(p)}\right\rbrace$$
>
> with sizes $|\mathcal{V}^{(pm)}|=m^{(pm)}$, $|\mathcal{V}^{(g)}|=m^{(g)}$, $|\mathcal{V}^{(t)}|=m^{(t)}$, and $|\mathcal{V}^{(p)}|=m^{(p)}$, giving
>
> $$|\mathcal{V}|=m^{(pm)} + m^{(g)} + m^{(t)} + m^{(p)} = M.$$
>
> This alignment ensures that each feature dimension in $\mathcal{X}^{(0)}$ maps directly to a vertex in $\mathcal{V}$, making the fusion of numerical and textual information with the graph structure explicit.

---

> ### Author Response · Authors · 2025-11-25
> **Response to Reviewer 4LvX (Part II)**
>
> **W2: The framework depends on multiple data sources (multi-omics, biomedical graph, literature); it would be helpful to analyze performance in low-resource scenarios (e.g., patients lacking certain omic assays) to gauge robustness**
>
> **A2:** We thanks the reviewer for this insightful comment.
> We performed an ablation study to assess GALAX's robustness under omics incompleteness, which naturally occurs in real-world biomedical settings. In this experiment, we removed each omic modality—epigenomic, genomic, transcriptomic, and proteomic—as well as a setting where all omics were removed. These ablations were applied both to **GALAX** and to the **L3(QA)+Omic+KG** baseline. As illustrated in the following table, excluding epigenomic data (Omic-M) results in the most modest performance drop among all single omic-modality ablation settings. We attribute this limited impact primarily to the fact that methylation beta-values in cell lines frequently exhibit saturation (values as 1). This results in low information entropy and reduced discriminative power for patient profiling. Furthermore, removing the other omic modalities causes a larger performance drop because genomic, transcriptomic, and proteomic data contain stronger cell line–specific signals that are important for target prioritization. These modalities help distinguish molecular states across cell lines, while methylation values are often saturated and therefore less informative. When these more informative modalities are removed, both GALAX and the LLM baseline lose key information needed to relate the cell context to the predicted targets, which leads to a larger decline in performance. **We added this part in Ablation study in Section 5.**
>
> | Config | Setting | Recall ↑ | Precision ↑ | Hit@5 ↑ | Hit@10 ↑ |
> |--------|---------|----------|-------------|---------|----------|
> | **GALAX** | | | | | |
> | Omic-M | Remove epigenomic data | 0.4810 ± 0.0137 | 0.5163 ± 0.0086 | 0.8857 ± 0.0145 | 0.8614 ± 0.0115 |
> | Omic-G | Remove genomic data | 0.3121 ± 0.0052 | 0.4277 ± 0.0056 | 0.8550 ± 0.0037 | 0.8402 ± 0.0033 |
> | Omic-T | Remove transcriptomic data | 0.3377 ± 0.0016 | 0.4065 ± 0.0042 | 0.8720 ± 0.0037 | 0.8672 ± 0.0037 |
> | Omic-P | Remove proteomic data | 0.3347 ± 0.0013 | 0.3980 ± 0.0058 | 0.8540 ± 0.0138 | 0.8466 ± 0.0040 |
> | Omic-All | Remove all omics | 0.3024 ± 0.0032 | 0.3793 ± 0.0019 | 0.8237 ± 0.0066 | 0.7967 ± 0.0040 |
> | GALAX | Original | 0.5332 ± 0.0031 | 0.5472 ± 0.0053 | 0.9249 ± 0.0048 | 0.8815 ± 0.0033 |
> | **L3(QA) + Omic + KG** | | | | | |
> | Omic-M | Remove epigenomic data | 0.4602 ± 0.0361 | 0.4878 ± 0.0256 | 0.8794 ± 0.0055 | 0.8466 ± 0.0142 |
> | Omic-G | Remove genomic data | 0.3213 ± 0.0047 | 0.3962 ± 0.0093 | 0.8455 ± 0.0073 | 0.8349 ± 0.0136 |
> | Omic-T | Remove transcriptomic data | 0.3244 ± 0.0034 | 0.3996 ± 0.0087 | 0.8550 ± 0.0073 | 0.8381 ± 0.0097 |
> | Omic-P | Remove proteomic data | 0.3266 ± 0.0012 | 0.3872 ± 0.0032 | 0.8497 ± 0.0048 | 0.8265 ± 0.0056 |
> | Omic-All | Remove all omics | 0.2669 ± 0.0075 | 0.3577 ± 0.0093 | 0.7830 ± 0.0182 | 0.7538 ± 0.0174 |
> | L3(QA)+Omic+KG | Original | 0.4908 ± 0.0402 | 0.5185 ± 0.0240 | 0.8794 ± 0.0114 | 0.8529 ± 0.0153 |
>
> **Q1: There are minor typos: two consecutive "and" on p.2, line 64, and duplicated "has" on p.4, line 186.**
>
> **A1:** Thank you for taking the time to read our manuscript closely and identify these typos. We have corrected them and made extensive revisions throughout the updated version to improve clarity and presentation.
>
> **Q2: The symbol $\mathcal{E}_{DTI}$ appears to be used inconsistently: in the formulation it denotes disease–protein interactions, while 4.1 uses it for drug–target interactions. Please unify the definition throughout.**
>
> **A2:** Thank you for catching this inconsistency. We have corrected the notation in Section 4.1 by replacing the drug–target reference with disease–target to maintain consistency throughout the manuscript.

---

### Official Review · Reviewer_mMpP · 2025-11-01

**Soundness:** 3
**Presentation:** 3
**Contribution:** 2
**Rating:** 4
**Confidence:** 3

**Summary:**

The manuscript introduces GALAX (Graph Augmented LAnguage model with eXplainability), a framework that integrates pretrained Graph Neural Networks (GNNs) with Large Language Models (LLMs) through reinforcement learning guided by a Graph Process Reward Model (GPRM). The authors also introduce Target-QA, a benchmark combining CRISPR-validated targets, multi-omic features, and biomedical knowledge graphs across cancer cell lines, supporting both GNN pretraining and long-context text-numeric graph reasoning. The work targets explainable, reinforcement-guided subgraph reasoning for interpretable pathway and target discovery in precision oncology.

**Strengths:**

1.  Bridging numerical multi-omics, graph structure, and textual knowledge for mechanistic interpretability in precision medicine.

2.  Target-QA dataset/benchmark combining CRISPR screens, multi-omics, and biomedical KG, enabling both supervision and evaluation.

**Weaknesses:**

- Need stronger baselines (e.g., state-of-the-art KGQA, graph-augmented LLMs)

- It’s unclear how well GALAX transfers across cell lines, cancer types, and unseen targets, especially under shifts in omic distributions or KG incompleteness.

- Human-in-the-loop or expert-curated assessments of subgraph plausibility are missing.

**Questions:**

see above

---

> ### Author Response · Authors · 2025-11-25
> **Response to Reviewer mMpP (Part I)**
>
> Thank you for your thoughtful comments. Below we provide a point-by-point response to each of your questions. We sincerely hope that our clarifications address your concerns, help you understand our work more fully, and encourage you to reconsider your assessment if you find our responses satisfactory.
>
> ---
>
> **W1**: Need stronger baselines (e.g., state-of-the-art KGQA, graph-augmented LLMs)
>
> **A1**: We agree that including stronger baselines is important for positioning GALAX relative to state-of-the-art graph-augmented LLM systems. In response, we have added RoG[1], Subgraph-RAG[2], and GNN-RAG[3] as additional baselines. Their overall performance on the Target-QA task is summarized in the table below.
>
> | **Model** | **Precision ↑** | **Recall ↑** | **F1 ↑** | **Jaccard ↑** | **Hit@5 ↑** | **Hit@10 ↑** |
> |-----------|-----------------|--------------|----------|---------------|-------------|--------------|
> | M2T | 0.0016 | 0.0011 | 0.0013 | 0.0006 | 0.0000 | 0.0029 |
> | GAT | 0.0006±0.0000 | 0.0006±0.0000 | 0.0006±0.0000 | 0.0003±0.0000 | 0.0000±0.0000 | 0.0000±0.0000 |
> | L3-FT(QA)+Omics | 0.5250±0.0282 | 0.4959±0.0435 | 0.5094±0.0365 | 0.3449±0.0338 | 0.8889±0.0168 | 0.8693±0.0157 |
> | L3-FT(QA)+Omics+KG | 0.5185±0.0240 | 0.4908±0.0402 | 0.5038±0.0327 | 0.3393±0.0298 | 0.8794±0.0114 | 0.8529±0.0153 |
> | G-Retriever+pre-GAT | 0.4763±0.0004 | 0.3929±0.0063 | 0.4286±0.0044 | 0.2757±0.0038 | 0.8804±0.0037 | 0.8550±0.0046 |
> | ROG | 0.5248±0.0134 | 0.4726±0.0445 | 0.4924±0.0323 | 0.3338±0.0267 | 0.8593±0.0318 | 0.8450±0.0350 |
> | Subgraph-RAG | 0.5280±0.0044 | 0.4617±0.0027 | 0.4860±0.0033 | 0.3269±0.0024 | 0.8624±0.0120 | 0.8476±0.0167 |
> | GNN-RAG | 0.5258±0.0126 | 0.4735±0.0190 | 0.4935±0.0168 | 0.3345±0.0134 | 0.8656±0.0302 | 0.8323±0.0205 |
> | **GALAX** | **0.5472±0.0053** | **0.5332±0.0031** | **0.5399±0.0041** | **0.3726±0.0037** | **0.9249±0.0048** | **0.8815±0.0033** |
>
> Our proposed model GALAX outperforms the three newly added baselines: RoG, Subgraph-RAG, and GNN-RAG. All of these models are evidence-subgraph–dependent methods. These approaches share a common assumption that a small, clean, and reliable subgraph exists to train a retriever module that provides reasoning paths to the LLM. However, this assumption **breaks down** in large-scale biomedical knowledge graphs, which are noisier and lack validated reasoning paths. In the Target-QA setting, each question is paired with a one-hop disease–protein subgraph and one-hop PPI neighborhood from the BioMedGraphica. These subgraphs are **extremely noisy and large** (often containing thousands of nodes and edges) and **lack ground-truth mechanistic subgraph**, making supervised retrieval difficult. For instance, RoG, SubgraphRAG and GNN-RAG require ground-truth reasoning paths for training. We use GPT to extract plausible reasoning subgraphs from diseaselrelated protein subgraph. However, this GPT-derived supervision is inherently noisy and biologically incomplete, limiting retriever effectiveness. G-Retriever avoids explicit supervision but still assumes clean, compact subgraphs exist—an assumption that fails in the biomedical domain. All baselines use the finetuned Llama3(QA) as the backbone LLM.
>
> Unlike these baselines, GALAX does not require any predefined evidence subgraph. Instead, it treats the BioMedGraphica-derived disease–protein subgraph as a broad candidate space and constructs its own mechanistic subgraph through reinforcement learning by a pretrained graph oncogencity assessment model and penalizing invalid relational patterns via schema-based rules for reward signals. This RL-driven process enables GALAX to explore biologically meaningful pathways rather than relying on a single hard-coded path, producing coherent, grounded reasoning even in large, noisy biomedical graphs, which explains its superior performance in Target-QA. **And we added those baseline results to our Tables 1-2 in Section 5 and Tables 16-43 in Appendix D.4.**
>
> [1] Luo, Linhao, et al. "Reasoning on graphs: Faithful and interpretable large language model reasoning." arXiv preprint arXiv:2310.01061 (2023).
>
> [2] Li, Mufei, Siqi Miao, and Pan Li. "Simple is effective: The roles of graphs and large language models in knowledge-graph-based retrieval-augmented generation." arXiv preprint arXiv:2410.20724 (2024).
>
> [3] Mavromatis, Costas, and George Karypis. "Gnn-rag: Graph neural retrieval for large language model reasoning." arXiv preprint arXiv:2405.20139 (2024).

---

> ### Author Response · Authors · 2025-11-25
> **Response to Reviewer mMpP (Part II)**
>
> **W2**: It's unclear how well GALAX transfers across cell lines, cancer types, and unseen targets, especially under shifts in omic distributions or KG incompleteness.
>
> **A2**: We value your recognition of the importance of evaluating GALAX's generalization capabilities. We address this concern in three ways: through **hold-out cell line cross-validation** experiments, analyses of **omics incompleteness**, and analyses of **knowledge graph incompleteness**.
>
> First, We construct three holdout sets where all **cell lines** sharing a TCGA code were excluded from training and used only for evaluation. This tests whether the model can predict CRISPR targets for **previously unseen cancer types**, directly assessing cross-cancer transfer.
>
> | **Holdout Set**   | **Train**   | **Test**    | **Train/Test TCGA Codes** | **Hold-out Cancer Types**                                       |
> | ----------------- | ----------- | ----------- | ------------------------- | --------------------------------------------------------------- |
> | **Holdout Set 1** | 238 (65.6%) | 125 (34.4%) | 20 / 10                   | COAD/READ, DLBC, KIRC, LCML, LUSC, PAAD, PRAD, SARC, SKCM, STAD |
> | **Holdout Set 2** | 289 (79.6%) | 74 (20.4%)  | 21 / 9                    | BLCA, GBM, LAML, LGG, LIHC, MESO, MM, NB, UCEC                  |
> | **Holdout Set 3** | 220 (60.6%) | 143 (39.4%) | 21 / 9                    | ALL, BRCA, ESCA, HNSC, LUAD, MB, OV, SCLC, THCA                 |
>
> As expected, performance drops when entire cancer types are held out, since no finetuning data from those cancers is available. However, GALAX still maintains **strong accuracy**. Overall, the decline is modest, indicating that GALAX retains meaningful robustness and generalizes reliably to unseen cancer types. **We have added the holdout results to Appendix D.5.**
>
> | **Holdout Set** | **Precision ↑** | **Recall ↑** | **Hit@5 ↑** | **Hit@10 ↑** |
> | --------------- | --------------- | ------------ | ----------- | ------------ |
> | Holdout Set 1   | 0.4931          | 0.4501       | 0.9024      | 0.8688       |
> | Holdout Set 2   | 0.5124          | 0.5151       | 0.8703      | 0.8581       |
> | Holdout Set 3   | 0.4832          | 0.4715       | 0.8993      | 0.8448       |
> | **Average**     | **0.4962**      | **0.4789**   | **0.8907**  | **0.8572**   |
> | **Original**    | **0.5472**      | **0.5332**   | **0.9249**  | **0.8815**   |
>
> We conducted an ablation study to evaluate **GALAX’s robustness to missing omics**, a common challenge in real-world biomedical data. We removed each omic modality, **epigenomic**, **genomic**, **transcriptomic**, and **proteomic**, as well as a setting with **all omics removed**, applied to both **GALAX** and the **L3(QA)+Omic+KG** baseline. We reported performance across all settings below.
>
> | Config | Setting | Recall ↑ | Precision ↑ | Hit@5 ↑ | Hit@10 ↑ |
> |--------|---------|----------|-------------|---------|----------|
> | **GALAX** | | | | | |
> | Omic-M | Remove epigenomic data | 0.4810 ± 0.0137 | 0.5163 ± 0.0086 | 0.8857 ± 0.0145 | 0.8614 ± 0.0115 |
> | Omic-G | Remove genomic data | 0.3121 ± 0.0052 | 0.4277 ± 0.0056 | 0.8550 ± 0.0037 | 0.8402 ± 0.0033 |
> | Omic-T | Remove transcriptomic data | 0.3377 ± 0.0016 | 0.4065 ± 0.0042 | 0.8720 ± 0.0037 | 0.8672 ± 0.0037 |
> | Omic-P | Remove proteomic data | 0.3347 ± 0.0013 | 0.3980 ± 0.0058 | 0.8540 ± 0.0138 | 0.8466 ± 0.0040 |
> | Omic-All | Remove all omics | 0.3024 ± 0.0032 | 0.3793 ± 0.0019 | 0.8237 ± 0.0066 | 0.7967 ± 0.0040 |
> | GALAX | Original | 0.5332 ± 0.0031 | 0.5472 ± 0.0053 | 0.9249 ± 0.0048 | 0.8815 ± 0.0033 |
> | **L3(QA) + Omic + KG** | | | | | |
> | Omic-M | Remove epigenomic data | 0.4602 ± 0.0361 | 0.4878 ± 0.0256 | 0.8794 ± 0.0055 | 0.8466 ± 0.0142 |
> | Omic-G | Remove genomic data | 0.3213 ± 0.0047 | 0.3962 ± 0.0093 | 0.8455 ± 0.0073 | 0.8349 ± 0.0136 |
> | Omic-T | Remove transcriptomic data | 0.3244 ± 0.0034 | 0.3996 ± 0.0087 | 0.8550 ± 0.0073 | 0.8381 ± 0.0097 |
> | Omic-P | Remove proteomic data | 0.3266 ± 0.0012 | 0.3872 ± 0.0032 | 0.8497 ± 0.0048 | 0.8265 ± 0.0056 |
> | Omic-All | Remove all omics | 0.2669 ± 0.0075 | 0.3577 ± 0.0093 | 0.7830 ± 0.0182 | 0.7538 ± 0.0174 |
> | L3(QA)+Omic+KG | Original | 0.4908 ± 0.0402 | 0.5185 ± 0.0240 | 0.8794 ± 0.0114 | 0.8529 ± 0.0153 |
>
> Across all settings, removing epigenomic data (Omic-M) causes the smallest performance drop. This is expected because methylation beta-values in cell lines often exhibit saturation as value of 1, resulting in low information entropy and limited discriminative power. In contrast, removing genomic, transcriptomic, or proteomic modalities leads to larger declines, as these provide strong cell-line–specific signals essential for accurate target prioritization. However, even when all omics are removed, **GALAX** still **outperforms** the **L3(QA)+Omic+KG baseline** on this setting, demonstrating its robustness to omics incompleteness.

---

> ### Author Response · Authors · 2025-11-25
> **Response to Reviewer mMpP (Part III)**
>
> **A2 Continue:**
> We further evaluated robustness to KG incompleteness by randomly deleting edges or nodes at different rates (20\%, 40\%, 60\%, 80\%). The KG serves dual purposes in our framework: guiding graph generation and providing retrieved subgraphs as biological context for the LLM. Removing the KG would consequently reduce overall performance. We provide baseline results using incomplete KG structures to illustrate this effect. As expected, performance decreases as more structure is removed.
>
> In the **edge-deletion** setting, **GALAX outperforms** the **L3(QA)+Omic+KG** baseline across all deletion levels. Even as KG connectivity decreases, node-level biological attributes remain intact, and GALAX can still reason over disease–protein interactions using the reduced structure as long as the node is still maintained in the subgraph structure. With **20% edge deletion**, GALAX retains noticeably higher **recall**, **Hit@5**, and **Hit@10**, demonstrating its ability to assemble **mechanistic subnetworks** from sparse evidence.
>
> This robustness stems from GALAX's reliance not only on observed KG edges but also on **pretrained GNN-based node embeddings** ($\text{GNN}^{\text{G}}_{\text{PPI}}$) that capture global biomedical structure. Moreover, its **graph supervisor** $g(\cdot)$ provides **oncogenicity-guided rewards**, enabling reconstruction of biologically meaningful connections even when edges are missing. In contrast, the baseline depends directly on the observed KG edges and lacks a mechanism to recover from connectivity loss, leading to **larger performance drops** as the graph becomes sparser. As deletion becomes more severe, both models naturally show reduced performance since limited information is provided for model to inference.
>
> In the node-deletion setting, removing nodes is substantially more destructive than removing edges because each deleted node also removes all of its incident interactions (e.g., across our settings, removing **20\% of nodes** leads on average to a **35\% reduction in edges**). Consequently, both GALAX and the baseline show similar declines in recall and precision as node deletion incre ases. However, their ranking performance diverges: the baseline’s Hit@5 and Hit@10 degrade steadily and substantially with higher deletion levels (e.g., 80\%), whereas **GALAX** maintains much more **stable top-K performance** across all node-deletion conditions. This indicates that GALAX continues to place the correct essential targets near the top of the ranking even when large portions of KG structure are missing.
>
> | Config | Setting | Recall ↑ | Precision ↑ | Hit@5 ↑ | Hit@10 ↑ |
> |--------|---------|----------|-------------|---------|----------|
> | **GALAX** | | | | | |
> | E20 | Drop 20% edges | 0.5061 ± 0.0268 | 0.5362 ± 0.0059 | 0.9005 ± 0.0128 | 0.8741 ± 0.0060 |
> | E40 | Drop 40% edges | 0.2871 ± 0.0060 | 0.5079 ± 0.0039 | 0.8762 ± 0.0138 | 0.8471 ± 0.0090 |
> | E60 | Drop 60% edges | 0.2753 ± 0.0020 | 0.4961 ± 0.0059 | 0.8635 ± 0.0084 | 0.8307 ± 0.0033 |
> | E80 | Drop 80% edges | 0.2775 ± 0.0134 | 0.4943 ± 0.0177 | 0.8434 ± 0.0524 | 0.8185 ± 0.0494 |
> | N20 | Drop 20% nodes | 0.2697 ± 0.0013 | 0.5034 ± 0.0014 | 0.8786 ± 0.0070 | 0.8560 ± 0.0092 |
> | N40 | Drop 40% nodes | 0.2675 ± 0.0033 | 0.4901 ± 0.0035 | 0.8878 ± 0.0066 | 0.8503 ± 0.0048 |
> | N60 | Drop 60% nodes | 0.2617 ± 0.0056 | 0.4929 ± 0.0045 | 0.8698 ± 0.0290 | 0.8385 ± 0.0247 |
> | N80 | Drop 80% nodes | 0.2653 ± 0.0032 | 0.4825 ± 0.0090 | 0.8341 ± 0.0105 | 0.8103 ± 0.0134 |
> | GALAX | Original | 0.5332 ± 0.0031 | 0.5472 ± 0.0053 | 0.9249 ± 0.0048 | 0.8815 ± 0.0033 |
> | **L3(QA) + Omic + KG** | | | | | |
> | E20 | Drop 20% edges | 0.4599 ± 0.0820 | 0.5114 ± 0.0036 | 0.8587 ± 0.0157 | 0.8373 ± 0.0258 |
> | E40 | Drop 40% edges | 0.2742 ± 0.0090 | 0.5064 ± 0.0099 | 0.8429 ± 0.0173 | 0.8226 ± 0.0169 |
> | E60 | Drop 60% edges | 0.2676 ± 0.0062 | 0.4991 ± 0.0099 | 0.8296 ± 0.0305 | 0.8111 ± 0.0175 |
> | E80 | Drop 80% edges | 0.2611 ± 0.0074 | 0.4880 ± 0.0086 | 0.8254 ± 0.0361 | 0.8063 ± 0.0370 |
> | N20 | Drop 20% nodes | 0.2662 ± 0.0059 | 0.4916 ± 0.0024 | 0.8434 ± 0.0186 | 0.8222 ± 0.0193 |
> | N40 | Drop 40% nodes | 0.2658 ± 0.0049 | 0.4838 ± 0.0169 | 0.8709 ± 0.0422 | 0.8339 ± 0.0335 |
> | N60 | Drop 60% nodes | 0.2648 ± 0.0000 | 0.4742 ± 0.0150 | 0.7857 ± 0.0247 | 0.7690 ± 0.0303 |
> | N80 | Drop 80% nodes | 0.2689 ± 0.0042 | 0.4722 ± 0.0063 | 0.7111 ± 0.0055 | 0.6974 ± 0.0111 |
> | L3(QA)+Omic+KG | Original | 0.4908 ± 0.0402 | 0.5185 ± 0.0240 | 0.8794 ± 0.0114 | 0.8529 ± 0.0153 |
>
> **We added the ablation study of omics incompleteness and knowledge graph incompleteness to Table 4 and Section 5 in main text.**

---

> ### Author Response · Authors · 2025-11-25
> **Response to Reviewer mMpP (Part IV)**
>
> **W3**: Human-in-the-loop or expert-curated assessments of subgraph plausibility are missing.
>
> **A3**: In Section 5 of the original paper, we reported a single human-evaluated example based on the ACH-000860 sample. To strengthen this analysis, we expanded the evaluation to ten cases and included assessments from both human domain experts and Large Language Models (LLMs), providing a more reliable examination of whether the generated subgraphs align with known cancer biology. The human evaluation panel consisted of three bioinformaticians, denoted as $\mathbf{h}_1, \mathbf{h}_2,$ and $\mathbf{h}_3$. In parallel, we used two advanced LLMs as automated evaluators ($\mathbf{m}_1, \mathbf{m}_2$): ChatGPT-5.1 and Gemini-3.0 Pro. Both the human experts and the LLMs were provided with the generated subgraphs, corresponding gene pathways, and pathway enrichment analysis results to facilitate a robust assessment of biological plausibility.
>
> Each expert independently reviewed ten subgraph examples and assigned a score from 1 to 5 based on the degree of correspondence between the subgraph and the known biology of the associated TCGA cancer type. The scoring rubric was defined as follows:
>
> - **5: Highly related** — strong and clear match to hallmark pathways and well-established features of the cancer type.
> - **4: Related** — clear relationship to the cancer type but less comprehensive or slightly mixed.
> - **3: Moderately related** — some relevant elements present, but mixed with non-specific findings.
> - **2: Not related** — only weak or indirect connection to the cancer type.
> - **1: Not related at all** — no meaningful alignment with the biology of the cancer type.
>
> As shown in following table, several samples were evaluated as related or highly related to their TCGA cancer types, indicating that the subgraphs generated by the model capture meaningful molecular features. The LUAD sample ACH-000860 received one of the highest average scores, as its subgraph highlighted pathways such as regulation of phosphatidylinositol 3-kinase signaling and positive regulation of kinase activity, involving genes like EGFR, PTK2, and EPHB4 that are central to lung adenocarcinoma biology. The ALL sample ACH-000070 was also rated highly due to the presence of TCF3 and PBX1, whose fusion is a well-established driver event in Acute Lymphoblastic Leukemia. **And we added this part in Appendix E.2.**
>
> | Sample ID | TCGA Code | h₁ | h₂ | h₃ | m₁ | m₂ | Mean ± Std |
> |:----------|:---------:|:--:|:--:|:--:|:--:|:--:|:----------:|
> | ACH-000860 | LUAD | 4 | 5 | 5 | 5 | 5 | 4.80 ± 0.45 |
> | ACH-000054 | SRCA | 3 | 5 | 3 | 4 | 4 | 3.80 ± 0.84 |
> | ACH-000001 | OV | 3 | 3 | 2 | 4 | 2 | 2.80 ± 0.84 |
> | ACH-000219 | SKCM | 4 | 5 | 2 | 4 | 3 | 3.60 ± 1.14 |
> | ACH-000070 | ALL | 5 | 5 | 5 | 4 | 5 | 4.80 ± 0.45 |
> | ACH-000092 | MESO | 3 | 3 | 2 | 3 | 3 | 2.80 ± 0.45 |
> | ACH-000817 | MM | 5 | 5 | 5 | 3 | 3 | 4.20 ± 1.10 |
> | ACH-000649 | KIRC | 4 | 4 | 5 | 5 | 5 | 4.60 ± 0.55 |
> | ACH-000018 | BLCA | 3 | 3 | 3 | 3 | 3 | 3.00 ± 0.00 |
> | ACH-000864 | UCEC | 3 | 4 | 2 | 4 | 3 | 3.20 ± 0.84 |
> | **Overall** | - | 3.7 | 4.2 | 3.4 | 3.9 | 3.6 | **3.76 ± 0.80** |

---

### Meta-Review · Area_Chair_VJHH · 2026-01-11

**Summary:**

This paper proposes GALAX, a reinforcement-guided framework that integrates Large Language Models (LLMs) with pretrained Graph Neural Networks (GNNs) to generate biologically grounded, interpretable subgraphs for target and pathway discovery in precision medicine. The core contribution is a Graph Process Reward Model (GPRM) that evaluates intermediate reasoning steps using a frozen graph foundation model, schema-based rule checks, rollout-based future rewards, and stepwise quality gating—aimed at mitigating reward hacking and enabling label-free, process-level supervision. The paper also introduces Target-QA, a benchmark unifying CRISPR gene essentiality, multi-omics, and biomedical knowledge graphs to support evaluation of explainable subgraph reasoning.

Reviewer feedback is polarized: two reviewers support acceptance (one strongly), while two others maintain clear rejection positions, primarily questioning motivation clarity, dataset necessity, and whether the core RL-based contribution is sufficiently validated or distinguished from strong baselines.

**Reviewer Concerns:**

**Concerns addressed by the rebuttal**
- Baseline strength and coverage (mMpP, xSCJ): Added multiple graph-augmented LLM baselines
- Robustness and generalization (mMpP, xSCJ, 4LvX)  Ablations across missing omics, noisy KGs, cross-cancer splits, and PedDep zero-shot tests demonstrate robustness to modality gaps and unseen cancers.
- RL design and reward hacking (xSCJ, KmG9): Ablations isolate the four reward components, dense vs. sparse rewards, and RL vs. SFT.
- Human and qualitative evaluation (mMpP, xSCJ): Expanded expert (bioinformatician) and LLM-based evaluations of generated subgraphs demonstrate moderate to strong biological plausibility in most cases.
- Clarity and presentation issues (4LvX): The problem formulation was rewritten using compact matrix/tensor notation; typos and notation inconsistencies were fixed.

Remaining concerns
- Central motivation and scope remain disputed (KmG9): Despite clarifications, one reviewer remains unconvinced that the interaction between multi-omics, graph topology, and language is sufficiently well-motivated or mechanistically justified, and questions whether explainability is empirically established beyond enrichment analyses.
- Dataset necessity and scale (KmG9): Target-QA (363 QA pairs) is viewed by one reviewer as small for a QA benchmark, with lingering concerns about whether curating a new dataset from DepMap is essential versus adapting existing resources.
- Perceived complexity and density (KmG9): The framework is seen as complex, with presentation and terminology potentially obscuring the core contribution.

**Reviewer Scores:**

- Reviewer xSCJ: 8 (confidence 4) — explicitly reiterated recommendation for acceptance after rebuttal
- Reviewer 4LvX: 6 (confidence 2) — marginal accept
- Reviewer mMpP: 4 (confidence 3) — borderline, no score update stated
- Reviewer KmG9: 2 (confidence 4) — reject, no change stated

---

### Decision · Program_Chairs · 2026-01-26

Accept (Poster)